# Neurodegenerative *VPS41* variants inhibit HOPS function and mTORC1-dependent TFEB/TFE3 regulation

Reini E N van der Welle[1] (iD), Rebekah Jobling[2], Christian Burns[3], Paolo Sanza[1], Jan A van der Beek[1] (iD), Alfonso Fasano[4,5,6,7], Lan Chen[3], Fried J Zwartkruis[8] (iD), Susan Zwakenberg[8], Edward F Griffin[9,10], Corlinda ten Brink[1], Tineke Veenendaal[1], Nalan Liv[1], Conny M A van Ravenswaaij-Arts[11], Henny H Lemmink[11] (iD), Rolph Pfundt[12], Susan Blaser[13], Carolina Sepulveda[4,5], Andres M Lozano[6,7,14,15], Grace Yoon[2], Teresa Santiago-Sim[16], Cedric S Asensio[3] (iD), Guy A Caldwell[9,10] (iD), Kim A Caldwell[9,10] (iD), David Chitayat[2,17,*] (iD) & Judith Klumperman[1,**] (iD)

## Abstract

Vacuolar protein sorting 41 (VPS41) is as part of the Homotypic fusion and Protein Sorting (HOPS) complex required for lysosomal fusion events and, independent of HOPS, for regulated secretion. Here, we report three patients with compound heterozygous mutations in *VPS41* (*VPS41^S285P* and *VPS41^R662\**; *VPS41^c.1423-2A>G* and *VPS41^R662\**) displaying neurodegeneration with ataxia and dystonia. Cellular consequences were investigated in patient fibroblasts and *VPS41*-depleted HeLa cells. All mutants prevented formation of a functional HOPS complex, causing delayed lysosomal delivery of endocytic and autophagic cargo. By contrast, *VPS41^S285P* enabled regulated secretion. Strikingly, loss of VPS41 function caused a cytosolic redistribution of mTORC1, continuous nuclear localization of Transcription Factor E3 (TFE3), enhanced levels of LC3II, and a reduced autophagic response to nutrient starvation. Phosphorylation of mTORC1 substrates S6K1 and 4EBP1 was not affected. In a *C. elegans* model of Parkinson's disease, co-expression of *VPS41^S285P*/*VPS41^R662\** abolished the neuroprotective function of VPS41 against α-synuclein aggregates. We conclude that the *VPS41* variants specifically abrogate HOPS function, which interferes with the TFEB/TFE3 axis of mTORC1 signaling, and cause a neurodegenerative disease.

**Keywords** Autophagy; HOPS complex; lysosome-associated disorder; mTORC1; TFEB/TFE3

**Subject Categories** Genetics, Gene Therapy & Genetic Disease; Neuroscience

## Introduction

While lysosomes are responsible for the degradation and recycling of intra- and extracellular substrates (Saftig & Klumperman, 2009), they are increasingly recognized as regulators of cellular homeostasis and key players in nutrient sensing and transcriptional

1   Section Cell Biology, Center for Molecular Medicine, Institute of Biomembranes, University Medical Center Utrecht, Utrecht University, Utrecht, The Netherlands
2   Department of Pediatrics, Division of Clinical and Metabolic Genetics, The Hospital for Sick Children, University of Toronto, Toronto, ON, Canada
3   Department of Biological Sciences, Division of Natural Sciences and Mathematics, University of Denver, Denver, CO, USA
4   Edmond J. Safra Program in Parkinson's Disease, Morton and Gloria Shulman Movement Disorders Clinic, Toronto Western Hospital, UHN, Toronto, ON, Canada
5   Division of Neurology, University of Toronto, Toronto, ON, Canada
6   Krembil Brain Institute, Toronto, ON, Canada
7   Center for Advancing Neurotechnological Innovation to Application (CRANIA), Toronto, ON, Canada
8   Section Molecular Cancer Research, Center for Molecular Medicine, University Medical Center Utrecht, Utrecht University, Utrecht, The Netherlands
9   Department of Biological Sciences, The University of Alabama, Tuscaloosa, AL, USA
10  Department of Neurology, Center for Neurodegeneration and Experimental Therapeutics, Nathan Shock Center for Basic Research in the Biology of Aging, University of Alabama at Birmingham School of Medicine, Birmingham, AL, USA
11  Department of Genetics, University Medical Center Groningen, University of Groningen, Groningen, The Netherlands
12  Department of Human Genetics, Radboud University Medical Center, Nijmegen, The Netherlands
13  Department of Diagnostic Imaging, Hospital for Sick Children, Toronto, ON, Canada
14  Department of Neurosurgery, Toronto Western Hospital, UHN, Toronto, ON, Canada
15  University of Toronto, Toronto, ON, Canada
16  GeneDx, Inc, Gaithersburg, MD, USA
17  The Prenatal Diagnosis and Medical Genetics Program, Department of Obstetrics and Gynecology, University of Toronto, Toronto, ON, Canada
    *Corresponding author. Tel: +1 416 586 4523; E-mail: david.chitayat@sinaihealth.ca
    **Corresponding author. Tel: +31 88 7556550; E-mail: J.klumperman@umcutrecht.nl

regulation (Luzio *et al*, 2007; Richardson *et al*, 2010; Settembre *et al*, 2015; Mony *et al*, 2016). Hence, the digestive properties of lysosomes provide them with a major role in the control of cellular metabolism and nutrient homeostasis. To accomplish this, HOPS (Homotypic Fusion and Protein Sorting), a multisubunit tethering complex, regulates fusion of lysosomes with endosomes and autophagosomes (Seals *et al*, 2000; Wurmser *et al*, 2000; Caplan *et al*, 2001; Pols *et al*, 2013a; Kant *et al*, 2015; Beek *et al*, 2019).

Vacuolar Protein Sorting 41 (VPS41) (*VPS41*, found on Chromosomal location; Chr7p14.1) is a defining component of HOPS, and a 100 kD protein that contains a WD40, TRP-like (tetratricopeptide repeat), CHCR (Clathrin Heavy Chain Repeat), and RING (Really Interesting New Gene)-H2 Zinc Finger domain (Radisky *et al*, 1997; Nickerson *et al*, 2009). *VPS41* knockout mice die early in utero, signifying VPS41 as an essential protein for embryonic development (Aoyama *et al*, 2012). HOPS-associated VPS41 is recruited to endosomes by binding to Rab7 and its interactor Rab interacting lysosomal protein, and to lysosomes by interacting with Arf-like protein 8b (Arl8b; Lin *et al*, 2014; Khatter *et al*, 2015). Depletion of VPS41 impairs HOPS-dependent delivery of endocytic cargo to lysosomes and causes a defect in autophagic flux (Takáts *et al*, 2009; Pols *et al*, 2013a). Independent of HOPS, VPS41 is required for transport of lysosomal membrane proteins from the *trans*-Golgi-Network (TGN) to lysosomes (Swetha *et al*, 2011; Pols *et al*, 2013b), akin to the Alkaline Phosphatase (ALP) pathway in yeast (Cowles *et al*, 1997a; Rehling *et al*, 1999; Darsow *et al*, 2001; Angers & Merz, 2009; Cabrera *et al*, 2010). Furthermore, in secretory cells and neurons, VPS41 is required for regulated secretion of neuropeptides (Burgess & Kelly, 1987; Orci *et al*, 1987; Tooze & Huttner, 1990; Eaton *et al*, 2000; Asensio *et al*, 2010; Asensio *et al*, 2013; Hummer *et al*, 2017). Together, these data show that VPS41, as part of HOPS, is required for lysosomal fusion events and, independent of HOPS, for transport of lysosomal membrane proteins and regulated secretion.

Interestingly, VPS41 overexpression protects dopaminergic neurons against neurodegeneration. This was shown in a transgenic *C. elegans* model of Parkinson's disease, in which α-synuclein is locally overexpressed (Hamamichi *et al*, 2008; Ruan *et al*, 2010; Harrington *et al*, 2012), and in human neuroglioma cells overexpressing α-synuclein (Harrington *et al*, 2012). Neuroprotection against α-synuclein requires interaction of VPS41 with Rab7 and adaptor protein-3 (AP-3) (Griffin *et al*, 2018). Recent studies in *C. elegans* showed that overexpression of human VPS41 also mitigates Aβ-induced neurodegeneration of glutamatergic neurons. This requires the small GTPase Arl8b rather than Rab7 or AP-3, indicating that VPS41, through different interaction partners, can trigger divergent neuroprotective mechanisms against Parkinson's disease and Alzheimer's disease (Griffin *et al*, 2018). However, how VPS41's function leads to neuroprotection remains to be elucidated.

Naturally occurring SNPs in *VPS41* have been described (T52R, T146P and A187T; Harrington *et al*, 2012; Ibarrola-Villava *et al*, 2015), and very recently a single patient with early onset dystonia and a homozygous canonical splice site variant in *VPS41* was identified (Steel *et al*, 2020). In this patient, cDNA studies demonstrated that the variant leads to in-frame skipping of exon 7. In our current study, we present three patients with severe neurological features (e.g., ataxia and dystonia accompanied by retinal dystrophy and mental retardation with brain MRI findings of cerebellar atrophy and thin corpus callosum) of unknown etiology. Exome sequencing

showed that the patients were compound heterozygous for variants in *VPS41* [NM_014396.19: c.853T > C, NP_055211.2: p. Ser285Pro (S285P); NM_014396.6: c.1984C > T, NP_055211.2: p. Arg662Stop (R662*); NM_014396.3: c.1423-2A > G, r.(spl?)]. At the cellular level, we show that expression of these *VPS41* variants prevents the formation of a functional HOPS complex, leading to a kinetic defect in the delivery of endocytosed and autophagic cargo to lysosomes. In addition, we find an inhibition of the mechanistic target of rapamycin complex 1 (mTORC1) toward TFEB/TFE3 and concomitantly a constitutive high level of autophagy with a failure to respond to changing nutrient conditions. Finally, we show that compound expression of VPS41[S285P] and VPS41[R662*] abolishes the neuroprotective effect of VPS41 in the *C. elegans* model of Parkinson's disease. To the best of our knowledge, this is the first study on the cellular consequences of biallelic *VPS41* variants in patients displaying neurological manifestations. Our molecular analysis shows that these mutations result in an unexpected defect in mTORC1 signaling, specifically in the TFEB/TFE3 axis regulating autophagy.

# Patients

### Clinical presentation of three patients with biallelic variants in *VPS41*

Family 1 (Fig EV1A)—Patients 1 and 2, two brothers, born via spontaneous and vaginal delivery, at term, to healthy and non-consanguineous parents. Their birth weights were 3.03 kg (10th–50th centile) and 3.34 kg (10th–50th centile), respectively. The birth length and head circumferences were not recorded. No abnormalities were noted after birth, and the babies were discharged on time. Both presented with neonatal hypotonia and poor eye contact and fixation at 2 months of age and both had ophthalmological examination showing hypopigmented and underdeveloped retina which disappeared at 1 year of age. Physical examination at this stage showed a high forehead with frontal bossing, retrognathia, deep set eyes, short and pointed nose, and prominent ears. There was no organomegaly. In infancy, they were noted to have global developmental delay, poor muscle tone, and marked intentional tremor which further impaired their fine and gross motor skills. Both brothers developed progressive spasticity of the lower limbs with some coarsening of the facial features more so in patient 2 with puffy eyelids, heavy eyebrows, and thick lips. Both brothers showed absent deep tendon reflexes (DTRs), and their plantars were extensor. Both developed upper extremity tremor and significant lower limbs' spasticity and ataxia (Movies EV1–EV3). Extensive investigation for metabolic diseases, including lysosomal and mitochondrial disorders, showed no detectable abnormalities. No urinary mucopolysaccharides and oligosaccharides were detected. Cerebrospinal fluid analysis for neurotransmitter levels showed no abnormalities. Brain MRI done in infancy on both brothers showed mild hypomyelination (Fig 1A and B). However, a repeat brain MRI study on the older brother (patient 1) at 4 years 7 months showed thin corpus callosum with a saber-shape configuration (Fig 1A) and the vermis, although normal in configuration and size initially, demonstrated volume loss on follow-up examinations. Furthermore, brain MRI done on the younger brother (patient 2) at 10 years of

age confirmed the thin corpus callosum, mild progression of the cerebellar atrophy, bilateral hypointensity in the globus pallidus (GP) compatible with early iron deposition, and abnormal hyperintensity in the dentate nucleus (Fig 1B). The overall findings were consistent with neurodegenerative rather than solely malformative brain disease. The growth charts for head circumference were within normal range (Fig EV1B).

DNA analyses using a gene panel including genes associated with mental retardation and dystonia as well as genes associated with iron deposition in the basal ganglia showed no detectable variants (Appendix Table S1). Microarray analysis (IDT xGen Exome Research Panel v1.0) revealed a *de novo* duplication of 2.568 Mb at 1q21.1 [GRCh37/hg19 chr1:145,804,790–148,817,029] in patient 1 only. This region partially overlaps with the 1q21.1 microduplication syndrome and is of unknown significance (Brunetti-pierri *et al*, 2008). Whole exome sequencing done on the brothers and their parents showed that both patients are compound heterozygote for variants in the *VPS41* gene in trans; they carried a missense variant in the WD40 domain [Chr7(GRCh37)[(NM_014396.3:c.853T > C, NP_055211.2:pSer285Pro (S285P)(heterozygote, paternal); hereafter referred to as $VPS41^{S285P}$] and a nonsense variant in the Clathrin Heavy Chain Repeat (CHCR) resulting in a truncated protein [c.1984C > T,NP_055211.2:pArg662Stop(R662*)(heterozygote, maternal); hereafter referred to as $VPS41^{R662*}$] (Fig EV1C). The $VPS41^{S285P}$ variant has mixed in silico predictors and is observed in 2/282128 (0.0007%) alleles in gnomAD. The $VPS41^{R662*}$ variant is predicted to cause loss of normal protein function either through protein truncation or nonsense-mediated mRNA decay. It is observed in 98/282180 (0.03%) alleles in gnomAD, and no individuals are reported to be homozygous. The gnomAD frequencies and predictions of pathogenicity by available software programs are shown in Table 1.

Compound heterozygous missense variants in the ITGB4 gene were also identified; however, ITGB4 is associated with autosomal recessive non-Herlitz junctional epidermolysis bullosa which does not fit the phenotype observed in this family.

A description of the whole exome sequencing technique used, the filtering strategy, variant analysis and confirmation by Sanger sequencing, as well as a list of rare variants detected, can be found in the Supplements (Appendix Supplementary Methods and Dataset EV1, respectively).

Family 2 (Fig EV1D)—Patient 3, a boy, born to a distantly related parents of Jewish descent (common ancestor in 18th century, Fig EV1D). The pregnancy was uneventful and delivery was at 40wk 1d by Cesarean section for decelerations and meconium-stained amniotic fluid. His birth weight was 3290 grams (10th–50th centile). On examination, he was noted to have a head circumference at the 3rd centile, retrognathia, short and upturned nose, relatively large ears, clinodactyly of both 5th fingers and hepatosplenomegaly. He was noted to be hypotonic and jittery. He had recurrent infections during infancy and childhood, photophobia, mild sensorineural hearing deficit which required hearing aids, tracheomalacia, and a severe global delay. He developed hypertension and a progressive spasticity of the lower limbs from the age of 5 years onwards. His facial features coarsened over time with heavy eyebrows, gingival hypertrophy, protruding tongue, thick lips, and thick ear lobes (Fig EV1E I–III). He had a pectus carinatum and short stubby hands. The liver and spleen

sizes gradually normalized, and by age 5, there was no longer hepatosplenomegaly. At that age, the boy had severe global developmental delay, a convergent strabismus with restricted vertical eye movements, axial hypotonia with peripheral spasticity, brisk DTRs at the upper limbs, and weak DTRs on the lower extremities. He had short Achilles tendons with equinus feet position. There was a dysmetry when reaching for objects, but no tremor. In the following years, the boy developed a scoliosis and severe spastic tetraplegia with contractures, first in the lower extremities and later in the upper extremities (starting in the hands). His skin felt soft and thick, and there was hirsutism on his back and legs. He never developed the ability to sit, walk, or speak. He was able to move in a walking device, but later lost that ability. His ability to cough decreased from the age of 12 years, and he developed swallowing problems, resulting in more upper airway infections. His height and head circumference remained at −2.5 SD during the years (Fig EV1B).

Ophthalmologic investigation was normal apart from retinal hypopigmentation. An ECG and cardiac ultrasound were normal. Extensive metabolic investigation including analysis for peroxisomal and lysosomal disorders, including urinary mucopolysaccharides and oligosaccharides as well as CDG (sialo transferrines), showed no abnormalities. A spinal X-ray showed a kyphosis at C3 (Fig EV1F IV) and X-rays of the hands showed hypoplastic distal phalanges on digits 2, 3, and 5 and a short metacarpal 1, bilaterally (Fig EV1F V). A cerebral MRI showed agenesis of the corpus callosum (Fig 1C), bilateral colpocephaly, a dysplastic cerebellum with polymicrogyria of the upper part, and a dysplastic pons.

Microarray analysis (Agilent 180 K custom HD-DGH microarray; (AMADID-nr. 27730)) revealed no chromosomal imbalances, targeted genetic analysis for several disease genes, and a gene panel including approximately 800 genes associated with intellectual disability, showed no detectable variants. Trio whole exome sequencing showed compound heterozygous variants in the *VPS41* gene; a paternal variant in the TPR-like domain [NM_014396.3: c.1423-2A > G p.?; hereafter referred to as $VPS41^{c.1423-2A>G}$] and a maternal nonsense variant in the CHCR domain [c.1984C > T, NP_055211.2:pArg662Stop(R662*)(heterozygote, maternal); hereafter referred to as $VPS41^{R662*}$]. The c.1423-2A > G variant is a splice site variant that destroys the canonical splice acceptor site in intron 17 and is predicted to cause abnormal gene splicing. The $VPS41^{R662*}$ nonsense variant is shared between all three patients and heterozygously present in the healthy brother of patient 3 (Fig EV1C). Patient 3 also carried a homozygous missense variant of unknown significance in *UPF3A* [NM_080687.2:c.707G > A, NP_542418.1:pArg236Gln] for which both parents and his healthy sibling were heterozygotes. This variant is located in an evolutionary conserved region, but not within a known functional domain of the protein. At this point, variants in this gene have not been associated with any human disease.

A description of the whole exome sequencing technique used, the filtering strategy, variant analysis, and confirmation by Sanger sequencing, as well as a list of rare variants detected, can be found in the Supplements (Appendix Supplementary Methods and Dataset EV2, respectively). The gnomAD frequencies and prediction of pathogenicity by available software programs are shown in Table 1.

# Results

### Patient fibroblasts and *VPS41^{KO}* cells contain small-sized, acidified lysosomes active for cathepsin B

To study the cellular effects of *VPS41* variants in patient cells, we obtained skin biopsies from the youngest brother of the first family (patient 2; *VPS41^{S285P/R662*}*), patient 3 (*VPS41*^{c.1423-2A>G/R662*}), an independent control (*VPS41^{WT/WT}*), and from the father and mother of patients 1 and 2 (*VPS41^{WT/S285P}* and *VPS41^{WT/R662*}*) and made

primary fibroblast cultures. In addition, we mimicked disease conditions in HeLa cells using CRISPR/Cas9 methodology. We chose to study fibroblasts from patient 2, because patient 1 also displayed the *de novo* duplication at Chr1q21.1, which partially overlaps with the 1q21.1 microduplication syndrome (Brunetti-pierri *et al*, 2008). The clinical features of patient 1 are more severe than in individuals suffering from 1q21.1 duplication syndrome; however, we cannot rule out that patient fibroblasts are affected by this duplication. Similarly, patient 3 bears a homozygous mutation in UPF3A of unknown pathogenicity, which possibly could induce alterations in fibroblasts.

**A**  Patient 1

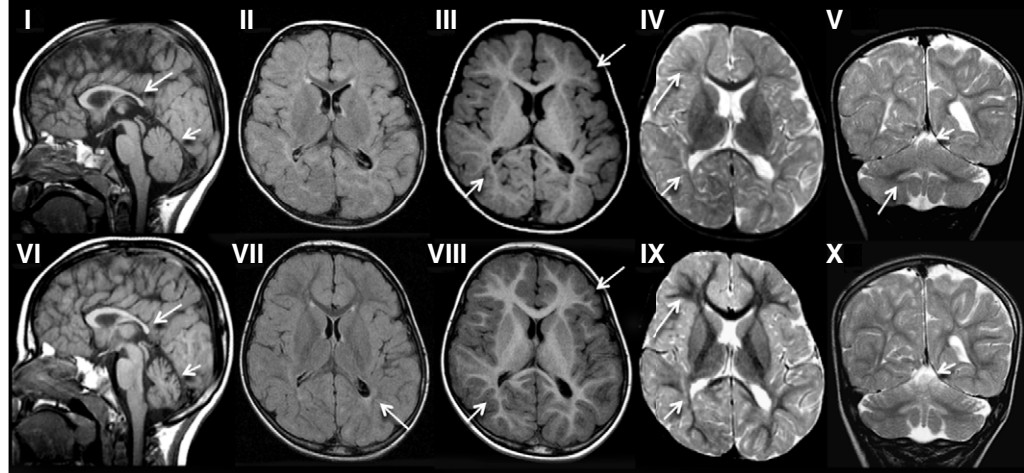

**B**  Patient 2

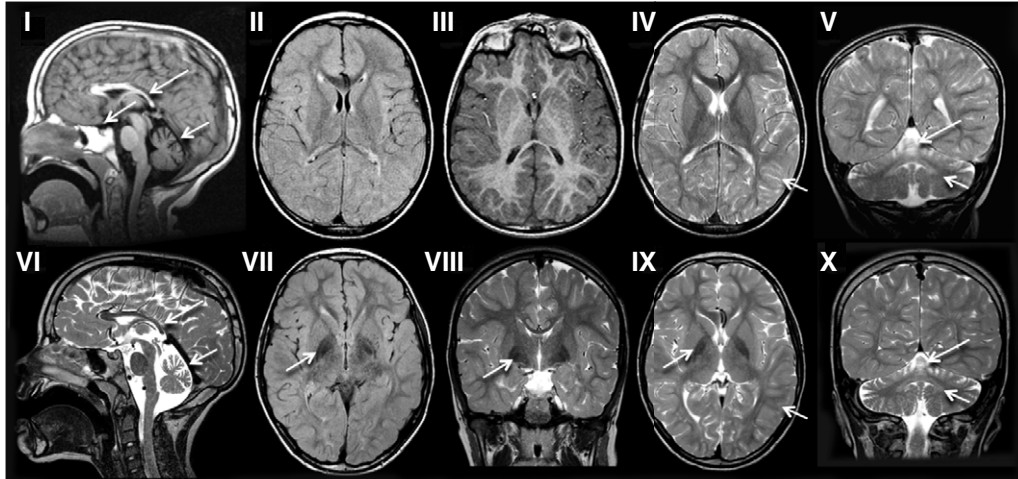

**C**  Patient 3

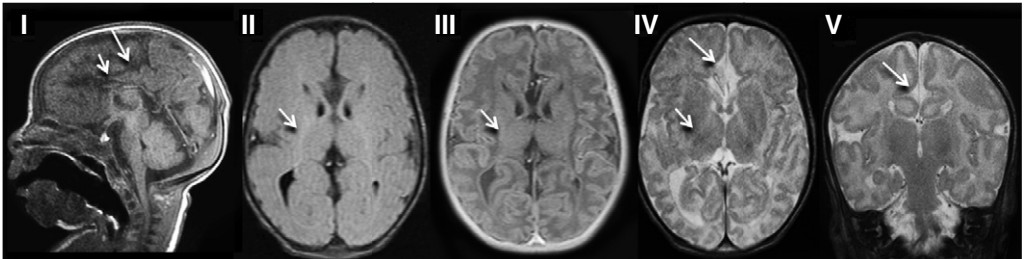

**Figure 1.**

◀ Figure 1.  **Mutations in *VPS41* cause a neurodegenerative disease.**

A   Patient 1 (older sibling) underwent 3 MRI studies, first and last are shown. Top row: 21 months; Bottom row: 4 years 7 months. The corpus callosum is thin on T1-weighted sagittal images (I, V, VI) with a saber shape (long arrows). The shape remains consistent over time. The vermis is normal in configuration and size initially (I), but demonstrates volume loss on follow-up (V, VI) examination (short arrows). FLAIR axial image (II) is age appropriate, while (VII) demonstrates periatrial increased signal (arrow). There is delay in myelin maturation present on T1 (III, VIII)- and T2 (IV, IX)-weighted axial images (long arrows) on initial examination, with further slow myelin development over time. There is deficiency of periatrial white matter volume on both T1 (III, VIII)- and T2 (IV, IX)-weighted axial images (short arrows). Coronal T2 image (V) demonstrates abnormally increased signal of the dentate nuclei (arrow), unchanged on follow-up image (X). Superior vermian atrophy (short arrows) over time.

B   Patient 2 (younger sibling). Top row: 4 years, 11 months; Bottom row: 9 years, 6 months. The corpus callosum is thin with a saber-shape (long arrows) configuration on T1-weighted sagittal (I) and T2-weighted sagittal (V, VI). The shape remains consistent over time, although the volume decreases slightly. The vermis demonstrates volume loss on both sagittal examinations (short arrows). There is iron deposition in the basal ganglia on FLAIR (VII) and T2-weighted coronal (VIII) and axial (IX) images at 9 years 8 months of age (arrow). This was not present on earlier imaging. Iron deposition was also present in the subthalamic nuclei (not shown). Myelin maturation is age appropriate on the initial T1-weighted axial image (III) and minimally delayed on T2-weighted image (IV) at 4 years 11 months of age (short arrow). Myelin is age appropriate (short arrow) on follow-up T2-weighted images (VIII, IX) at both ages. The dentate nuclei (short arrows) are bright on T2 coronal images (V, VI, X). There is progressive cerebellar hemisphere volume loss.

C   Patient 3, 3 weeks old. Sagittal T1-weighted image (I) demonstrates a very thin severely hypoplastic remnant of corpus callosum (short arrow), a short cingulate sulcus (long arrow), and slender pons. FLAIR image (II) reveals faintly increased signal (short arrow) in the posterior limb of internal capsule (PLIC). There is lack of myelin maturation in the PLIC (short arrows) on T1-IR (III)- and T2 (IV)-weighted axial images. Focal widening of the interhemispheric CSF space (arrow) on T2-weighted image (IV) reflects the absence of rostral fiber tracts. Cingulate gyrus is present (long arrow). No traversing callosal fibers are shown at this level (I).

Table 1.  **Prediction of pathogenicity of *VPS41* variants.**

| *VPS41* (NM_014396.3) | gnomAD global | SIFT | CADD | PolyPhen2 HDIV | PolyPhen2 HVAR | Patient |
|---|---|---|---|---|---|---|
| c.853 T > C p.S285P | 0.0000071; 2/282128 | 0.048 damaging | 23.5 | 0.956 possibly damaging | 0.361 benign | 1 and 2 |
| c.1984 C > T p.R662* | 0.00035; 98/282180 | (b) | 40 | (b) | (b) | 1, 2 and 3 |
| c.1423-2A > G p.? (a) | 0.000032; 1/31398 | (b) | 34 | (b) | (b) | 3 |

Frequency of the specific *VPS41* variants in the general population based on the gnomAD database ([variants observed]/[total of individuals studied]) (http://gnomad.broadinstitute.org/) and prediction of pathogenicity based on the programs SIFT, (https://sift.bii.a-star.edu.sg/), CADD (https://cadd.gs.washington.edu/), and PolyPhen (http://genetics.bwh.harvard.edu/pph2/).
(a): Three different splice site analysis programs (MaxEnt, NNSPLICE, and SSF) predict a total loss of wild-type acceptor splice site (Interactive Biosoftware - Created by Alamut Visual v.2.15.0).
(b): Not determined; SIFT and PolyPhen2 programs can only be used for analysis of missense mutations.

First, we studied the effects of the *VPS41* mutations on lysosomal morphology in the $VPS41^{S285P/R662*}$ fibroblasts. Electron microscopy (EM) showed a high variation in appearance of endolysosomal compartments (Fig EV2A–D), but no inclusion bodies. This indicates that mutations in *VPS41* do not cause a classical lysosomal storage phenotype (Papa *et al*, 2010)ʼ which is in agreement with the metabolic studies performed on patient fibroblasts, leucocytes, and urine (see patient descriptions). To study lysosomal acidification and hydrolase activity, we performed fluorescence microscopy using MagicRed cathepsin B. This showed an increase in the number of cathepsin B-active puncta as compared to control or parental cells (Appendix Fig S1A, quantified in S1A'). Concomitantly, immunofluorescence of the lysosomal membrane protein LAMP-1 showed a significant increase in both patients (Fig 2A, quantified in 2A'). Similarly, HeLa$^{VPS41KO}$ cells, i.e., HeLa cells knockout for *VPS41* using CRISPR/Cas9 methodology (Fig 2B), showed numerous small LAMP-1 puncta (Fig 2C, quantified in 2C') and an increase in acidic compartments, investigated using Lysotracker-Green (Appendix Fig S1B, quantified in S1B').

LAMP-1 resides in lysosomes as well as late endosomes, which by EM can be distinguished by the presence or absence of degraded material, respectively (Peden *et al*, 2004; Huotari & Helenius, 2011; Pols *et al*, 2013a; Klumperman & Raposo, 2014). By immuno-EM, we found that $VPS41^{S285P/R662*}$ fibroblasts contain LAMP-1-positive late endosomes and lysosomes (Fig 2D), which were also positively labeled for cathepsin B (Appendix Fig S1C). However, in agreement with the fluorescent microscopy observations, morphologically identified, LAMP-1-positive lysosomes in patient-derived cells were significantly smaller than in $VPS41^{WT/WT}$, $VPS41^{WT/S285P}$ or $VPS41^{WT/R662*}$ fibroblasts (Fig 2E). The relative labeling density (an indication for protein concentration per membrane unit) of LAMP-1 in lysosomes remained equivalent to control cells (Fig 2F), but since lysosomes were smaller, the total number of gold particles (indication for total amount of LAMP-1) had decreased (Fig 2G). Similar data were obtained for LAMP-2 (Appendix Fig S2A and B).

These data show that lysosomes in patient cells contain LAMP-1, LAMP-2, and cathepsin B, but are considerably smaller in size than in control cells. We conclude that biallelic expression of $VPS41^{R662*}$ with $VPS41^{S285P}$ or $VPS41^{c.1423-2A>G}$, as well as the absence of *VPS41* by gene knockout, induces an increase in small-sized, enzymatically active lysosomes, which at steady state conditions contain normal LAMP levels.

## VPS41$^{R662*}$ is not detectable in fibroblasts

To establish expression levels of the VPS41 variants, we performed quantitative Western blots of patients and parental fibroblasts. Wild-type VPS41 was readily detectable in $VPS41^{WT/R662*}$ fibroblasts

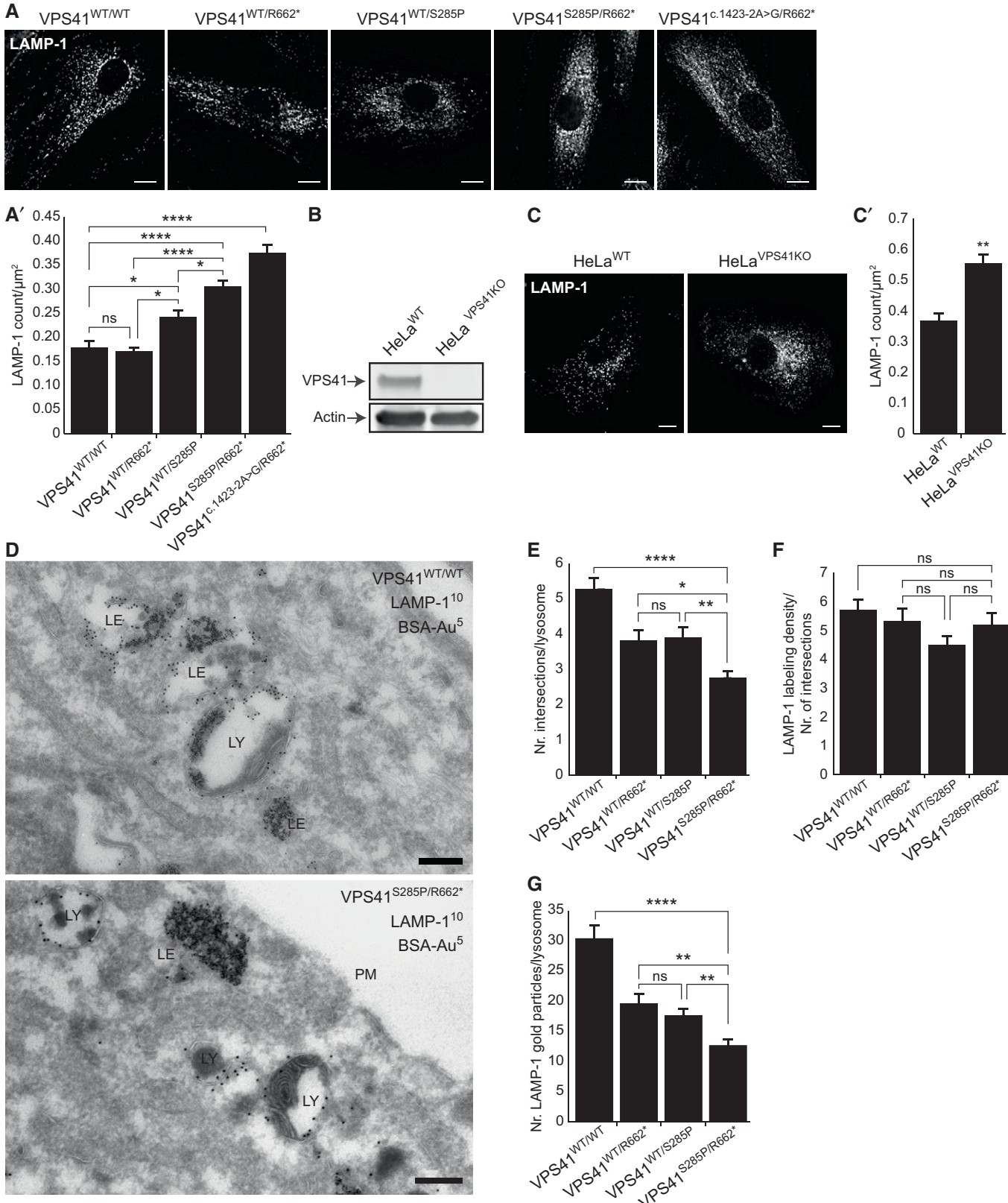

Figure 2.

◄

**Figure 2. VPS41 patient and HeLa VPS41 knockout cells contain more but smaller lysosomes.**

A   Immunofluorescence microscopy of control, parental, and patient fibroblasts labeled for LAMP-1. Patient fibroblasts ($VPS41^{S285P/R662*}$ and $VPS41^{c.1423-2A>G/R662*}$) show more LAMP-1 puncta, which are distributed throughout the cell. (A') Quantification shows a significant increase in the number of LAMP-1-positive compartments for both patients. > 15 Cells per condition were quantified ($n = 3$). Scale bars, 10 μm.

B   HeLa VPS41 knockout (HeLa$^{VPS41KO}$) cells made using CRISPR/Cas9 methodology. Western blot analysis confirms a full knockout. The same HeLa$^{WT}$ and HeLa$^{VPS41KO}$ samples were analyzed in Appendix Fig S8C, showing the same actin control.

C   LAMP-1 immunofluorescence of HeLa$^{WT}$ and HeLa$^{VPS41KO}$ cells. Similar to patient-derived fibroblasts, more LAMP-1-positive compartments are seen in HeLa$^{VPS41KO}$ cells (quantified in C'). > 10 Cells per cell line per experiment were quantified ($n = 3$). Scale bars, 10 μm.

D   Immuno-electron microscopy of $VPS41^{WT/WT}$ and $VPS41^{S285P/R662*}$ fibroblasts incubated for 2 h with BSA conjugated to 5 nm gold (BSA-Au$^5$) and labeled for LAMP-1 (10 nm gold particles). Lysosomes are recognized by the presence of degraded, electron-dense material. LE = late endosome, LY = lysosome, PM = plasma membrane. Scale bar, 200 nm.

E   Morphometrical analysis showing that lysosomes in patient fibroblasts are significantly smaller than in control and parental cells. > 100 Randomly selected lysosomes per condition were quantified.

F   Relative labeling density of LAMP-1. Number of LAMP-1 gold particles per lysosome was divided by the number of grid intersections, representing lysosomal size. No significant difference was found between $VPS41^{WT/WT}$, $VPS41^{WT/S285P}$, $VPS41^{WT/R662*}$ or $VPS41^{S285P/R662*}$ fibroblasts. > 53 Lysosomes per condition were quantified.

G   Quantitation of LAMP-1 gold particles per lysosome. $VPS41^{S285P/R662*}$ lysosomes have significantly less LAMP-1 then control and parental cells. > 53 Lysosomes per condition were quantified. Similar results were obtained for LAMP-2 (Appendix Fig S2).

Data information: Data are represented as mean ± SEM. *$P < 0.05$, **$P < 0.01$, ****$P < 10^{-5}$. One-way ANOVA with Tukey's correction (A', E, F and G) or Unpaired t-test (C'). Exact P-values are reported in Appendix Table S3.
Source data are available online for this figure.

from the mother of patients 1 and 2 (Fig 3A). Also in the paternal $VPS41^{WT/S285P}$ fibroblasts there was a strong signal for full-length VPS41, in this case representing both VPS41$^{WT}$ and VPS41$^{S285P}$ (Fig 3A). VPS41$^{R662*}$, the variant shared between all 3 patients, encodes for a premature stop codon predicted to result in a truncated protein of lower molecular weight. However, we could not detect a lower VPS41$^{R662*}$ band in either $VPS41^{S285P/R662*}$ patient or $VPS41^{WT/R662*}$ maternal fibroblasts (Fig 3A, Appendix Fig S3A). Treatment with proteasome inhibitor MG132, to prevent degradation of VPS41$^{R662*}$, did not change this outcome. Since the VPS41$^{R662*}$ variant is recognized by the VPS41 antibody, these data indicate that VPS41$^{R662*}$ protein levels in fibroblasts are below detection. RNA bearing a premature stop codon can be prematurely degraded via Nonsense-Mediated Decay (NMD). To determine whether this is the case for $VPS41^{R662*}$, we performed RT–PCR on $VPS41^{WT/WT}$, $VPS41^{WT/R662*}$, $VPS41^{WT/S285P}$, and $VPS41^{S285P/R662*}$ fibroblasts. Indeed, maternal and patient fibroblasts showed a significant decrease in VPS41 mRNA levels (Appendix Fig S3B). Together, these data show that $VPS41^{R662*}$ mRNA levels are below detection due to premature degradation of the mutant mRNA. In contrast to VPS41$^{R662*}$, the VPS41$^{S285P}$ variant was readily detectable in Western blots of $VPS41^{S285P/R662*}$ fibroblasts (Fig 3A). Since this is the only allele that is expressed, the total VPS41 expression was lower than in control cells (Fig 3'). Finally, our Western blot analysis revealed a striking reduction, to circa 10% of wild-type levels, of VPS41 protein levels in $VPS41^{c.1423-2A>G/R662*}$ fibroblasts (Fig 3A, quantified in 3A' and Appendix Fig S3A). Since the VPS41$^{R662*}$ variant is not expressed, these data indicate that the $VPS41^{c.1423-2A>G}$ variant is expressed at a very low level.

We conclude from these data that fibroblasts of patients 1 and 2 only contain substantial levels of VPS41$^{S285P}$, whereas cells of patient 3 only contain circa 10% of the VPS41$^{c.1423-2A>G}$ variant.

## VPS41$^{S285P}$ interacts with other HOPS subunits, Arl8b and Rab7

In control conditions, VPS41 interacts with other HOPS subunits and the small GTPases Rab7 and Arl8b (Lin et al, 2014; Khatter

et al, 2015). To study whether VPS41 mutations affect HOPS assembly, we performed co-immunoprecipitation (co-IP) studies. We focused on VPS41$^{S285P}$, being the only variant significantly expressed in patient cells (Fig 3A), and included VPS41$^{R662*}$ as negative control, since VPS41–HOPS interaction requires the presence of the C-terminal RING domain absent in VPS41$^{R662*}$ (Fig EV1C; Hunter et al, 2017). We performed the co-IP studies in HeLa cells expressing GFP-tagged constructs of VPS41$^{WT}$, VPS41$^{S285P}$, or VPS41$^{R662*}$ and, respectively, FLAG- and HA-tagged constructs of the HOPS core components VPS18 or VPS33A. As expected, the truncated VPS41$^{R662*}$ mutant clearly showed reduced interactions with the other HOPS subunits (Fig 3B and Appendix Fig S4A). By contrast, the point mutation VPS41$^{S285P}$ did not affect interaction with VPS18 and VPS33A (Fig 3B and Appendix Fig S4A). Additionally, we examined in HeLa$^{VPS41KO}$ cells the interaction between endogenous VPS18 and GFP-tagged VPS41 constructs. We found that GFP-VPS41$^{S285P}$ and GFP-VPS41$^{WT}$ bind endogenous VPS18 with similar affinities, whereas no interaction between endogenous VPS18 and VPS41$^{R662*}$ was found (Appendix Fig S4B). These data indicate that VPS41$^{S285P}$ can correctly bind to the HOPS complex, whereas VPS41$^{R662*}$ lacks this ability. Hence, if VPS41$^{R662*}$ would be expressed at sufficient levels, it would not be incorporated in the HOPS complex.

Next, we performed co-IPs using transiently transfected HeLa$^{WT}$ cells with APEX-V5-tagged VPS41 constructs and FLAG and GFP-tagged constructs of Rab7 and Arl8b, respectively. This showed that both VPS41$^{S285P}$ and VPS41$^{R662*}$ bind Rab7 and Arl8b to the same extent as VPS41$^{WT}$ (Fig 3C). The subcellular distribution of the distinct VPS41 constructs was explored by immunofluorescence microscopy. VPS41$^{WT}$ was present in the cytoplasm as well as in distinct fluorescent puncta that colocalized with LAMP-1, consistent with previous studies (Khatter et al, 2015; Jia et al, 2017; Fig 3D). Remarkably, both VPS41$^{S285P}$ and VPS41$^{R662*}$ showed a similar distribution as VPS41$^{WT}$, despite the lack of the C terminus in VPS41$^{R662*}$. Since VPS41$^{R662*}$ binds Rab7 and Arl8b (Fig 3C), these interactions could mediate recruitment of VPS41$^{R662*}$ to late endosomes and lysosomes independent of its C-terminus or the HOPS complex.

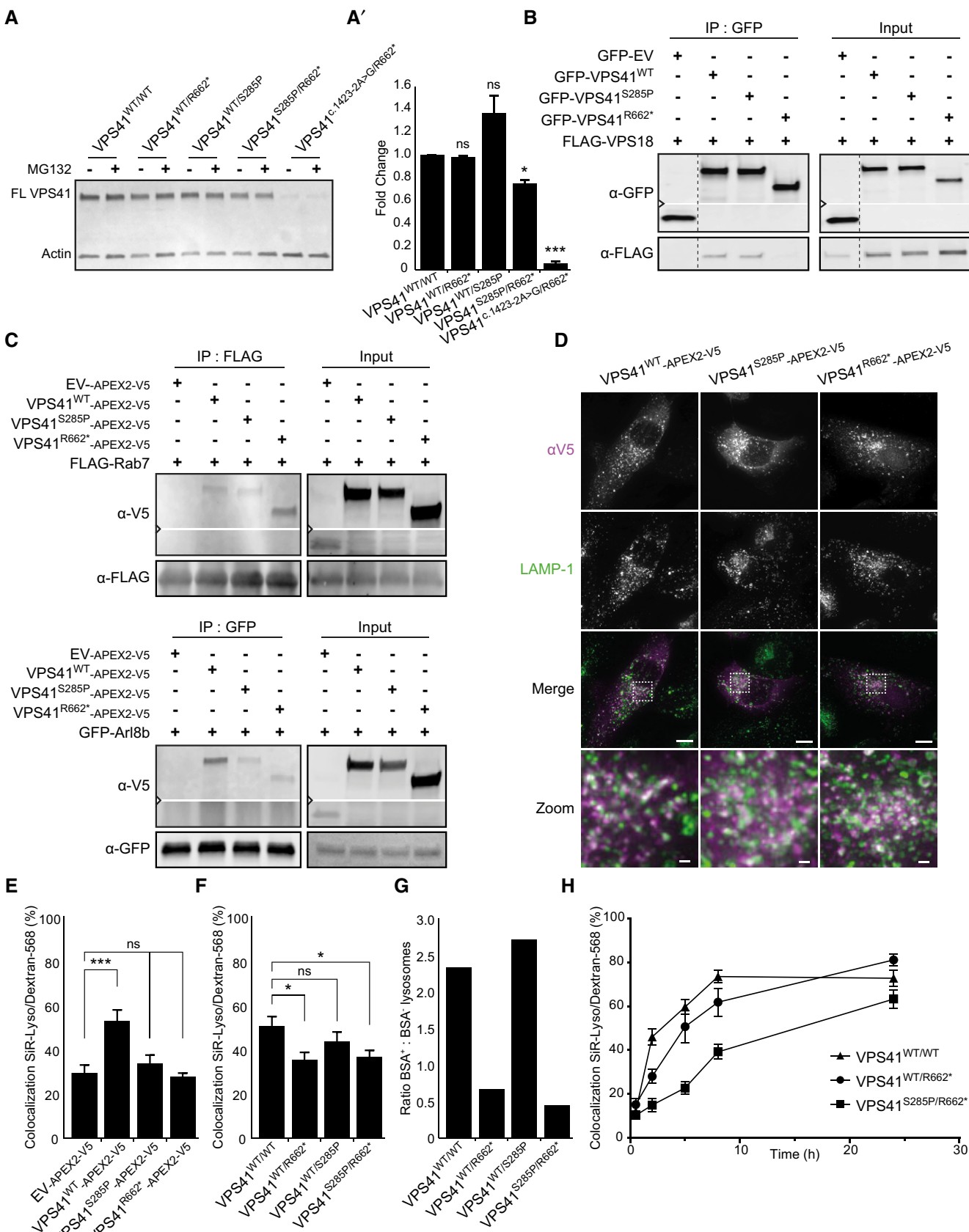

**Figure 3.**

**Figure 3. VPS41 variants delay HOPS-dependent late endosome–lysosome fusion.**

A   Western blot of primary fibroblasts derived from patient 2 ($VPS41^{S285P/R662*}$), his mother ($VPS41^{WT/R662*}$), father ($VPS41^{WT/S285P}$), patient 3 ($VPS41^{c.1423-2A>G/R662*}$) and an unrelated, healthy control ($VPS41^{WT/WT}$) (Western blot of longer exposure time shown in Appendix Fig S3A). Full-length VPS41 (FL VPS41), representing $VPS41^{WT}$ and $VPS41^{S285P}$, is observed in all cells. Truncated $VPS41^{R662*}$ in patients and maternal fibroblasts is not detectable. $VPS41^{R662*}$ is also not visible in fibroblasts treated with 50 µM MG132 (4 h) to inhibit proteasomal degradation, indicating that the mRNA encoding for this mutant is degraded. (A') Quantification of VPS41 protein levels show a reduction in both patients compared with $VPS41^{WT/WT}$, with only 10% remaining VPS41 levels in $VPS41^{c.1423-2A>G/R662*}$ ($n = 2$).

B   Immunoprecipitation (IP) on HeLa cells co-expressing GFP-empty vector (EV), GFP-$VPS41^{WT}$, GFP-$VPS41^{S285P}$, or GFP-$VPS41^{R662*}$ and FLAG-VPS18. GFP-$VPS41^{WT}$ and GFP-$VPS41^{S285P}$ both interact with FLAG-VPS18 and as shown in Appendix Fig S4A with HA-VPS33A. Interaction between GFP-$VPS41^{R662*}$ and FLAG-VPS18 is strongly reduced. Note that the stop codon in $VPS41^{R662*}$ leads to a truncated protein of lower molecular weight ($n = 3$).

C   IP on Hela cells co-expressing $VPS41^{WT}$-APEX2-V5, $VPS41^{S285P}$-APEX2-V5, or $VPS41^{R662*}$-APEX2-V5 and FLAG-RAB7 or GFP-Arl8b. The IP was performed on FLAG or GFP, respectively. All VPS41 variants interact with Rab7 and Arl8b ($n = 3$).

D   HeLa$^{VPS41KO}$ cells transfected with $VPS41^{WT}$-APEX2-V5, $VPS41^{S285P}$-APEX2-V5, or $VPS41^{R662*}$-APEX2-V5 constructs labeled for LAMP-1 and V5 immunofluorescence microscopy. All VPS41 variants colocalize with LAMP-1, indicating that $VPS41^{S285P}$ and $VPS41^{R662*}$ are recruited to late endosomes/lysosomes. Scale bars 10 µm; zoom of squared area, 1 µm.

E   Quantification of endocytosis-rescue experiments in HeLa cells. HeLa$^{VPS41KO}$ cells transfected with $VPS41^{WT}$-APEX2-V5 show a significant increase in colocalization of endocytosed Dextran-568 and SiR-Lysosome cathepsin D (SiR-Lyso), indicating rescue of the endocytosis phenotype. Neither VPS41 variant rescues this HOPS complex functionality. >13 Cells per cell line per experiment were quantified ($n = 3$).

F   Quantification of lysosomal delivery of endocytosed cargo in fibroblasts, based on fluorescent data. $VPS41^{WT/WT}$, $VPS41^{WT/S285P}$, $VPS41^{WT/R662*}$, and $VPS41^{S285P/R662*}$ primary fibroblasts were incubated with Dextran-568 and SiR-Lysosome cathepsin D (SiR-Lyso) for 2 and 3 h, respectively. Colocalization representing delivery of Dextran to enzymatically active lysosomes is reduced in $VPS41^{WT/R662*}$ and $VPS41^{S285P/R662*}$ cells. >11 Cells per cell line per experiment were quantified ($n = 3$).

G   Quantification of lysosomal delivery of endocytosed cargo in fibroblasts, based on EM data. $VPS41^{WT/WT}$, $VPS41^{WT/S285P}$, $VPS41^{WT/R662*}$, and $VPS41^{S285P/R662*}$ fibroblasts were incubated with BSA-Au$^5$ for 2 h and labeled for LAMP-1 (10 nm gold particles) immuno-EM (Fig 2D). LAMP-1-positive lysosomes were scored for presence of BSA-Au$^5$, and the ratio between BSA$^+$ and BSA$^-$ lysosomes was calculated. >46 Lysosomes per condition were quantified. Both $VPS41^{WT/R662*}$ and $VPS41^{S285P/R662*}$ show a strong decrease in BSA-positive lysosomes indicating a fusion defect between late endosomes and lysosomes.

H   $VPS41^{WT/WT}$, $VPS41^{WT/R662*}$, and $VPS41^{S285P/R662*}$ primary fibroblasts incubated with Dextran-568 for 0.5, 2, 5, 8, and 24 h. Colocalization of Dextran-568 and SiR-Lysosome cathepsin D (SiR-Lyso) reveals a delay in lysosomal delivery in maternal ($VPS41^{WT/R662*}$) and patient ($VPS41^{S285P/R662*}$) cells at 2 h of Dextran uptake. After 5 h, maternal cells show similar to control colocalization levels. Patient cells show only after 24 h colocalization levels similar to control, indicating a delay rather than a block in late endosome–lysosome fusion. > 10 Cells per cell line were quantified.

Data information: Data are represented as mean ± SEM. *$P < 0.05$, ***$P < 10^{-4}$. Unpaired $t$-test (A'), one-way ANOVA with Bonferroni correction (E) or one-way ANOVA with Tukey's correction (F). Exact $P$-values are reported in Appendix Table S3.

Source data are available online for this figure.

---

Summarizing, these co-IP experiments show that VPS41$^{S285P}$ has retained the capacity to bind the HOPS components VPS18 and VPS33A, as well as Rab7 and Arl8b. VPS41$^{R662*}$ does not bind other HOPS components, but still interacts with Rab7 and Arl8b. Both VPS41$^{S285P}$ and VPS41$^{R662*}$ are distributed between the cytoplasm and LAMP-1-positive endo-lysosomes.

## VPS41$^{S285P}$ causes a defect in HOPS-dependent late endosome–lysosome fusion

Depletion of VPS41 by RNAi results in a decrease in HOPS-dependent fusion between late endosomes and lysosomes (Swetha *et al*, 2011; Pols *et al*, 2013a). To establish the effect of VPS41$^{S285P}$ on HOPS functionality, we transiently transfected HeLa$^{VPS41KO}$ cells with APEX2-V5 tagged constructs of VPS41$^{WT}$, VPS41$^{S285P}$, or VPS41$^{R662*}$. Cells were incubated for 2 h with the endocytic marker Dextran-Alexa Fluor 568 (Dextran-568) combined with SiR-Lysosome to mark active cathepsin D compartments. Colocalization between these probes indicates the transfer of endocytosed cargo to enzymatically active lysosomes, which is HOPS dependent. VPS41$^{KO}$ cells transfected with Empty Vector (EV) showed low levels of colocalization (Fig 3E and Appendix Fig S5A), which was increased by transfection with VPS41$^{WT}$, indicating a restoration of HOPS function (Fig 3E and Appendix Fig S5A). As expected, transfection with VPS41$^{R662*}$ did not rescue the endocytosis defect, since VPS41$^{R662*}$ fails to bind other HOPS components (Fig 3B and Appendix Fig S4). Surprisingly, however, VPS41$^{S285P}$ also failed to rescue the endocytosis defect (Fig 3E and Appendix Fig S5A). Thus, even though VPS41$^{S285P}$ binds VPS18 and VPS33A (Fig 3B and Appendix Fig S4)

and localizes to endo-lysosomes (Fig 3D), it does not form a functional HOPS complex. A plausible explanation for this defect is misfolding of the VPS41$^{S285P}$ protein due to the Arginine to Proline substitution.

These data show that, despite its ability to bind other HOPS components, VPS41$^{S285P}$ cannot form a functional HOPS complex.

## Patient fibroblasts are compromised in delivery of endocytosed cargo to enzymatically active lysosomes

The data so far indicate that all patient-derived VPS41 variants prevent formation of a functional HOPS complex. To assess the process of late endosome–lysosome fusion in patient cells, we performed the Dextran-568 and SiR-Lysosome cathepsin D colocalization assay in primary fibroblasts of patient 2 and his parents. A similar assay with $VPS41^{c.1423-2A>G/R662}$ cells of patient 3 failed to be successful, since these cells are very vulnerable and died upon incubation with SiR-Lysosome. We found a significant endocytosis defect in $VPS41^{S285P/R662*}$ patient fibroblasts (Fig 3F and Appendix Fig S5B) and, unexpectedly, also in the maternal $VPS41^{WT/R662*}$ cells. By contrast, $VPS41^{WT/S285P}$ (paternal) fibroblasts were not affected in endocytosis (Fig 3F and Appendix Fig S4B).

To pinpoint the block in endocytosis at the EM level, we incubated primary fibroblasts for 2 h with the endocytic marker BSA conjugated to 5 nm gold particles (BSA-Au$^5$) and processed cells for immuno-EM. Sections were labeled for LAMP-1 or LAMP-2 and randomly screened for LAMP-positive lysosomes (Klumperman & Raposo, 2014) that were scored positive or negative for BSA-Au$^5$

(Fig 2D). In agreement with the data from florescence microscopy, this showed a decrease in the delivery of BSA-Au$^5$ to LAMP-positive lysosomes in $VPS41^{S285P/R662*}$ patient and $VPS41^{WT/R662*}$ maternal fibroblasts (Fig 3G). The paternal $VPS41^{WT/S285P}$ fibroblasts were not affected. Collectively, these fluorescent and EM data indicate that transfer of endocytosed cargo to enzymatically active, LAMP-positive lysosomes is affected in fibroblasts of patient 2 and of his mother.

Since the mother of the patients does not show a clinical pheno-type, we reasoned that the defect in lysosomal delivery could be kinetic rather than a complete block. To test this, we incubated control, maternal, and patient fibroblasts with Dextran-568 for several time points, after which we determined colocalization with SiR-Lysosome (Fig 3H). After 2 h, $VPS41^{WT/R662*}$ as well as $VPS41^{S285P/R662*}$ fibroblasts again displayed significant lower levels of colocalization than $VPS41^{WT/WT}$ cells. After 5 h, however, the difference between maternal and control fibroblasts was abolished, whereas patient fibroblasts still showed a significant lower level of Dextran in lysosomes. After 24 h of Dextran-568 uptake, patient fibroblasts showed a similar level of colocalization with SiR-Lyso-some as control and maternal cells (Fig 3H).

Together, these data show that expression of $VPS41^{S285P}$ or absence of $VPS41$ causes a deficiency in transfer of endocytic cargo to lysosomes, representing a defect in HOPS-dependent endolyso-mal fusion. This defect is a delay rather than a block in fusion, which in patient cells is more severe than in maternal fibroblasts, indicating that transport kinetics is an important determinator of pathogenesis. The data also show that in maternal fibroblasts the consequence of carrying the $VPS41^{R662*}$ variant is not fully compen-sated for by the $VPS41^{WT}$ allele, yet this does not result in a patho-genic phenotype.

## Patient fibroblasts have a defect in autophagic response to starvation

In addition to late endosome–lysosome fusion, the HOPS complex is required for fusion of autophagosomes with lysosomes (Jiang *et al*, 2014; McEwan *et al*, 2015; Nakamura & Yoshimori, 2017). A block in autophagosome-lysosome fusion results in increased numbers of autophagosomes containing lipidated LC3 (LC3II) and hence an increase in LC3II:LC3I ratio. We quantified this on Western blot and found that under nutrient-rich conditions, the LC3II:LC3I ratio is increased in $VPS41^{S285P/R662*}$ (patient 2) fibroblasts compared with control cells, indicating that patient cells have more autophagosomes (Fig 4A, quantified in 4A'). To verify these findings, we performed immunofluorescent labeling of LC3 in patient and control fibroblasts. This confirmed that $VPS41^{S285P/R662*}$ cells contain significantly more LC3-positive compartments than $VPS41^{WT/WT}$ cells (Fig 4B, quanti-fied in 4B'). Starvation induced an increase in numbers of autophagosomes in both control and patient fibroblasts (Fig 4B, quantified in 4B'). Intriguingly, however, whereas in control fibrob-lasts this increase was 13.4-fold, in patient fibroblasts only a 3.8-fold increase was attained. These data indicate that $VPS41$ patient fibrob-lasts sustain a higher basal level of autophagy and are less respon-sive to nutrient starvation than control fibroblasts.

We further studied this phenomenon in HeLa$^{VPS41KO}$ cells. Under nutrient-rich conditions, LC3II levels in HeLa$^{VPS41KO}$ cells were significantly elevated compared to HeLa$^{WT}$ cells (Fig 4C, quantified in 4C'). A similar upregulation was seen in a previous study in

$VPS41$-depleted HeLa cells (Ding *et al*, 2019). Concomitantly, immunofluorescence of HeLa$^{VPS41KO}$ cells revealed an increase in LC3 puncta (Fig 4D, quantified in 4D'). Strikingly, starvation of HeLa$^{VPS41KO}$ cells did not further increase LC3II protein levels or number of LC3 puncta (Fig 4C and D). Rescue of HeLa$^{VPS41KO}$ cells with VPS41$^{WT}$ reduced the number of LC3 puncta and restored the capacity of cells to respond to starvation and restimulation (Fig 4D, quantified in 4D'). By contrast, expression of VPS41$^{S285P}$ or VPS41$^{R662*}$ did not restore the autophagic flux (Fig 4D, quantified in 4D'). Together, these data show that both patient fibroblasts and $VPS41^{KO}$ cells have increased basal autophagy levels. Patient fibrob-lasts respond to starvation, but not to a similar extent as control fibroblasts. On the contrary, HeLa$^{VPS41KO}$ cells are impaired in their responsiveness to starvation.

To follow the autophagic flux from autophagosome to lyso-some, we next transfected HeLa$^{WT}$ and HeLa$^{VPS41KO}$ cells with an LC3$^{GFP/RFP}$ tandem construct. GFP and RFP co-label autophago-somes, whereas the GFP signal is quenched in the acidic environ-ment after autophagosome–lysosome fusion (Kimura *et al*, 2007). Immunofluorescence of HeLa$^{VPS41KO}$ cells showed a significant higher level of GFP/RFP colocalization than HeLa$^{WT}$ cells, indicat-ing that in the absence of VPS41, autophagosome–lysosome fusion is impaired (Fig 4E, quantified in 4E'). To confirm these findings, we incubated HeLa$^{WT}$ and HeLa$^{VPS41KO}$ cells with Bafilomycin A1 (BafA1), which increases the lysosomal pH and prevents degrada-tion of LC3. Indeed, Western blots of BafA1-treated WT cells showed an increase in LC3II protein levels, however, not to a simi-lar level as non-treated, non-starved $VPS41^{KO}$ cells (Fig EV3A, quantified in EV3B and C). Strikingly, starvation of WT cells in the presence of Baf1A resulted in similar LC3II levels as starved $VPS41^{KO}$ cells in the absence of Baf1A (Fig EV3C). Overall, BafA1 treatment of $VPS41^{KO}$ cells had little effect on LC3II protein levels (Fig EV3A, quantified in EV3B). These data indicate that LC3II in $VPS41^{KO}$ cells is less well degraded by lysosomes than in control cells, consistent with the HOPS-dependent role of VPS41 in autophagosome–lysosome fusion.

Together, the data indicate that $VPS41^{S285P/R662*}$ fibroblasts have increased basal autophagic levels and are, to some extent, respon-sive to autophagic stimuli. $VPS41^{KO}$ cells also have high basal autophagy levels, but are virtually insensitive to starvation. The dif-ferences between patient fibroblasts and HeLa$^{VPS41KO}$ cells could be due to differences in autophagic responsiveness in these cell types. Alternatively, the patient cells might have developed epigenetic compensatory mechanisms.

## VPS41 deficiency causes a HOPS-dependent defect on the TFE3 but not S6K1/4EBP1 axis

A complex of v-ATPase/Ragulator/Rag GTPases present at the lyso-somal membrane senses nutrient status in lysosomes and, in nutri-ent-rich conditions, recruits the mTORC1 complex. Upon nutrient deprivation, mTORC1 dissociates from the lysosomal membrane and loses its kinase activity toward p70 ribosomal protein S6 kinase 1 (S6K1), 4E-binding protein 1 (4EBP1), and the transcription factors TFEB and TFE3. Consequently, TFEB and TFE3 translocate to the nucleus where they activate the CLEAR (Coordinated Lysoso-mal Expression and Regulation) network that induces lysosomal biogenesis and autophagy (Palmieri *et al*, 2011). LC3 is a target of

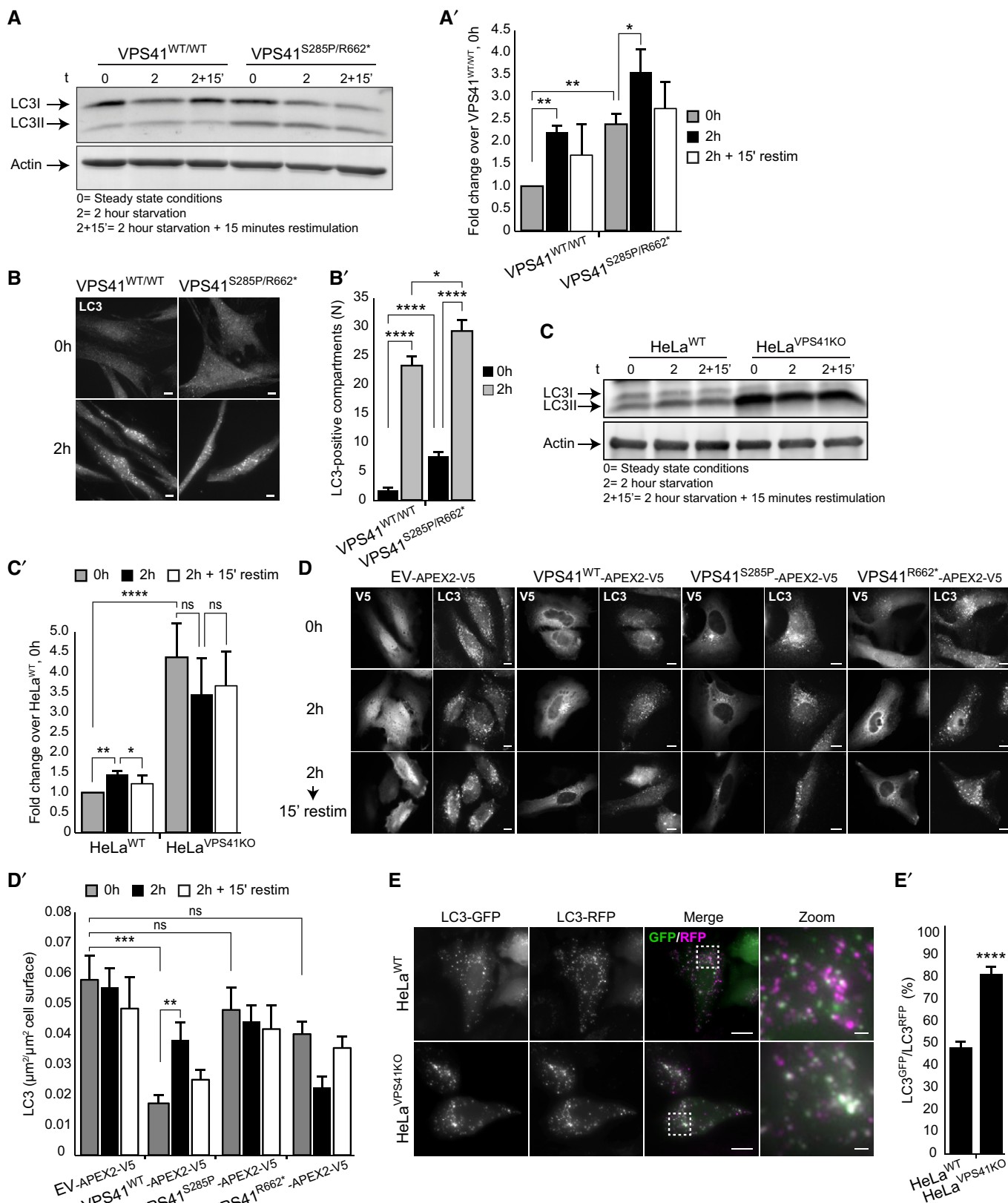

**Figure 4.**

Figure 4. Patient fibroblasts and VPS41$^{KO}$ cells show decreased autophagic flux and response to autophagic stimuli.

A   Western blot of LC3 expression levels in control and patient fibroblasts. In steady state conditions (0 h), patient fibroblasts have a higher ratio of lipidated LC3 (LC3II):
     LC3I than control cells. Induction of autophagy by nutrient starvation (2 h incubation with minimal EBSS medium) results in a raise in LC3II:LC3I ratio in both control
     and patient cells, but in VPS41$^{S285P/R662*}$ fibroblasts this increase is only modest (quantified in A') (n = 2).
B   Immunofluorescence of LC3 in VPS41$^{WT/WT}$ and VPS41$^{S285P/R662*}$ fibroblasts under steady state and starved conditions. At steady state conditions (0 h), patient
     fibroblasts contain more LC3-positive compartments. Nutrient starvation (2 h) increases the number of LC3-positive autophagosomes in control fibroblasts 13.4-fold,
     and in VPS41$^{S285P/R662*}$ fibroblasts only 3.8-fold, indicating a reduced responsiveness to starvation (quantified in B'). > 54 Cells per cell line were quantified. Scale bars,
     10 μm.
C   Western blot analysis of HeLa$^{VPS41KO}$ cells shows a fourfold increase in LC3II protein levels in steady state conditions (0 h) compared with HeLa$^{WT}$ cells. In contrast to
     control cells, nutrient starvation (2 h) did not increase LC3II protein levels in HeLa$^{VPS41KO}$ cells, indicating irresponsiveness to nutrient availability (quantified in C')
     (n = 3).
D   Rescue experiments. HeLa$^{VPS41KO}$ cells transfected with EV-APEX2-V5, VPS41$^{WT}$-APEX2-V5, VPS41$^{S285P}$-APEX2-V5, or VPS41$^{R662*}$-APEX2-V5 and labeled for V5 and LC3 by
     immunofluorescence microscopy. Rescue with VPS41$^{WT}$-APEX2-V5 decreases the number of LC3-positive compartments in steady state conditions (0 h) and restores
     responsiveness to nutrient starvation (2 h) and replenishment (2-hour starvation followed by 15 min restimulation). Neither of the mutant VPS41 variants rescues
     this autophagy phenotype (quantified in D'). >15 Cells per condition were quantified (n = 3). Scale bars, 10 μm.
E   Immunofluorescence of HeLa$^{WT}$ and HeLa$^{VPS41KO}$ cells transfected with LC3$^{GFP/RFP}$ tandem construct. The increased percentage of GFP/RFP-positive compartments in
     HeLa$^{VPS41KO}$ cells indicates a block in autophagic flux (quantified in E'). > 12 Cells per cell line were quantified. Scale bars 10 μm; zoom, 1 μm.

Data information: Data are represented as mean ± SEM. *P < 0.05, **P < 0.01, ***P < 10$^{-4}$, ****P < 10$^{-5}$. Unpaired t-test (A' and E') or one-way ANOVA with Tukey's (B')
or Bonferroni correction (C' and D'). Exact p-values are reported in Appendix Table S3.
Source data are available online for this figure.

the CLEAR network, and mTORC1 inhibition results in enhanced LC3 levels (Palmieri et al, 2011). Since patient-derived fibroblasts and VPS41$^{KO}$ cells show increased LC3 levels under basal conditions (Fig 4A–D), we investigated mTORC1 and TFE3 localization and activity in VPS41 patient fibroblasts and VPS41$^{KO}$ cells.

We first studied recruitment of mTORC1 to lysosomes by immunofluorescence microscopy, monitoring colocalization with LAMP-1. As expected, we found significant overlap between mTORC1 and LAMP-1 puncta in control VPS41$^{WT/WT}$ fibroblasts grown in nutrient-rich conditions. After 2-h starvation, mTORC1 had redistributed to the cytoplasm, and after 15-min restimulation, the colocalization with LAMP-1-positive lysosomes was partially restored (Fig 5A). The parental derived fibroblasts VPS41$^{WT/S285P}$ and VPS41$^{WT/R662*}$ followed this same pattern (Appendix Fig S6). However, in patient-derived VPS41$^{S285P/R662*}$ fibroblasts, mTORC1–LAMP-1 colocalization was strikingly less in all conditions and did not change upon starvation or restimulation. Quantitation of mTORC1/LAMP-1 colocalization, for each cell type relative to the control condition (0 h), clearly showed that VPS41$^{S285P/R662}$ fibroblasts do not alter mTORC1 localization in response to starvation (Fig 5A, quantified in 5A'). To monitor the effect of mTORC1 dissociation on TFEB/TFE3 localization, we labeled fibroblasts for endogenous TFE3 (Fig 5B). In VPS41$^{WT/WT}$, VPS41$^{WT/S285P}$, and VPS41$^{WT/R662*}$ fibroblasts, TFE3 showed a normal localization pattern, and translocated to the nucleus in response to starvation (Fig 5B, quantified in 5B'). However, in VPS41$^{S285P/R662*}$ fibroblasts, TFE3 was constitutively found in the nucleus, regardless of nutrient conditions. A similar constitutive nuclear localization of TFE3 was observed in fibroblasts obtained from patient 3 (VPS41$^{c.1423-2A>G/R662*}$, Fig EV4A). These data show that TFE3 is continuously present in the nucleus of patient cells, regardless of nutrient status.

We then performed similar experiments in HeLa VPS41$^{KO}$ cells. Like patient fibroblasts, these cells showed impaired lysosomal recruitment of mTORC1 (Fig EV4B). Reintroducing VPS41$^{WT}$ into HeLa$^{VPS41KO}$ cells increased the colocalization of mTORC1 with the lysosomal marker cathepsin D from 15 to 30%. Expression of VPS41$^{S285P}$ or VPS41$^{R662*}$ slightly increased lysosomal recruitment of mTORC1, but to a much lesser extent than VPS41$^{WT}$ (Fig EV4C

and D). Concomitantly, HeLa$^{VPS41KO}$ cells showed the "TFE3 translocation to the nucleus phenotype" (Fig 5C, quantified in 5C'), and the same was observed in PC12$^{VPS41KO}$ cells (Fig EV4E, Appendix Fig S7A and B). Expression of VPS41$^{WT}$ in HeLa$^{VPS41KO}$ cells rescued the TFE3 phenotype after starvation and restimulation, whereas expression of VPS41$^{S285P}$ or VPS41$^{R662*}$ had no effect; TFE3 was present in the nucleus in all conditions (Fig 5D, quantified in 5D'). Notably, transfection of cells affects membrane integrity causing lysosomal stress. This likely explains the nuclear localization of TFE3 in HeLa$^{VPS41KO}$ cells transfected with VPS41$^{WT}$ when cultured under steady state conditions (Fig 5D). Similar to TFE3, TFEB is also phosphorylated by mTORC1 to prevent nuclear translocation. Phosphorylation induces a molecular weight shift visible on Western blot. To address whether TFEB phosphorylation is affected in VPS41$^{KO}$ cells, we analyzed this in control conditions and upon starvation. We used the mTORC1 inhibitor Torin-1 as positive control to show the molecular weight shift of TFEB. Indeed, starvation of HeLa$^{WT}$ cells resulted in a similar molecular weight shift as observed after Torin-1 treatment (Fig EV4F). By contrast, HeLa$^{VPS41KO}$ cells did not respond to starvation or Torin-1 treatment, indicating that TFEB phosphorylation is impaired (Fig EV4F). The continuous nuclear localization of TFE3 in patient cells predicts an increase in expression of lysosomal and autophagy proteins (Settembre et al, 2011; Martina et al, 2014). Western blot analysis indeed showed that protein levels of LAMP-1 and the lysosomal hydrolase cathepsin B, both CLEAR targets, are increased in patient fibroblasts (Appendix Fig S8A and B). Similar results were obtained for LAMP-1 and cathepsin D in VPS41$^{KO}$ cells (Appendix Fig S8C). Collectively, these data show that in cells lacking VPS41 or expressing VPS41$^{S285P}$ and/or VPS41$^{R662*}$, regulation of the mTORC1/TFE3 axis is perturbed, resulting in higher levels of autophagy and lysosome proteins and a continuous activation of autophagy, independent of nutrient status.

A possible explanation for the mTORC1/TFE3 phenotype is that the defect in HOPS function results in insufficient delivery of nutrients to lysosomes and subsequent mTORC1 dissociation. If so, any block in HOPS function should give this phenotype. To test this, we made HeLa$^{KO}$ cells for VPS11, VPS18, and VPS39, which are part of HOPS but not required for the ALP/LAMP pathway (Pols et al,

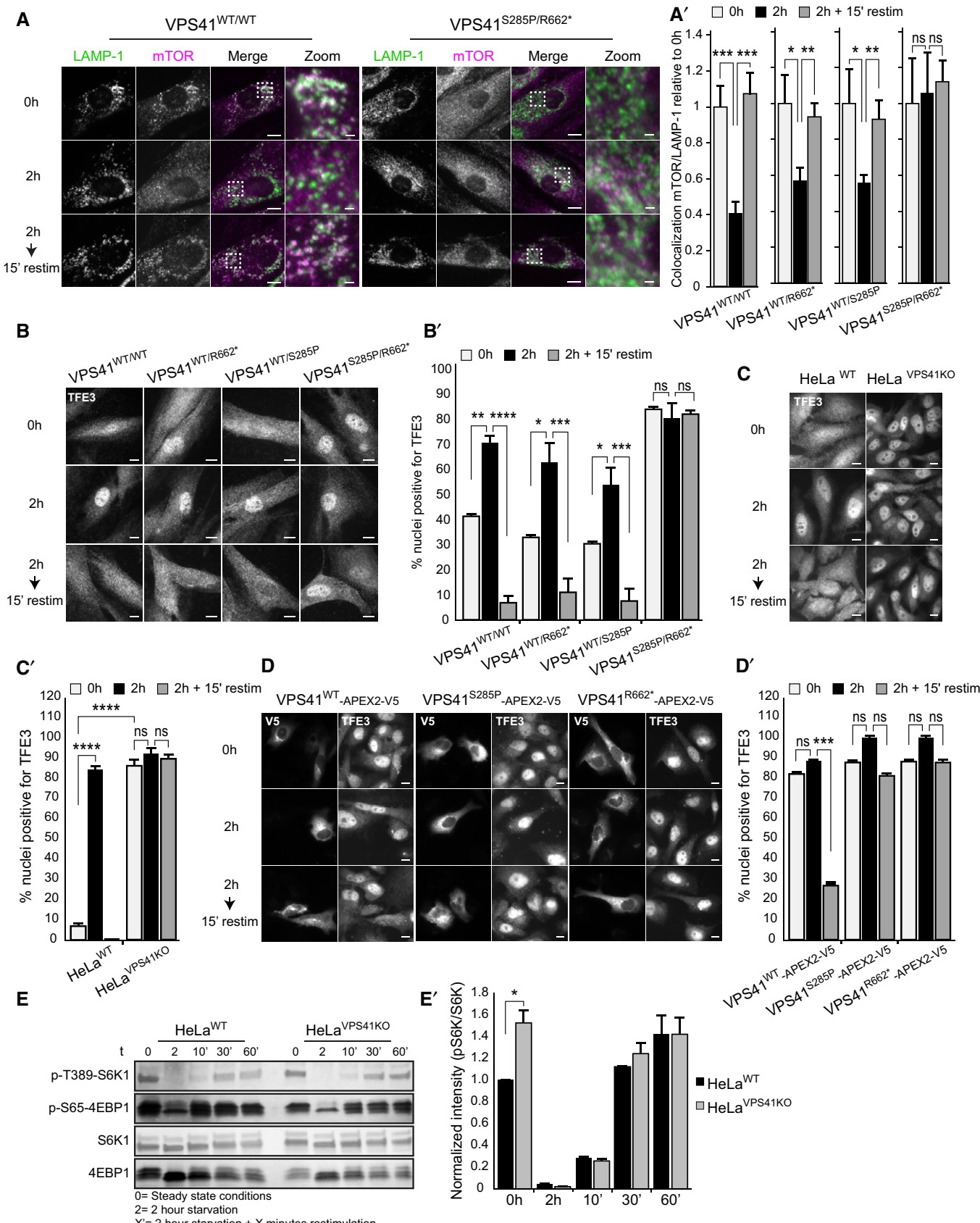

**Figure 5.**

◄

**Figure 5. Patient fibroblasts and *VPS41^KO^* cells exhibit mTORC1 inhibition toward TFE3.**

A Immunofluorescence of control, parental, and patient fibroblasts labeled for LAMP-1 and mTOR. In steady state conditions (0 h), VPS41^S285P/R662*^ fibroblasts show less colocalization between mTOR and LAMP-1. VPS41^WT/WT^, VPS41^WT/S285P^, and VPS41^WT/R662*^ show an appropriate mTOR response upon nutrient deprivation (2 h) or restimulation (2-h starvation followed by 15-min restimulation) (Appendix Fig S5A). > 10 Cells per cell line were quantified (A'). Quantifications are performed relative to colocalization under steady state conditions per cell line (n = 3). Scale bars, 10 μm; zoom, 1 μm.

B Immunofluorescence of VPS41^WT/WT^, VPS41^WT/S285P^, VPS41^WT/R662*^, and VPS41^S285P/R662*^ fibroblasts labeled for TFE3. In VPS41^S285P/R662*^ fibroblasts, TFE3 is constitutively localized in the nucleus regardless of nutrient state (quantified in B'). > 25 Cells per condition were quantified (n = 3). Scale bars, 10 μm.

C TFE3 immunofluorescence in HeLa^WT^ and HeLa^VPS41KO^ cells. In HeLa^VPS41KO^ cells, TFE3 constitutively localizes in the nucleus (quantified in C'). > 83 Cells per condition were quantified (n = 3). Scale bars, 10 μm.

D Rescue experiments of HeLa^VPS41KO^ cells. Expression of VPS41^WT^-APEX2-V5, VPS41^S285P^-APEX2-V5, or VPS41^R662*^-APEX2-V5. Reintroduction of VPS41^WT^ rescues TFE3 localization, whereas expression of mutant VPS41 has no effect (quantified in D'). > 83 Cells per condition were quantified (n = 3). Scale bars, 10 μm.

E Western blot of phosphorylated mTORC1 substrates S6K and 4EBP1 after starvation (2 h) and restimulation (10, 30 or 60 min). HeLa^VPS41KO^ cells show comparable levels of phospho-S6K and phospho-4EBP1, with no difference in recovery after restimulation (quantified in E') (n = 3).

Data information: Data are represented as mean ± SEM. *$P < 0.05$, **$P < 0.01$, ***$P < 10^{-4}$, ****$P < 10^{-5}$. One-way ANOVA with Bonferroni correction (A'–D') or unpaired *t*-test (E'). Exact *P*-values are reported in Appendix Table S3.

Source data are available online for this figure.

2013b; Appendix Fig S9A and B). Similar to *VPS41^KO^* cells, an increase in endolysosomal compartments was observed by immunofluorescent labeling of cathepsin D (Fig EV5A, quantified in EV5A'). *VPS11^KO^* and *VPS18^KO^* cell lines showed increased LC3II protein levels in basal conditions and a constitutive nuclear localization of TFE3, independent of nutrient status (Fig EV5B, quantified in EV5B' and Appendix Fig S9C). These data show that the increase in endolysosomal compartments and the mTORC1/TFE3 phenotype is caused by a dysfunctional HOPS complex. We then studied the effect of *VPS41* or HOPS depletion on the canonical mTORC1 substrates S6K1 and 4EBP1, required for cellular growth. Intriguingly, phosphorylation of these substrates was not affected by the absence of *VPS41* or other HOPS subunits (Fig 5E, quantified in 5E' and Fig EV5C). Likewise, phosphorylation of ULK1, another substrate of mTORC1 involved in autophagy initiation, was not affected in HOPS knockout cells (Fig EV5D). These data imply that the HOPS complex selectively regulates the mTORC1-dependent control of the MiT/TFE family of transcription factors.

We conclude from these data that *VPS41* patient cells show a decreased association of mTORC1 with lysosomes, a continuous nuclear localization of TFE3, and increased expression of LC3II and other CLEAR proteins. Notably, the autophagy defect is specific for the patient-derived fibroblasts and not observed in any of the parental cell lines. *VPS41^KO^* cells show a similar lysosomal dissociation of mTORC1 and constitutive nuclear localization of TFE3 whereas phosphorylation of S6K1 and 4EBP1 is unaffected. This mTORC1/TFE3 phenotype is HOPS dependent, since depletion of other HOPS subunits results in a similar phenotype. The phenotype cannot be rescued by expression of VPS41^S285P^ or VPS41^R662*^, which is in line with our other findings showing that these variants cannot restore HOPS function.

### VPS41^S285P^ allows for normal regulated secretion in PC12 cells

In secretory cells, VPS41 is required for secretory protein sorting and secretory granule biogenesis in a pathway that is independent of the HOPS complex (Asensio *et al*, 2013; Burns *et al*, 2020). This VPS41 function requires the N-terminal residues 1–36 for interaction with AP-3 and the presence of the C-terminal located CHCR domain (Asensio *et al*, 2013; Margarita Cabrera *et al*, 2010). It was suggested that VPS41 might form a coat on AP-3 containing membranes that exit the TGN (Rehling *et al*, 1999; Darsow *et al*, 2001; Asensio *et al*,

2013). Since VPS41^R662*^ lacks both the RING domain and part of the CHCR domain, a function of this variant in regulated secretion is prohibited (Asensio *et al*, 2013). However, VPS41^S285P^ could theoretically still exert this HOPS-independent function. To test this, we first investigated whether VPS41^S285P^ can interact with AP-3. Pulldowns of recombinant VPS41 constructs with the hinge-ear domain of AP-3D1 showed that VPS41^S285P^ binds AP-3 with equivalent affinity as VPS41^WT^ (Fig 6A and Appendix Fig S10).

To directly test the effect of VPS41^S285P^ on regulated secretion, we made use of PC12^VPS41KO^ cells. By using EGF-ALEXA647 degradation as readout, we established that these cells show a similar endocytosis defect as patient fibroblasts and *VPS41^KO^* HeLa cells (Appendix Fig S11A and B) and also display the HOPS-dependent mTORC1/TFE3 phenotype (Fig EV4E). Regulated protein secretion was measured as previously described (Asensio *et al*, 2010). Briefly, cells were incubated in Tyrode's buffer containing 2.5 mM (basal) or 90 mM (stimulated) K^+ and the supernatant and cell lysates were analyzed by quantitative fluorescent immunoblotting (Fig 6B). As previously shown, depletion of *VPS41* from PC12 cells resulted in decreased cellular SgII protein levels, as well as in a reduction in the regulated secretion of the proteins that are present (Asensio *et al*, 2013). Reintroducing VPS41^WT^ in PC12 *VPS41^KO^* cells rescued cellular SgII levels and recovered regulated secretion (Fig 6C and D). Interestingly, similar results were obtained by expression of the VPS41^S285P^ variant (Fig 6C and D), indicating that its role in regulated secretion has been preserved.

Together, these data show that expression of VPS41^S285P^ rescues the regulated secretory pathway in *VPS41^KO^* PC12 cells. Thus, the VPS41^S285P^ variant is defective in HOPS-dependent endocytosis and autophagy pathways, but retains its HOPS-independent role in regulated secretion.

### Co-expression of VPS41^S285^ and VPS41^R662*^ abolishes neuroprotection in a *C. elegans* model of Parkinson's disease

The clinical symptoms of the *VPS41* patients (Fig 1) overlap with Parkinson's disease (Jankovic, 2008; Reich and Savitt, 2016). Previously, a screen for neuroprotective factors in a transgenic *C. elegans* model for Parkinson's disease showed that overexpression of human VPS41 protects against α-synuclein-induced neurodegeneration (Hamamichi *et al*, 2008; Ruan *et al*, 2010; Harrington *et al*,

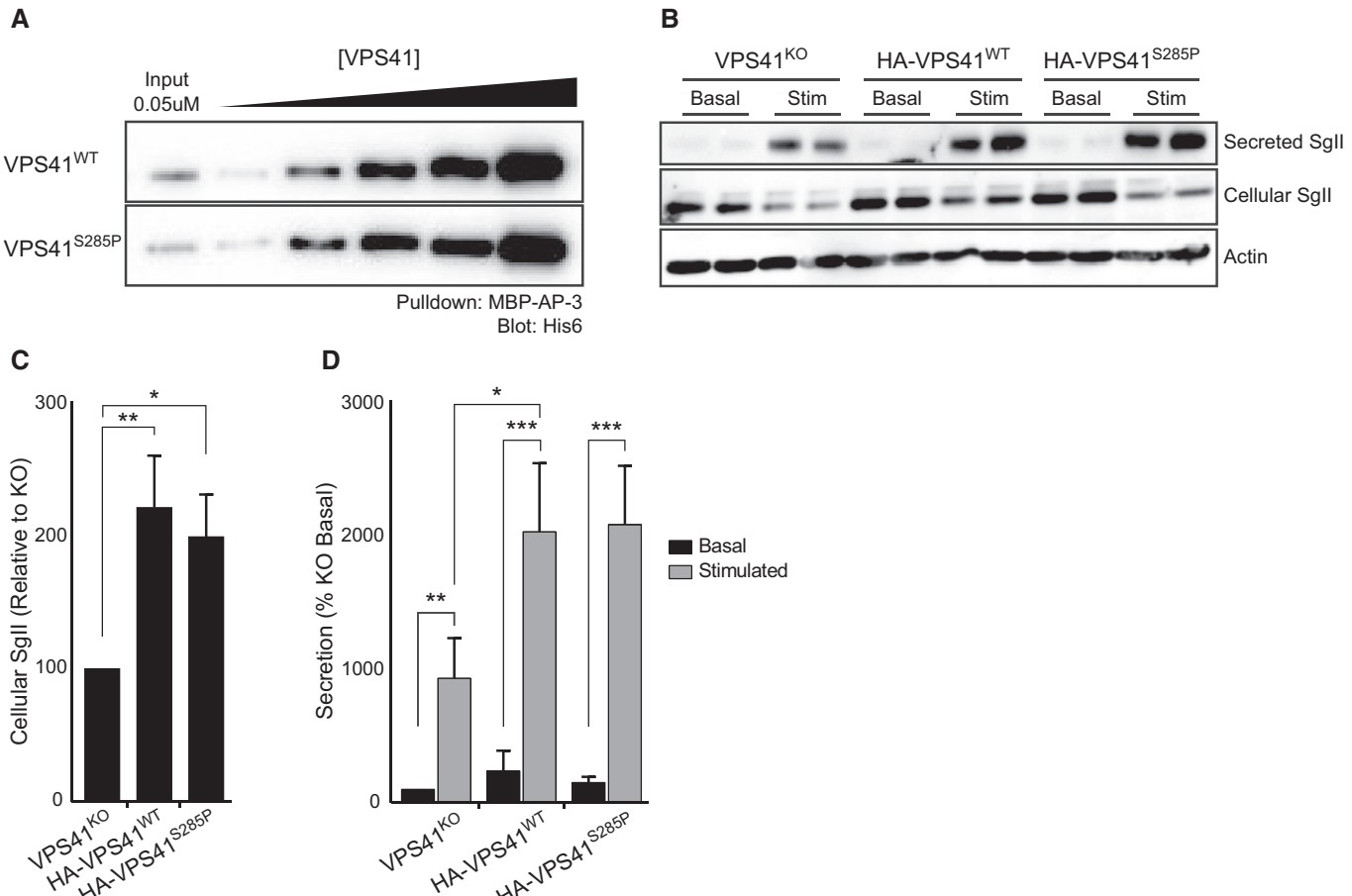

**Figure 6. VPS41^{S285P} rescues regulated secretion in PC12 VPS41^{KO} cells.**

A   Western blot of pulldown of AP-3 incubated with HIS-6 VPS41^{WT} or VPS41^{S285P}. There is no difference in affinity between VPS41^{WT} and VPS41^{S285P} (See Appendix Fig S10 for MBP-AP-3 pulldown input).

B   Western blot on cellular or secreted secretogranin II (SgII) of PC12 cells VPS41^{KO}, or VPS41^{KO} transduced with HA-VPS41^{WT} or HA-VPS41^{S285P} lentivirus. Cells were washed and incubated for 30 min in Tyrode's solution containing 2.5 mM K$^+$ (basal) or 90 mM K$^+$ (stimulated). No difference in secreted SgII levels was observed between VPS41^{WT} and VPS41^{S285P}.

C, D   Cellular (C) and secreted (D) secretogranin II (SgII) were measured by quantitative fluorescence immunoblotting (n = 3). VPS41^{S285P} rescues intracellular SgII levels and regulated secretion of SgII to the same extent as VPS41^{WT}.

Data information: Data are represented as mean ± SEM. *$P < 0.05$, **$P < 0.01$, ***$P < 10^{-4}$. One-way ANOVA (C and D). Exact P-values are reported in Appendix Table S3.

2012). The neuroprotective effect of VPS41 depends on the presence of the WD40 and CHCR domains and ability to interact with Rab7 and AP-3 (Harrington *et al*, 2012; Griffin *et al*, 2018). To explore the impact of patient mutations in this Parkinson's disease model, we made transgenic nematodes, co-expressing the human VPS41 (hVPS41) variants found in patients 1 and 2. We crossed these with isogenic strains overexpressing either GFP alone or human α-synuclein-GFP to mimic Parkinson's disease.

First, we combined VPS41^{WT} with VPS41^{S285P} or VPS41^{R662*} expression, reflecting paternal and maternal conditions of patients 1 and 2, and studied if expression of these mutants induces neurodegeneration in the absence of α-synuclein. At 7 or 10 days post-hatching, there was no statistically significant change in neurodegeneration in any of these strains (Fig 7A and B). Likewise, when we co-expressed VPS41^{S285P} with VPS41^{R662*} there was no discernable increase in neurodegeneration. This demonstrates that co-expression of the

patient-specific *VPS41* variants does not increase neurodegeneration in (aging) animals.

We then analyzed VPS41 neuroprotective function in strains overexpressing α-synuclein. At 7 days post-hatching, only ~43% of the animal population overexpressing α-synuclein retains the full complement of dopaminergic neurons and at 10 days only ~35%. Previous studies show that expression of VPS41^{WT} rescues neurodegeneration at day 7 and day 10, albeit it with reduced efficiency at day 10 (Ruan *et al*, 2010; Harrington *et al*, 2012). Here, we found that co-expression of VPS41^{WT} with either VPS41^{S285P} or VPS41^{R662*} significantly enhanced survival of dopaminergic neurons at day 7, but at day 10 this protective effect was lost (Fig 7C and D). In animals co-expressing VPS41^{S285P} and VPS41^{R662*}, reflecting patient conditions, we found no protective effect at all at either day 7 or 10 (Fig 7C and D). Moreover, neurodegeneration at day 10 was significantly increased indicating that co-expression of the clinical *VPS41*

variants adds additional stress besides α-synuclein overexpressing and aging. Collectively, these data indicate that the capacity of VPS41 to protect dopaminergic neurons against α-synuclein overexpression is lost upon co-expression of the patient variants. This implies that VPS41-mediated neuroprotection depends on the HOPS complex.

## Discussion

The vacuolar protein sorting-associated protein 41 (VPS41) has a major contribution in vesicle-mediated trafficking to lysosomal compartments including endocytic transport and the autophagic pathway. VPS41 is part of the HOPS complex, a multisubunit

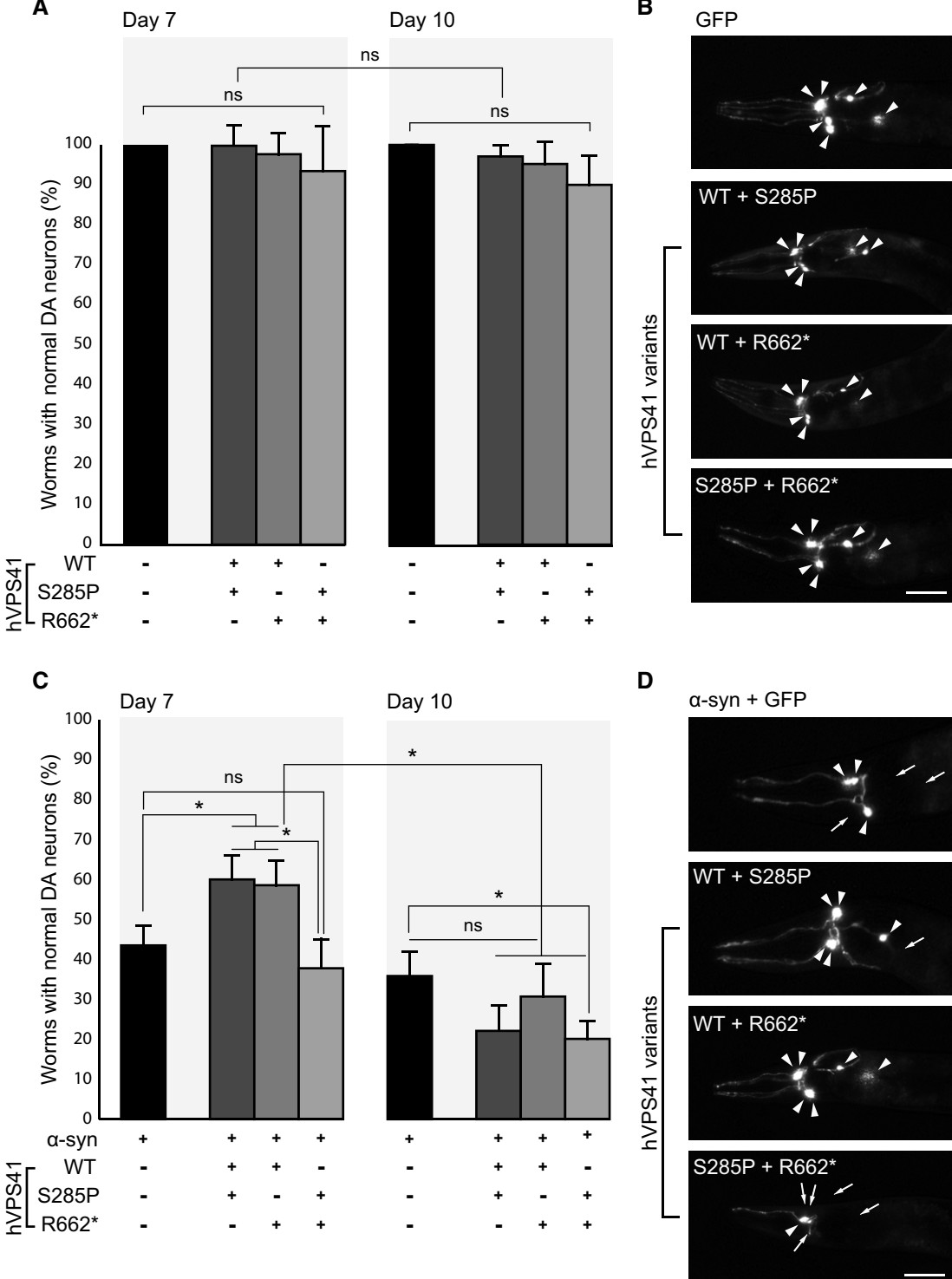

**Figure 7.**

**Figure 7.   Compound heterozygous expression of hVps41 variants fails to rescue α-synuclein-induced neurodegeneration in *C. elegans*.**

A   Graph representing the percentage of adult *C. elegans* animals with 6 normal dopaminergic neurons. Heterozygous expression with hVPS41$^{WT}$ (strains UA386 or UA387) or compound heterozygous expression of hVPS41 variants (strain UA388) does not yield significant changes in neurodegeneration compared with the P$_{dat-1}$:: GFP (strain BY250) control at either day 7 or day 10 post-hatching. *n* = 30 Adult worms for each of 3 independent experiments for GFP (total of 90 worms) and *n* = 90 for each independent transgenic strain (30 worms/trial × 3 independent transgenic lines = 270 worms), for each of 3 independent experiments.

B   Representative images of the neuroprotection assay described in (A), where worms express GFP specifically in the anterior 6 DA neurons. Intact DA neurons are indicated with arrowheads. hVPS41 variant backgrounds are expressed as indicated. Scale bar 20 μm.

C   Graph representing the percentage of animals with 6 normal dopaminergic neurons in the anterior region of P$_{dat-1}$::GFP; P$_{dat-1}$:: α-syn (strain UA44) animals with heterozygous expression of hVPS41 variants. Heterozygous expression of hVPS41$^{WT}$ with either variant (strains UA389 or UA390) significantly rescues neurons from α-synuclein-induced degeneration at day 7 whereas compound heterozygous expression (strain UA391) fails to rescue neurodegeneration. At day 10, none of the heterozygous backgrounds significantly rescues α-synuclein-induced neurodegeneration. *n* = 30 Adult worms for each of 3 independent experiments for the α-synuclein (α-syn) strain (total of 90 worms) and *n* = 90 for each independent transgenic strain (30 worms/trial × 3 independent transgenic lines = 270 worms) for each of 3 independent experiments.

D   Representative images of DA neurons from *C. elegans* expressing P$_{dat-1}$::GFP; P$_{dat-1}$:: α-syn, with or without hVPS41 variants, as described in (C). Intact DA neurons are indicated with arrowheads and missing neurons are indicated with arrows. Scale bar 20 μm.

Data information: Data are represented as mean ± SD. *$P$ < 0.05. One-way ANOVA with Tukey's correction (A and C). Exact *P*-values are reported in Appendix Table S3.

tethering complex which mediates fusion of lysosomes with late endosomes and autophagosomes (Radisky *et al*, 1997; Nickerson *et al*, 2009). Independently of the HOPS complex, VPS41 is involved in the transport of lysosomal membrane proteins from the trans-Golgi network (Pols *et al*, 2013b) and in regulated secretion of neuropeptides (Asensio *et al*, 2013). Intriguingly, VPS41 was previously identified as neuroprotector in *C. elegans* and mammalian models of Parkinson's disease (Ruan *et al*, 2010; Harrington *et al*, 2012; Griffin *et al*, 2018). Here, we present three patients with a neuronal disorder, bearing compound heterozygous variants in *VPS41* and show the consequences of this on the molecular and cellular level. We demonstrate that all *VPS41* variants give rise to a non-functional HOPS complex and a kinetic defect in the delivery of endocytic and autophagic cargo to enzymatically active lysosomes. Most strikingly, we find that lack of HOPS function causes increased dissociation of mTORC1 from lysosomes and the transfer of TFE3 to the nucleus, resulting in continuously increased autophagy levels. By contrast, phosphorylation of mTORC1 substrates S6K and 4EBP1 is not affected. Our studies thereby link HOPS complex functionality to mTORC1 signaling, specifically for the TFEB/TFE3 axis.

Patients 1 and 2, two brothers, were diagnosed with global developmental delay, hypotonia, ataxia, and dystonia. By exome sequencing, we identified a missense variant in the WD40 domain (*VPS41$^{S285P}$*) and a nonsense variant in the CHCR domain resulting in a premature stopcodon (*VPS41$^{R662*}$*). The third patient, presenting with severe developmental delay, axial hypotonia, and spastic quadriplegia, was identified with a splice site variant in the TPR-like domain (*VPS41$^{c.1423-2A>G}$*) and the same nonsense variant in the CHCR domain as observed in the siblings (*VPS41$^{R662*}$*). By Western blot, we found that only the VPS41$^{S285P}$ variant was expressed at significant levels in patient fibroblasts. The truncated VPS41$^{R662*}$ form could not be detected at all, whereas expression levels of *VPS41$^{c.1423-2A>G}$* were strikingly low. Hence, cells from patients 1 and 2 only contained the VPS41$^{S285P}$ variant, whereas patient 3 only contained VPS41$^{c.1423-2A>G}$ at about 10% of wild-type VPS41 levels which could explain the more severe phenotype observed in patient 3. The three patients described here share features with a recently reported patient bearing a homozygous *VPS41* splice site variant (NM_014396.3c.450 + 1G>T) (Steel *et al*, 2020). All patients show the autosomal recessive inheritance pattern, early onset generalized dystonia with developmental delay, and brain MRI characterized by thinning of the corpus callosum and atrophy of the cerebellar

vermis. One of our patients (Patient 1) presents a slightly different phenotype, characterized by a less severe intellectual disability and ataxia without overt dystonia, clearly indicating that the phenotypic expression of *VPS41*-related disorders is variable.

Since *VPS41$^{S285P}$* is the only variant expressed at substantial levels, we investigated whether it could form a functional HOPS complex. By co-IPs using tagged constructs, we found that VPS41$^{S285P}$ interacts with other HOPS subunits and by immunofluorescence that this variant is recruited to endolysosomal membranes. However, expression of VPS41$^{S285P}$ in *VPS41$^{KO}$* HeLa cells did not rescue the HOPS-dependent fusion between lysosomes and late endosomes or autophagosomes. Since a serine to proline substitution is predicted to affect protein folding, the inability of VPS41$^{S285P}$ to form a functional HOPS complex may be due to a folding defect. Importantly, these data demonstrate that all patient variants affect HOPS function, either because of lack of expression and/or an inability to form a functional complex. In accordance, HOPS-dependent transport of endocytosed Dextran to enzymatically active lysosomes is significantly decreased in patient cells as compared to the healthy controls. Intriguingly, the *VPS41$^{WT/R662*}$* cells of the mother of the two siblings also showed decreased delivery to lysosomes. This effect was apparent after 2 h Dextran incubation but disappeared after 5 h. By contrast, lysosomal delivery in patient cells reached control levels only after 24 h uptake. This shows that expression of the *VPS41* variants causes a delay rather than a block in HOPS-dependent transport to lysosomes and implies that the kinetics of cargo delivery to lysosomes is a factor of importance in the development of *VPS41*-associated pathology.

Independent of HOPS, VPS41 is also required for formation of secretory granules and transport of lysosomal membrane proteins via the ALP/LAMP-carrier pathway (Cowles *et al*, 1997b; Darsow *et al*, 2001; Swetha *et al*, 2011; Pols *et al*, 2013b). Both functions were reported to depend on the interaction of VPS41 with AP-3 (Cowles *et al*, 1997a; Rehling *et al*, 1999; Darsow *et al*, 2001; Angers & Merz, 2009; Cabrera *et al*, 2010). By co-IP, we found that VPS41$^{S285P}$ still binds AP-3 with an affinity comparable to VPS41$^{WT}$. By immuno-EM of patient-derived fibroblasts, we found no effect on the concentration of LAMP-1 or LAMP-2 in lysosomal membranes. Also, cathepsin B and cathepsin D were readily transported to and activated in lysosomes. Though these steady state assays do not give information on transport kinetics, they do indicate that lysosomes in patient cells contain normal levels of lysosomal proteins.

Interestingly, we found that expression of VPS41$^{S285P}$ fully rescued the secretion defect in VPS41$^{KO}$ PC12 cells. Likewise, in a parallel study we found that expression of VPS41$^{S285P}$ in VPS41$^{KO}$ INS-1 cells restored insulin secretion (Burns *et al*, 2020). This shows that expression of VPS41$^{S285P}$ fails to restore HOPS function, but does rescue the regulated secretion pathway in PC12 and INS cells. It is conceivable that this activity of VPS41$^{S285P}$ in secretion is important for disease development, but how exactly VPS41 regulates secretion —direct or indirect?—remains to be elucidated. In future studies, we can use the VPS41$^{S285P}$ variant to distinguish between these HOPS-dependent and HOPS-independent functions of VPS41.

A striking finding in our studies is that expression of the *VPS41* variants leads to a decreased lysosomal localization of mTORC1, a constitutive nuclear localization of TFE3, and impaired TFEB phosphorylation. This was seen in patient-derived fibroblasts and recapitulated in HeLa and PC12 cells knockout for *VPS41*, indicating that this phenotype occurs across different cell types. TFE3 and TFEB are part of the MiTF/TFE family of transcription factors (Slade & Pulinilkunnil, 2017). Both TFE3 and TFEB translocate to the nucleus upon nutrient starvation and bind CLEAR elements in promotors of autophagic and lysosomal genes (Sardiello *et al*, 2009; Palmieri *et al*, 2011). In *VPS41* disease cells, the continuous nuclear localization of TFE3 results in elevated levels of target genes like LAMP-1 and cathepsin B, which is in agreement with the observed increase in number of small-sized, LAMP-1-positive compartments. Even though lysosomal size is controlled by a diversity of pathways (de Araujo *et al*, 2020), we speculate that due to HOPS complex dysfunctionality and subsequent impaired fusion between endolysosomal compartments, these newly formed organelles remain small because of decreased membrane input, as is observed in both patient and *VPS41*$^{KO}$ cells.

It is tempting to speculate that impaired endocytic cargo delivery deprives lysosomes of nutrients, thereby causing a state of starvation. However, phosphorylation of S6K and 4EBP1, canonical substrates of mTORC1, was not affected in cells depleted of a functional HOPS complex. Indeed, also *VPS18*$^{KO}$ and *VPS39*$^{KO}$ cells showed constitutive nuclear localization of TFE3, but normal S6K and 4EBP1 phosphorylation in response to nutrient starvation and replenishment. Also, phosphorylation of ULK1 was not affected in any of the knockout cells. These findings suggest that HOPS-deficient cells are not continuously starved or impaired in nutrient sensing, but that the HOPS complex specifically regulates mTORC1-dependent control of the TFEB/TFE3 axis. Interestingly, cells depleted of Folliculin (FLCN), the GTPase activating protein (GAP) for the lysosome-associated RagC/D GTPases necessary for membrane recruitment of mTORC1, are also selectively affected in the TFEB/TFE3 axis, with no effect on S6K and 4EBP1 phosphorylation (Lawrence *et al*, 2019). As explanation for a differential regulation of mTORC1 substrates, it was recently proposed that amino acids, and not growth factors, are required for FLCN-mediated regulation of TFEB/TFE3 (Napolitano *et al*, 2020). Possibly the HOPS complex, by facilitating late endosome–lysosome fusion, is required for the delivery of degradable material for the supply of amino acids. This endolysosomal fusion might not be necessary for the growth factor-dependent regulation of S6K/4EBP1. Indeed, cells treated with chloroquine, which recently was shown to block late endosome–lysosome fusion, also show translocation of TFE3 to the nucleus (Roczniak-ferguson *et al*, 2012; Mauthe *et al*, 2018).

Additionally, HOPS could be involved in the association of mTORC1 and/or TFEB/TFE3 with lysosomes. TFEB/TFE3 and mTORC1 both bind to active Rags on the lysosomal membrane, thereby enabling the phosphorylation of TFEB/TFE3 (Martina & Puertollano, 2013). By contrast, S6K and 4EBP1 do not associate with the lysosomal membrane and, therefore, lysosomal association of mTORC1 is less crucial for phosphorylation of these latter substrates (Napolitano *et al*, 2020). Since patient and *VPS41*$^{KO}$ cells show decreased association of mTORC1 with the lysosomal membrane, an interesting hypothesis is that the HOPS complex enables lysosomal association and retention of mTORC1 and TFE3/TFEB, possibly by cooperating with the Rags.

Currently, patients with mutations in HOPS core components *VPS11*, *VPS16*, *VPS33A*, and *VPS41* have been reported in literature, which all show a neurodegenerative phenotype (Chintala *et al*, 2009; Edvardson *et al*, 2015; Zhen & Li, 2015; Cai *et al*, 2016; Hörtnagel *et al*, 2016; Kondo *et al*, 2016; Zhang *et al*, 2016; Dursun *et al*, 2017; Beek *et al*, 2019; Pavlova *et al*, 2019; Steel *et al*, 2020). During our studies, an additional patient with a homozygous splice site variant (NM_014396.3c.450 + 1G>T) in *VPS41* was identified, displaying a neurological phenotype similar to the patients discussed here (Steel *et al*, 2020). For future studies, it will be interesting to obtain cells from this patient and screen for the mTORC1/TFE3 phenotype.

VPS41 was previously reported to confer neuroprotection in both *C. elegans* and mammalian models for Parkinson's disease (Hamamichi *et al*, 2008; Ruan *et al*, 2010; Harrington *et al*, 2012; Griffin *et al*, 2018). Overexpression of α-synuclein results in degeneration of dopaminergic neurons, which is prevented by overexpression of human *VPS41*. Both the WD40 and CHCR domain of *VPS41* are necessary for this neuroprotection (Ruan *et al*, 2010; Harrington *et al*, 2012), which are mutated in VPS41$^{S285P}$ and VPS41$^{R662*}$, respectively. We found that transgenic nematodes co-expressing hVPS41$^{WT}$ with either VPS41$^{S285P}$ or VPS41$^{R662*}$retained the neuroprotective effect, resulting in a decrease in neurodegeneration. However, co-expressing VPS41$^{S285P}$ and VPS41$^{R662*}$ did not result in neuroprotection and even exacerbated cell death in aging worms. Since we found that the *VPS41* patient cells completely lack HOPS function, the neurodegenerative phenotype in the nematodes is likely HOPS related. Moreover, since HOPS-dependent autophagosome–lysosome fusion is important for clearance of aggregated material, which particularly tends to accumulate in neurons over the course of aging (Takáts *et al*, 2009; Jiang *et al*, 2014; McEwan *et al*, 2015; Jia *et al*, 2017), overexpression of VPS41 in the *C. elegans* model possibly induces protein clearance through autophagy. The intersection of lysosomal dysfunction and neurodegeneration has become a critical junction in the mechanistic underpinnings of Parkinson's disease (Mazzulli *et al*, 2011). This is best exemplified by mutations in the gene encoding the lysosomal protein GBA1, which causes Gaucher's disease and has become the most prevalent genetic risk factor for Parkinson's disease (Do *et al*, 2019). In the context of these results, it is tempting to speculate that an increased understanding of autolysosomal trafficking defects in *VPS41* patient cells may serve to inform therapeutic strategies for neurodegenerative disorders (Griffin *et al*, 2018).

In summary, our data imply that *VPS41* patients may represent a novel class of lysosomal disorders in which lysosomes are enzymatically active, but due to delayed trafficking kinetics inefficiently

reached by endocytic and autophagic cargo. Moreover, patient cells are selectively inhibited in mTORC1-mediated TFEB/TFE3 regulation. Interestingly, many other neurodegenerative diseases show an opposite phenotype, i.e., with hyperactivity of mTORC1 (An *et al*, 2003; Wong, 2013; Lam *et al*, 2017). These observations are of particular importance with regard to the design of a possible treatment of VPS41 and other HOPS-related disorders (Chintala *et al*, 2009; Laplante & Sabatini, 2012; Wong, 2013; Dursun *et al*, 2017; Pavlova *et al*, 2019).

# Materials and Methods

## Patient material and identification of *VPS41* variants

Informed consent was obtained from all subjects and the experiments conformed to the principles set out in the WMA Declaration of Helsinki and the Department of Health and Human Services Belmont Report. Parents provided written informed consent for publication of clinical pictures. Information about whole exome sequencing, filtering, and variant analysis is presented in Appendix Supplementary Methods.

## Antibodies, reagents, and lysosomal functionality assays

Antibodies and reagents used in this study and their specific dilutions or used concentrations are specified in Appendix Table S2. To study endocytosis, cells were incubated for 2 h with 10,000 MW Dextran-Alexa Fluor 568 (Invitrogen), washed five times using warm PBS, fixed with 4% PFA, embedded in Prolong DAPI (Invitrogen), and imaged. Lysosomal acidity was determined using Lysotracker™ Red (Molecular Probes) for 30 min after which cells were washed five times using 37°C PBS, fixed with 4% PFA, embedded in Prolong DAPI (Invitrogen), and imaged. To visualize active lysosomes, we used the SiR-Lysosome (Spirochrome) probe. Cells were incubated for 3 h, washed with PBS, fixed with 4% PFA, embedded in Prolong DAPI (Invitrogen), and imaged. For the visualization of cathepsin B-active compartments, cells were incubated with MagicRed cathepsin B substrate (Molecular Probes) for 30 min and imaged at RT using live-cell imaging. To assess EGF degradation, WT and *VPS41KO* PC12 cells were washed twice with PBS and starved of serum for 2 h in DMEM with 0.1% BSA (GoldBio). During starvation, EGF-biotin (GoldBio) streptavidin-647 conjugate was prepared. EGF-biotin (5 μg/ml) was incubated for 30 min at 4°C at a 5:1 ratio to streptavidin-Alexa647 (Life Technologies). Following starvation, cells were washed twice with ice-cold PBS on ice and incubated with EGF-A647 conjugate at a final concentration of 100 ng/ml for 1 h on ice. Excess unbound EGF was removed by washing with ice-cold PBS with 0.5% BSA. Cells were chased for indicated times at 37°C before fixation with 4% PFA in PBS for 20 min at room temperature. Cells were analyzed by flow cytometry (CyAn ADP Analyzer, Beckman Coulter, USA) or scanning confocal microscopy.

For the RT–PCR, mRNA was isolated from cells using RNeasy kit (Qiagen) and converted to cDNA using TaqMan reverse transcriptase reagents (Applied Biosystems). Real-time PCR was performed using PCR mastermix with SYBR green (Applied Biosystems) with primers that were ordered from realtimeprimers.com.

## Cell culture and light and electron microscopy

Patient-derived fibroblasts and HeLa cells were cultured in High Glucose Dulbecco's Modified Eagle's Medium (Invitrogen) supplemented with 10% FCS in a 5% $CO_2$-humidified incubator at 37°C. Cells were transfected using X-tremeGENE™ HP (Roche) according to manufacturer's instructions. To induce autophagy, cells were starved for 2 h at 37°C using EBSS (Thermo Fisher) and restimulated for 15 min with complete medium. Cells destined for immunofluorescence were washed with PBS once and fixed using 4% wt/vol paraformaldehyde (PFA, Polysciences Inc.) for 20 min. Cells were washed three times with PBS and permeabilized with 0.1% Triton X-100 (Sigma) for 5 min. Samples were blocked for 15 min using a 1% BSA solution. Samples were labeled, embedded in Prolong DAPI (Invitrogen), and imaged. When imaging transfected cells, cells with similar transfection levels were selected. Images were taken using a Deltavision wide-field microscope (Applied Precision) with a 100x/1.4A oil immersion objective. Image analysis was performed using Volocity software (Perkin Elmer) and macros written in Fiji (Schindelin *et al*, 2012). The ComDet plugin was used for dot detection for quantifications of immunofluorescent images (Eugene Katrukha, Utrecht University). For TFE3 localization experiments, cells were scored positive when a defined nuclear outline was visible due to nuclear labeling of TFE3.

For resin EM, cells grown in 6cm dishes were fixed in 2% wt/vol PFA, 2.5% wt/vol GA (Electron Microscopy Sciences) in Na-cacodylate buffer (Karnovsky fixative) for 2 h at RT. Subsequently, the fixative was replaced for 0.1 M Na-cacodylate buffer, pH 7.4. Postfixation was performed using 1% wt/vol $OsO_4$, 1.5% wt/vol K3Fe(III)(CN)6 in 0.065 M Na-cacodylate buffer for 2h at 4°C. Next, cells were stained with 0.5% uranyl acetate for 1 h at 4°C, dehydrated with ethanol, and embedded in Epon (Electron Microscopy Sciences). Ultrathin sections were stained with uranyl acetate and lead citrate using AC20 (Leica). Images were taken on a TECNAI T12 electron microscope (FEI Thermo Fisher Scientific).

For immuno-EM, cells were grown in 6cm dishes were incubated with BSA-Au[5] (Cell Microscopy Core, UMC Utrecht) for 2h in culture medium at 37°C, washed with medium, and fixed by adding freshly made 2% FA, 0.2% GA in 0.1 M phosphate buffer (pH 7.4) to an equal volume of medium for 15 min. After 15 min, the fixative was refreshed to continue fixation for 2h at RT. Cells were stored in 1% formaldehyde at 4°C until further processing. Then, cells were washed with PBS/0.05 M glycine, scraped in 1% gelatin/PBS, pelleted in 12% gelatin/PBS, solidified on ice, and cut into small blocks. The blocks were infiltrated in 2.3 M sucrose overnight at 4°C, mounted on aluminum pins, and frozen in liquid nitrogen (Slot & Geuze, 2007). Ultrathin sections were prepared using an ultra-cryomicrotome (Leica). To pick up ultrathin sections, a 1:1 mixture of 2.3 M and 1.8% wt/vol methylcellulose was used. Ultrathin sections were immuno-gold labeled and stained as described (Slot & Geuze, 2007). Protein A-conjugated colloidal gold particles were made in house (Cell Microscopy Core, UMC Utrecht).

## CRISPR/Cas9 knockout cells

### HeLa^VPS41KO

Hela cells were transiently transfected with pSpCas9(BB)-2A-GFP (PX458) (Ran *et al*, 2013) encoding sgRNAs targeting *VPS41* using

X-tremeGENE (Merck) according to the manufacturers recommendation. sgRNAs were designed with http://crispor.tefor.net/ (Haeussler et al, 2016) and target coding sequences in the N-terminal region encoded by the third coding exon. The sgRNA used is sgRNA2 AAGTATTTCAGTTACCCCAT. GFP-positive cells were sorted using a FACSAria II flow cytometer (BD) and plated in 10-cm dishes. Colonies were picked from these plates after 1 week and expanded. To confirm VPS41 absence, total cell lysates were analyzed by Western blotting using mouse anti-VPS41 (SC-377271, Santa Cruz).

### HeLa$^{VPS11KO/VPS18KO/VPS39KO}$

To generate VPS11, VPS18, and VPS39 knockout HeLa cells, a WT HeLa parental stain was transfected with pSpCas9(BB)-2A-Puro (PX459) plasmids containing gsRNAs for VPS11, VPS18, or VPS39 using Effectene transfection reagent (QIAGEN). gsRNAs were designed using against the first exons of VPS11, VPS18, and VPS39 using an optimized CRISPR design tool. sgRNA sequences that generated the knockout cells used in this study were GTG TGTCACCCGTAGTTTGT for VPS11, CGAGAACTCGCTGTCCCGCT for VPS18, and GCCTCTGCAAATCGACTGTC for VPS39. We selected for transfected cells using 1 µg/ml Puromycin. Surviving cells were recovered for 2 weeks and split into 96-well plates at single cell/well dilution. Single cell colonies were selected and expanded. For VPS18 and VPS11 clones, knockout was confirmed by the absence of VPS18 or VPS11 protein on Western blot. VPS39 knockout clones were confirmed via genomic DNA isolation followed by PCR of VPS39 exon 1, sequencing, and analysis using the TIDE sequence analysis tool (https://tide.nki.nl/). The selected VPS39 knockout clone had two deletions of 5 and 8 nucleotides in the first exon, resulting in a frameshift.

### PC12$^{VPS41KO}$

The human codon-optimized Cas9 and chimeric guide RNA expression plasmid (pX459v2) developed by the Zhang laboratory were obtained from Addgene (Ran et al, 2013). To generate gRNA plasmids, a pair of annealed oligos (20 base pairs) were ligated into the single guide RNA scaffold of pX459v2. The following gRNAs sequences were used: Forward (rat): 5′-CACCGACTCTCAGACT GAGCTATGG-3′; Reverse (rat): 5′-AAACCCATAGCTCAGTCTGA GAGTC-3′ to generate the PC12$^{VPS41KO}$ line. Rat VPS41 lentiviral plasmid was generated by amplifying VPS41 from rat cDNA using the following primers: WT Forward: 5′-CCTCCATAGAAGACAC CGACTCTAGACACCATGGCGGAAGCAGAGGAG-3′; WT Reverse: 5′-TATGGGTAACCCCCAGATCCACCGGTCTTCTTCATCTCCAGGAT GGCA-3′. The PCR products were then subcloned by Gibson ligation into pLenti-CMV-GFP-Puro. pLenti-CMV-GFP-Puro was a gift from Paul Odgren (Addgene plasmid #73582). To test for the presence of indels, the resulting PCR products were ligated into pBlueScript II KS. Isolated plasmids from 10 random colonies were analyzed for the presence of indels by Sanger sequencing. Insertions in the initiator codon of the VPS41 gene were observed in all clones of either a single or a triple T.

### Co-immunoprecipitation and Western blotting

Cells were seeded in 15-cm dishes and transfected with the appropriate constructs as indicated in the respective figures and according to abovementioned transfection protocols. Cells were washed three times with ice-cold PBS and lysed using a CHAPS lysis buffer (50 mM Tris pH = 7.5, 150 mM NaCl, 5 mM MgCl$_2$, 1mM DTT, 1% (w/v) CHAPS) complemented with protease and phosphatase inhibitor (Roche). Cells were scraped, collected, and spun down at 16,000 rcf/g for 15 min at 4°C. Protein levels were equalized between samples using a Bradford Protein Assay (Bio-Rad). 10% of the sample was saved as input control. The remainder of the samples was incubated for 1 h with uncoated protein G beads (Millipore) to remove a-specifically bound proteins. Beads were spun down and the supernatant was incubated with beads together with 2 µg antibody against the prey. Samples were incubated overnight, extensively washed, and eluted using SDS sample buffer and run on precast gradient (4–20%) gels (Bio-Rad). Gels were transferred using Trans-Blot® Turbo™ RTA Mini PVDF Transfer Kit and the Trans-Blot® Turbo™ Transfer system (Bio-Rad). Membranes were blocked with Odyssey® Blocking Buffer (LI-COR) in PBS for 1h at RT and incubated with primary antibody diluted in Odyssey® Blocking Buffer (LI-COR) in 0.1% TBST overnight at 4°C, rocking. Membranes were washed extensively with 0.1% TBST and incubated with secondary antibodies, diluted in Odyssey® Blocking Buffer (LI-COR) in 0.1% TBST at RT for 1 h, rocking. The membranes were again washed extensively with 0.1% TBST, followed by 2 washing steps with PBS and 1 washing step with MiliQ. Membranes were scanned using the Amersham™ Typhoon™ Laser Scanner (GE Healthcare Life Sciences).

### Plasmids

The GFP-VPS41 constructs were cloned from hVPS41 cDNA (Origene) into a pDonor201 vector (Invitrogen) using PCR. Next, using the Gateway system and GFP-pcDNA-Dest53 (kindly provided by Prof. R. Roepman, Radboud University Medical Center Nijmegen), a recombination reaction was performed. The VPS41 mutant variants were made using the QuickChange Site-Directed Mutagenesis kit (Agilent) using the primers 5′-CAGCTTGTTGTACTTCCGTATGT AAAGGAGA and 5′-TCTCCTTTACATACGGAAGTACAACAAGCTG to generate GFP-VPS41$^{S285P}$ and 5′-GTTTATCTTCTGAGCTGAATG GGTTAATAGCC and 5′-GGCTATTACCCATTCAGCTCAGAAGATA AAC to generate GFP-VPS41$^{R662*}$. The pcDNA3-APEX2-V5 construct (kindly provided by Dr. S. de Poot) was used as empty vector in pulldown experiments. VPS41-APEX2-V5 was generated via a BamHI insert using pEGFP-C1-mCherry-VPS41 (kindly provided by Prof. J.J.C. Neefjes, University Medical Center Leiden) using the primers 5′-AGAGGGATCCATGGCGGAAGCAGAGGAGCAG and 5′-AGAGTCTAGACTATTTTTTCATCTCCAAAATTG. VPS41$^{S285P}$-APEX2-V5 and VPS41$^{R662*}$-APEX2-V5 were generated via an identical approach using pEGFP-C1-mCherry-VPS41$^{S285P}$ and pEGFP-C1-mCherry-VPS41$^{R662*}$. These VPS41 variants were made using the abovementioned Mutagenesis kit (Agilent) and the same primer pairs.

For the C. elegans experiments, the wild-type (wt) P$_{dat-1}$::hVPS41 construct was assembled as previously described (Harrington et al, 2012). Briefly, human VPS41 cDNA was obtained from Open Biosystems (Huntsville, AL). Plasmid entry vectors were generated using Gateway Technology (Invitrogen) to clone PCR amplified constructs into pDONR221 to generate a hVPS41 entry clone. The hVPS41 entry clone was used to clone hVPS41 into the Gateway expression vector, pDEST-P$_{dat-1}$. The mutant hVPS41 alleles were generated by site-directed mutagenesis of the P$_{dat-1}$::hVPS41 construct using the

primers 5'-ATTCTACATCAGTGGACTTGCACCTCTCTGTGATCAGC
TTGTTGTACTTCCGTATGTAAAGGAGATTTCAGAAAAAACGGAAA
GAGAATACTGTGCCAG and 5'-CTGGCACAGTATTCTCTTTCCG
TTTTTTCTGAAATCTCCTTTACA TACGGAAGTACAACAAGCTGA
TCACAGAGAGGTGCAAGTCCACTGATGTAGAAT to generate the
$P_{dat\text{-}1}$::hVPS41$^{S285P}$ construct and 5'-GATCTGTCAACAGAGAAAC
TTTGTAGAAGAGACAGTTTATCTTCTGA GCTGAATGGGTAATAGC
CGAAGTGCCCTGAAGATGATTATGGAGGAATTACA and 5'-TGTA
ATTCCTCCATA ATCATCTTCAGGGCACTTCGGCTATTACCCATTCA
GCTCAGAAATAAACTGTCTCTTCTACAAAGTTTCTCTGTTACAGATC
to generate the $P_{dat\text{-}1}$::hVPS41$^{R662*}$ construct. The identities of
the Gateway entry and expression constructs were validated by
DNA sequencing.

### *Caenorhabditis elegans* strains

Transgenic *C. elegans* strains were generated by directly injecting a
$P_{unc\text{-}54}$::tdTomato co-injection marker (50 ng/µl) and $P_{dat\text{-}1}$::hVPS41
wt (25 ng/µl) with $P_{dat\text{-}1}$::hVPS41$^{S285P}$ (25 ng/µl) or $P_{dat\text{-}1}$::
hVPS41$^{R662*}$ (25 ng/µl) constructs or $P_{dat\text{-}1}$::hVPS41$^{S285P}$ (25 ng/µl)
with $P_{dat\text{-}1}$::hVPS41$^{R662*}$ (25 ng/µl) into the gonads of strain N2 Bris-
tol hermaphrodites. Progeny of injected animals were screened for
expression of the hVPS41 expression constructs by evidence of fluo-
rescence in the body-wall muscles from the $P_{unc\text{-}54}$::tdTomato co-
expression marker. Several stable lines of each were isolated and
crossed with isogenic UA44 ($baIn11$[$P_{dat\text{-}1}$::α-syn, $P_{dat\text{-}1}$::GFP]) and
BY250 ($vtIs7$[$P_{dat\text{-}1}$::GFP]) males. Heterozygous hermaphroditic
progeny expressing both tdTomato and GFP were transferred and
allowed to self-fertilize. Several lines of each $P_{dat\text{-}1}$::hVPS41wt + $P_{dat\text{-}1}$::hVPS41$^{S285P}$ [baEx215] in UA44 (UA389) and BY250 (UA386) were
analyzed for neurodegeneration at days 7 and 10. Likewise, $P_{dat\text{-}1}$::
hVPS41wt + $P_{dat\text{-}1}$::hVPS41$^{R662*}$[baEx216] in UA44 (UA390) and
BY250 (UA387) was analyzed. Additionally, $P_{dat\text{-}1}$::hVPS41$^{S285P}$ + $P_{dat\text{-}1}$::
hVPS41$^{R662*}$ [baEx217] in the UA44 (UA391) or BY250 (UA388) back-
grounds was analyzed. For all strains, three representative lines of each
were selected for further analysis.

### Dopaminergic neurodegeneration analysis in *Caenorhabditis elegans*

*Caenorhabditis elegans* dopaminergic neurons were analyzed for
degeneration as previously described (Hamamichi *et al*, 2008;
Harrington *et al*, 2012). Briefly, synchronized progeny from the
BY250, UA44, and experimental *hVPS41* variant strains were
produced from a 3-h egg-lay and grown at 20°C. Animals were
analyzed at days 7 and 10 post-hatching (4- and 7-day-old adults).
On the day of analysis, the 6 anterior dopaminergic neurons [4 CEP
(cephalic) and 2 ADE (anterior deirid)] were examined in 30 adult
hermaphrodite worms, which were immobilized on glass cover slips
using 3 mM levamisole and transferred onto 2% agarose pads on
microscope slides. In quantifying neurodegeneration, animals are
scored as "normal" when a full complement of all 6 anterior
dopaminergic neurons are present and the neuronal processes are
fully intact, as previously reported (Cao *et al*, 2005; Hamamichi *et
al*, 2008; Harrington *et al*, 2010). In total, at least 90 adult worms
(30 worms/trial × 3 trials = 90 total animals) were analyzed for
each independent strain and at least 270 adult worms were analyzed
for each independent transgenic strain (30 worms/trial × 3

### The paper explained

#### Problem

VPS41 is, as part of the HOPS complex, required for lysosomal fusion
events. Independently of HOPS, VPS41 is required for formation of
secretory granules. VPS41 prevents degeneration of dopaminergic
neurons overexpressing the Parkinson's disease-related protein α-
synuclein. Here, we present three patients bearing compound
heterozygote mutations in *VPS41* displaying a severe neurological
disorder. We address the question how these mutations affect endo-
cytosis, autophagy, secretory pathways, and neuroprotection.

#### Results

We show that mutations in or depletion of VPS41 causes a delay in
HOPS-dependent endocytic and autophagic cargo delivery to enzy-
matically active lysosomes and that neuroprotection against α-synu-
clein is reduced. Moreover, the disease-causing mutations specifically
impair mTORC1 activity toward TFE3 and TFEB, causing constitutive
activation of these transcription factors. As a result, lysosomal biogen-
esis and autophagosome formation is continuously upregulated inde-
pendent of nutrient conditions. By contrast, HOPS-independent
function of VPS41 in secretory transport is preserved.

#### Impact

Our study shows that *VPS41* patients represent a new class of lysoso-
mal disorders in which lysosomes are functional, but insufficiently
reached by cargo due to a trafficking defect. Besides this, we show
that VPS41, as part of the HOPS complex, is involved in the differen-
tial regulation of mTORC1 toward TFE3/TFEB, which is of potential
importance for treatment of HOPS-related disorders.

independent transgenic lines × 3 trials = 270 total animals/con-
struct). Values of each independent experiment were normalized to
the BY250 control, and representative analyses were averaged and
statistically analyzed by Prism and one-way ANOVA.

### Protein purification

The appendage domain of rat AP-3 delta was amplified by PCR and
cloned in-frame into pET15b-MBP-MCS. The recombinant proteins
were induced overnight at 20°C in *Escherichia coli* BL21 (DE3) using
0.1 mM IPTG. The bacteria were lysed by sonication in in 50 mM
Tris, 300 mM NaCl, pH 8.0, and MBP fusions purified by incubation
with amylose resin beads (NEB) and eluted with 50 mM maltose in
50 mM Tris, 300 mM NaCl, pH 8.0. Two days after infection with
baculoviruses produced according to the manufacturer's instruc-
tions (Invitrogen), Hi5 cells were harvested by centrifugation and
resuspended in binding buffer (50 mM Tris, 300 mM NaCl, pH 8.0)
with 1 mM PMSF and protease inhibitors (Complete, Roche). The
cells were lysed by sonication and pelleted for 20 min at 21,000 *g*
using a 4°C Eppendorf 5424 centrifuge. Supernatant was passed
through a 0.2 µm filter and incubated with Co-Nta agarose
(GoldBio) for 1 h at 4°C. The agarose was then washed five times in
binding buffer with 10–40 mM imidazole and His-hVPS41 eluted
with 500 mM imidazole. The eluates were immediately desalted on
a Secadex-5 column (Prometheus) into lysis buffer. 50 µg of MBP or
MBP-AP-3 were loaded onto amylose resin for 30 min at 4°C.
Unbound protein was removed by several washing steps, and vary-
ing amounts of VPS41 were incubated with the MBP resin to assess
interaction. Bound protein was eluted and evaluated by quantitative
fluorescence immunoblotting.

## Secretion assays

PC12 cells were plated on poly-L-lysine, washed, and incubated in Tyrode's buffer containing 2.5 mM K+ (basal) or 90 mM K+ (stimulated) for 30 min at 37°C. The supernatant was collected, cell lysates prepared as previously described (Asensio *et al*, 2010), and samples were analyzed by quantitative fluorescence immunoblotting.

## Statistics

Quantification of all Western blots was performed using Fiji software (Schindelin *et al*, 2012). Two-tailed Students *t*-tests were performed to compare two samples whereas ANOVA was performed for comparison of multiple samples to analyze statistical significance. Data distribution was assumed to be normal, but this was not formally tested. All error bars represent the Standard Error of the Mean (SEM) or Standard Deviation (SD) as indicated in the specific figure legends.

## Data availability

This study includes no data deposited in external repositories.

**Expanded View** for this article is available online.

## Acknowledgements

We are indebted to our colleagues of the Center for Molecular Medicine for fruitful discussions. We thank MCLS master student Bas van Zuijlen for the generation of the *VPS11* and *VPS18* knockout cell lines, Dr. Stephanie de Poot (Cell biology, UMC Utrecht) for kindly providing the APEX2 constructs and Prof. R. Roepman (Radboud UMC, Nijmegen, Netherlands) and Prof. J.J.C. Neefjes (Leiden UMC, Netherlands) for providing the GFP-pcDNA-Dest53 and pEGFP-C1-mCherry-VPS41 constructs, respectively. Prof. Catherine Rabouille (Hubrecht, The Netherlands) is acknowledged for her critical input throughout our studies. We thank Dr. I. Stolte-Dijkstra (UMC Groningen, Netherlands) for providing information and patient fibroblasts from the Dutch *VPS41* patient and Dr. H.E. Westerlaan (UMC Groningen, Netherlands) for providing the MRI data of this patient. Support for the *C. elegans* studies came from a grant from the National Institutes of Health [R15 NS075684-01 to G.A.C.]. N.L. is supported by a ZonMW TOP grant [40-00812-98-16006 to J.K]. P.S and RvdW are supported by a DFG grant [FOR2625 to J.K. as part of the Research Consortium]. The electron microscopy within this work is part of the research program National Roadmap for Large-Scale Research Infrastructure (NEMI) 2017 – 2018 with project number 184.034.014, which is (partly) financed by the Dutch Research Council (NWO). We are very grateful for the patients and their parents for their help and support.

## Author contributions

Experiments: RENW, CB, PS, CB, LC, FJZ, SZ, EFG, JAB, and TV; Clinical investigation on the patients: RJ, AF, CMAR-A, HHL, SB, CS, AML, GY, and DC; Sequencing analysis and variant identification: HHL, RP, and TS-S; Study design: RENW, CB, PS, JAB, NL, GAC, CSA, KAC, DC, and JK; Manuscript reviewing and revision: AF, FJZ, JAB, NL, CMAR-A, SB, CSA, GAC, KAC, and DC; Study design and supervision: CA, GAC, KAC, DC, and JK; Funding: JK; Manuscript writing: RENW, CMAR-A, SB, DC, and JK.

## Conflict of interest

The authors declare that they have no conflict of interest.

## For more information

- https://omim.org/entry/605485
- https://www.movementdisorders.org/
- https://www.cellbiology-utrecht.nl/research/klumperman.html
- https://www.lmp.utoronto.ca/faculty/david-chitayat
- https://www.michaeljfox.org/

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
