## [Review Process File · EMBO Molecular Medicine]

Neurodegenerative VPS41 variants inhibit HOPS function and mTORC1-dependent TFEB/TFE3 regulation

Reini van der Welle, Rebekah Jobling, Christian Burns, Paolo Sanza, Jan van der Beek, Alfonso Fasano, Lan Chen, Fried Zwartkruis, Susan Zwakenberg, Edward Griffin, Corlinda Brink, Tineke Veenendaal, Nalan Liv, Conny van Ravenswaaij-Arts, Henny Lemmink, Rolph Pfundt, Susan Blaser, Carolina Sepulveda, Andres Lozano, Grace Yoon, Teresa Santiago-Sim, Cedric Asensio, Guy A. Caldwell, Kim Caldwell, David Chitayat, and Judith Klumperman

DOI: 10.15252/emmm.202013258

Corresponding author(s): Judith Klumperman (J.Klumperman@umcutrecht.nl) , Reini van der Welle (R.E.N.vanderWelle@umcutrecht.nl), David Chitayat (David.Chitayat@sinaihealthsystem.ca)

Review Timeline:

Submission Date:	7th Aug 20
Editorial Decision:	8th Sep 20
Revision Received:	11th Jan 21
Editorial Decision:	1st Feb 21
Revision Received:	11th Feb 21
Editorial Decision:	18th Feb 21
Revision Received:	19th Feb 21
Accepted:	22nd Feb 21

Editor: Jingyi Hou

Transaction Report:

8th Sep 2020

Dear Dr. Klumperman,

Thank you for the submission of your manuscript to EMBO Molecular Medicine. We have now received feedback from the three referees whom we asked to evaluate your manuscript. As you will see from the reports below, the referees acknowledge the potential interest of the study. However, they also raise a series of concerns about your work, which should be convincingly addressed in a major revision of the present manuscript.

Without repeating all the points raised in the reviews below, some of the most substantial issues are the following:

- Referees pointed out that additional experiments are required to provide more in-depth mechanistic insights into the pathological role of VPS41 dysfunction, and to make the study more conclusive, which we would ask you to adequately address.
- Ref #2 mentioned a very recent publication (PMID: 32808683) related to the current study. While the other publication would not affect our evaluation of the novelty of the present manuscript due to the EMBO press "scooping protection policy", attention should be given to placing the current study in the context of existing literature.
- Ref #2 pointed out that it would be more informative to generate knock-in *C. elegans* mutants followed by a more in-depth phenotypic analysis, which we would encourage you to address. However, generating knock-in mutants is not mandatory for publication.
- The referees' concerns with regard to the clinical data and mutational analyses need to be carefully addressed.

We would welcome the submission of a revised version within three months for further consideration. Please note that EMBO Molecular Medicine strongly supports a single round of revision and that, as acceptance or rejection of the manuscript will depend on another round of review, your responses should be as complete as possible.

We are aware that many laboratories cannot function at full efficiency during the current COVID-19/SARS-CoV-2 pandemic and have therefore extended our "scooping protection policy" to cover the period required for a full revision to address the experimental issues. Please let me know should you need additional time, and also if you see a paper with related content published elsewhere.

I look forward to receiving your revised manuscript.

Sincerely,
Jingyi

Jingyi Hou
Editor
EMBO Molecular Medicine

*** Instructions to submit your revised manuscript ***

**** PLEASE NOTE **** As part of the EMBO Publications transparent editorial process initiative (see our Editorial at <https://www.embopress.org/doi/pdf/10.1002/emmm.201000094>), EMBO Molecular Medicine will publish online a Review Process File to accompany accepted manuscripts.

To submit your manuscript, please follow this link:

<https://embomolmed.msubmit.net/cgi-bin/main.plex>

- 1) a .docx formatted version of the manuscript text (including Figure legends and tables). Please make sure that the changes are highlighted to be clearly visible to referees and editors alike.
- 2) separate figure files*
- 3) supplemental information as Expanded View and/or Appendix. Please carefully check the authors guidelines for formatting Expanded view and Appendix figures and tables at <https://www.embopress.org/page/journal/17574684/authorguide#expandedview>
- 4) a letter INCLUDING the reviewers' reports and your detailed responses to their comments (as Word file)

Also, and to save some time should your paper be accepted, please read below for additional information regarding some features of our research articles:

- 5) The paper explained: EMBO Molecular Medicine articles are accompanied by a summary of the articles to emphasize the major findings in the paper and their medical implications for the non-specialist reader. Please provide a draft summary of your article highlighting

6) For more information: There is space at the end of each article to list relevant web links for further consultation by our readers. Could you identify some relevant ones and provide such information as well? Some examples are patient associations, relevant databases, OMIM/proteins/genes links, author's websites, etc...

7) Author contributions: the contribution of every author must be detailed in a separate section (before the acknowledgments).

8) EMBO Molecular Medicine now requires a complete author checklist (<https://www.embopress.org/page/journal/17574684/authorguide>) to be submitted with all revised manuscripts. Please use the checklist as a guideline for the sort of information we need WITHIN the manuscript as well as in the checklist. This is particularly important for animal reporting, antibody dilutions (missing) and exact p-values and n that should be indicated instead of a range.

9) Every published paper now includes a 'Synopsis' to further enhance discoverability. Synopses are displayed on the journal webpage and are freely accessible to all readers. They include a short stand first (maximum of 300 characters, including space) as well as 2-5 one sentence bullet points that summarise the paper. Please write the bullet points to summarise the key NEW findings. They should be designed to be complementary to the abstract - i.e. not repeat the same text. We encourage inclusion of key acronyms and quantitative information (maximum of 30 words / bullet point). Please use the passive voice. Please attach these in a separate file or send them by email, we will incorporate them accordingly.

You are also welcome to suggest a striking image or visual abstract to illustrate your article. If you do please provide a jpeg file 550 px-wide x 400-px high.

10) A Conflict of Interest statement should be provided in the main text

11) Please note that we now mandate that all corresponding authors list an ORCID digital identifier. This takes <90 seconds to complete. We encourage all authors to supply an ORCID identifier, which will be linked to their name for unambiguous name identification.

Currently, our records indicate that there is no ORCID associated with your account.

Please click the link below to provide an ORCID:
Link Not Available

12) The system will prompt you to fill in your funding and payment information. This will allow Wiley to send you a quote for the article processing charge (APC) in case of acceptance. This quote takes into account any reduction or fee waivers that you may be eligible for. Authors do not need to pay any fees before their manuscript is accepted and transferred to our publisher.

Photos 400-800 DPI

*Additional important information regarding figures and illustrations can be found at <http://bit.ly/EMBOPressFigurePreparationGuideline>

***** Reviewer's comments *****

Referee #1 (Comments on Novelty/Model System for Author):

This is a very interesting and exciting manuscript identifying and analyzing the first patients with hereditary mutations in the VPS41 gene. This manuscript meets the aims of EMBO Mol Med and the high interest of the Cell Biology/Trafficking community in the endosome/autophagosome fusion with lysosomes as well the scientists community interested in neurodegenerative diseases will bring a strong attention to this manuscript. However, the manuscript needs some improvements which should be easily practicable for the authors.

The applied imaging techniques and in particular the double immunogold EM images are brilliant.

Referee #1 (Remarks for Author):

In addition to its role as subunit of the HOPS tethering complex which mediates the fusion between lysosomes and late endosomes and autophagosomes, the VPS41 protein has HOPS-independent functions in the transport of lysosomal membrane proteins, regulated secretion of neuropeptides, and protection of neurodegeneration in a *C. elegans* model of Parkinson disease, as previous studies from various laboratories have shown. In this manuscript van der Welle et al. report on the identification of the first compound heterozygous VPS41 patients characterized among others by mental retardation, cerebellar atrophy, retinal dystrophy, ataxia, and kyphosis. To gain insight into the molecular pathomechanisms of this novel phenotype, the authors analyzed fibroblasts from the patients and their parents, as well as rescue experiments using wildtype and mutant VPS41 in VPS41-ko HeLa, PC12 cells, and in the *C. elegans* model of PD. The data confirmed that in patient cells the HOPS complex is dysfunctional and prolongs the kinetics of cargo along the endocytic and autophagic pathways. This is associated with dissociation from lysosomes and inactivation of mTORC1 leading subsequently to translocation of the transcriptional activator of lysosomal/autophagy genes, TFE3, into the nucleus.

This is an important and well performed study combining a wide range of cell biological and biochemical approaches including high quality immunofluorescence and immunogold electron microscopy.

The manuscript lacks some information on the ko cells generated, careful proof-reading, and few additional experiments to confirm some of the statements.

Main points:

The VPS41 ko in PC12 cells should be documented by Western blotting, and information on the mutation generated by CRISPR/Cas in the used HeLa and PC12 clones have to be provided.

Fig. 2 and text p8: what is the explanation for the higher number of small size lysosomes in VPS defective fibroblasts?

p8, 3rd paragraph: the normal cathepsin B activity and presence of Lamp1 are not sufficient markers for 'small-sized, enzymatically active lysosomes'. Additional specific activities of lysosomal enzymes should be measured

Fig. 3a and text p8 last paragraph: the text does not fit at all with the western blot: no independent control (healthy patient's fibroblasts) is shown (and should be included); the VPS42S285P/R662* shows 70% of VPS41-WT/R662* signal (instead of complete loss) VPS41 H466R/R662* is shown in the quantification but not in the blot, etc.

p9: delete the sentence: The substitution of serine to proline in VPS41S285P is predicted to induce a defect in protein folding (<http://genetics.bwh.harvard.edu/pph2/>)42. This prediction program failed in many mutational analyses, and for your H466R mutation as well

Not the H466R mutation results in 10% of VPS41-WT/R662*, but the H466R/R662*. Since the overexpressed R662* mutant appears stable (Fig. 3b), quantitative RT-PCR should be performed to analyze mRNA decay/stability in patients fibroblasts

P10: does the re-expression of VPS41 mutants lead to increased number of small size lysosomes? How much is the -fold increase in mRNA level of WT and mutant VPS41 in comparison with the endogenous VPS41 mRNA level in HeLa cells?

p15: the message of Fig. 6d is unclear for me

Minor points

p3, line 4: the VPS41 S285 mutation is incomplete

p4, 2nd paragraph- define the chromosomal localization of Chr7p14.1

p10: endocytosis defect of dextran 568 in SirLyso-loaded patient fibroblasts: did you used during the complete study the same passages of different patient cells which might be responsible for the unexpected defect in maternal VPS41-WT/R662* cells?

Suppl. Fig.S6a: I wonder about the low colocalization of endocytosed Dextran

568 after 2 h with SirLyso-labeled lysosomes even in VPS41WT-APEX2-V5 rescued HeLaVPS41 ko cells?

p13, last line: Fig. S9a,b should be Fig. 8a,b-please provide a representative blot

consequently p14: (Fig. S10a and S10b, quantified in S10b') should be Fig. S9a, b, b' and in the 2nd paragraph S10c should be Fig. S9c

Referee #2 (Remarks for Author):

In this manuscript the authors report the identification of mutations of the VPS41 gene, a component of the HOPS complex, in three patients with a neurodegenerative phenotype characterized by intellectual disability, ataxia and dystonia, thus linking HOPs dysfunction to neurodegeneration. Next, the authors analyzed the cellular phenotype of patient-derived cell lines and performed biochemical and molecular analyses, with the aim of identifying the mechanism by

which loss-of-function of VPS41 results in the patients' phenotype.

The link between HOPS loss-of-function and neurodegeneration is potentially novel and interesting. However, a very recent publication (see below) reporting the identification of mutations in several components of the HOPS complex, including VPS41, in patients with dystonia and lysosomal abnormalities diminishes, at least partially, the novelty of the present manuscript. For this reason the mechanistic aspects become very important as these would significantly increase the interest and novelty of the present paper. Unfortunately, most of the mechanistic studies performed in patients' cell lines and/or in transfected and gene-edited cell lines are superficial and inconclusive, thus making it difficult to determine the pathological relevance of the authors' observations. Much more in depth cell biology and biochemical studies are needed to determine the disease-relevant aspects of VPS41 dysfunction. Finally, the part on *C.Elegans* does not add much to the manuscript because it was performed by overexpression of the VPS41 mutants. It would have been more informative to generate knock-in mutants followed by in depth phenotype analysis. In conclusion, in my opinion this paper reports many potentially interesting observations but none of them was analyzed in depth. One possibility to improve the paper would be to focus on fewer aspects and generate more conclusive and disease-relevant data.

CLINICAL DATA AND MUTATION ANALYSES

In general, I found this part incomplete. Specific points below:

1. Given the recent publication of several cases with pathogenic variants in the HOPS complex (PMID: 32808683), the authors need to compare the clinical findings and the degree of clinical overlap of their cases with those from this very recent publication. For instance the facial phenotype of patient 3 of the present manuscript is very similar to the one of patients with specific types of lysosomal storage diseases (e.g. mucopolysaccharidoses). This is potentially very interesting but the authors did not even comment on this aspect.
2. It would be helpful to show the pedigrees of the two families. Are there unaffected siblings in both families who can be evaluated for the presence of the VPS41 variants?
3. For each VPS41 variant, gnomAD frequency and prediction of pathogenicity by multiple available software should be presented, possibly in a table format.
4. A list of rare variants found by exome sequencing would be important instead of the supplementary table 1 on the targeted gene analysis that is not informative.
5. The reason for discarding fibroblasts of patient 2 and 3 from the analyses is confusing. Are the 1q21.1 dup and the UPF3A variants considered pathogenic? If they are not, why not including these cells in the analyses to confirm the findings observed in cells from patient 1? Are GAG levels or activities of lysosomal enzymes affected in fibroblasts?
6. The clinical description of all three cases should be more consistent and should include in all three cases: growth parameters (especially head circumference over time), developmental milestones, and cognitive development. Do patients 1 and 2 have liver and spleen enlargement, retinal dystrophy, or skeletal abnormalities like case 3? Was an echocardiogram performed?
7. The genomic coordinate of the 1q21.1 duplication should be included. Does this region overlap with the 1q21.1 duplication syndrome?
8. The hand X-ray of case 3 is interesting because it shows metacarpal pointing, a skeletal finding often seen in lysosomal storage disorders.
9. The metabolic studies performed need to be specified. Were GAG levels measured in the patients? Were lysosomal enzyme activities measured?

LYSOSOMAL/AUTOPHAGIC PHENOTYPE

It is unclear how the lysosomal-autophagic compartments are affected by VPS41 dysfunction. It is equally unclear how the cellular phenotype is an indication of a disease-relevant dysfunction.

Specific points below:

10. Immuno-EM analyses in Figure 1 show that cells carrying VPS41 mutants display smaller lysosomes. This analysis was performed on lysosomes labeled with BSA, which, however, the authors then show to only label a subpopulation of lysosomes, as endocytosis is defective in VPS41-deficient cells (Figure 3). What is the size of BSA-negative lysosomes in VPS41-mutated cells? Most importantly, why are lysosomes smaller and is this decrease in size relevant to the disease phenotype (i.e. are lysosomes dysfunctional?).

11. Analysis of the autophagic flux requires further investigation. The defective increase of LC3-II in patient fibroblasts indicates lack of autophagy induction, rather than lack of fusion. Accordingly, if newly synthesized autophagosomes are not able to fuse with lysosomes, there should be a higher increase of LC3-II in patient cells compared to controls, whereas the data indicate the opposite. In order to properly evaluate this point and discriminate induction from fusion, the experiments performed in Figure 4d using GFP-RFP-LC3 should be performed not only in VPS41-deficient cells but also in cells reconstituted with VPS41-mutants. In addition, the authors should add treatment with bafilomycin to better follow the autophagy defect and also IF of LC3 on fibroblasts. Why are LC3 levels significantly higher in the VPS41 KO HeLa compared to patients' fibroblasts?

12. Co-ip experiments in Figure 3b show a weak interaction of VPS41 only with one overexpressed member of the HOPS complex (VPS11). To assess whether VPS41 mutants correctly assemble into the HOPS complex, endogenous components of the complex (VPS18, VPS11, VPS33A) should be assessed for their interaction with WT and mutant VPS41 (as previously described in manuscript PMID: 28931724) transfected in VPS41-KO cells.

mTORC1 SIGNALING AND TFE3/TFEB

The role of VPS41 in the modulation of mTORC1 signaling and regulation of TFE3/TFEB function is unclear. Based on the data presented, the disease-relevance of the observed TFE3 nuclear translocation remains questionable. What is the consequence of TFE3 activation? Also, what was the rationale of analyzing TFE3 and not TFEB?

Some specific points:

13. What is the mechanism by which VPS41 loss-of-function impairs mTORC1 lysosomal localization? Why would this effect on mTORC1 affect only TFE3 phosphorylation (and likely TFEB) and does not affect S6K and 4E-BP1 phosphorylation? The data on mTORC1 activity (only Fig. 5f!!!) are too limited and not convincing.

14. Fig 5A: Analysis of mTORC1 lysosomal localization is puzzling. How was quantification of mTOR localization at the lysosome performed? How many cells? The authors claim that mTOR lysosomal localization is impaired in patient fibroblasts compared to control cells. However, Figure 5a' shows a strong impairment in mTOR lysosomal localization in WT/R662* and WT/S285P fibroblasts in both basal and re-feeding conditions, in which the levels of mTOR-LAMP1 colocalization appear even lower than in starved control cells. Thus, no conclusions can be made on mTORC1 lysosomal localization based on fibroblast. To this end, the same analysis should be done on VPS41-KO HeLa cells reconstituted with WT and mutant VPS41.

15. How was the "% nuclei positive for TFE3" calculated? In the images in Figure 5d, VPS41-KO cells reconstituted with WT-VPS41 show reduced nuclear TFE3 with no increase in cytosolic signal, which does not seem a "rescue" to WT cells. Is it possible that the APEX-V5 tag on VPS41 affects its function? These and other key experiments of the paper should be reproduced with untagged versions of WT and mutant VPS41. Finally, a more extensive and unbiased analysis of TFE3/TFEB targets should be analyzed by RNA-seq analysis.

16. Fig 5B: The authors should check the phosphorylation status of both TFE3 and TFEB either by

using specific phospho-antibodies or by following molecular weight shift in presence and absence of mTOR inhibitors.

17. Fig S9 in the text actually corresponds to S8. S10 in the text is actually S9.

Referee #3 (Comments on Novelty/Model System for Author):

Although it would have been preferable to examine more mammalian neuronal models, the rise of cell lines to address basic questions of mutant VPS41 function was adequate.

Referee #3 (Remarks for Author):

This manuscript presents the identification of rare human VPS41 mutations as the basis for a complex neurodevelopmental/neurodegenerative disorder. The functionality of the patient mutations are further investigated in human cell lines and a *C. elegans* model of dopaminergic neuron degeneration. These efforts generally support a loss of function model wherein the mutant VPS41 can no longer fully support function of the HOPS complex in which it resides. This has consequences on a broad range of lysosome properties. Such changes are consistent with a well known role for the HOPS complex in supporting fusion between endosomes and lysosomes. Collectively this manuscript identifies new human mutations in VPS41 as a cause of a neurological disorder and establishes that this is due to a loss-of-function mechanism via a variety of cell biological assays. The experiments are performed to a high standard of quality and the conclusions drawn are generally reasonable. The concerns that I outline below are relatively modest and should be readily addressable.

1. The title claims defects in mTORC1 signaling that are not supported by the data that is presented in Figure 5E which shows normal activation of mTORC1 signaling by amino acids in Figure 5F. This is the only piece of data in the manuscript that examines mTORC1 signaling and it contradicts the title. My suggestion is that the title should better represent the data.
2. Can the authors better explain the basis for defining the human condition arising from VPS41 mutations as neurodegenerative rather than neurodevelopmental? Although the *C. elegans* model is interesting, it does not establish that neurodegeneration is happening in humans with these mutations.
3. Speaking of LC-3 lipidation (page 12), the authors state that: "Notably, the autophagy defect is specific for the patient derived fibroblasts and not observed in any of the parental cell lines, indicative that this phenomenon is important for disease development." However, no evidence was presented to indicate a cause and effect relationship between defective autophagy and disease development.
4. Key mTORC1 signaling data in Figure 5F should be quantified. Ideally, the mTORC1 phosphorylation site on ULK1 should also be examined as it is a major link between mTORC1 and autophagy.
5. The basis for the following concluding statement from page 20 is not clear: "In summary, our data suggest that the VPS41 patients may represent a novel class of lysosomal disorders in which lysosomes are enzymatically active but due to delayed trafficking kinetics are insufficiently reached by nutrients." It is not clear how the authors arrived at this conclusion or which nutrients this refers

to. Although defects were noted in the subcellular localization of TFE3 and its regulation by changes in amino acid availability, mTOR signaling as presented in Figure 5E responded normal to the same stimuli. This suggests that the defect may be more narrowly related to TFE3 regulation rather than nutrient sensing mechanisms that are shared by TFE3 and mTORC1. It may not be necessary to perform further experiments to address this matter if the discussion can be modified to address this concern.

6. In addition to reference 60 (page 19), the first demonstration of TFE3 translocation to the nucleus in response to chloroquine was in the following study: PMID: 22692423

7. Author names are missing from some references such as #21 and #30.

Dear Jingyi Hou,

Thank you for sending us the reviewer comments on our manuscript, which in this revision on request of reviewer 3 is renamed to "Neurodegenerative *VPS41* variants inhibit HOPS function and mTORC1-dependent TFEB/TFE3 regulation". We hereby submit the revised version, which is substantially modified along the comments of the reviewers. We have performed many new experiments, added new figures, made textual changes and added new literature references. We have succeeded to address the vast majority of comments and are pleased to present you this improved manuscript. We hope you now find our work acceptable for publication in EMBO Molecular Medicine.

Sincerely,
Judith Klumperman

Overview of Figure changes:

Altered Figures:

- Figure 3A, A'
- Figure 5A'

New figures:

- Figure 4A, A', B, B'
- Figure 5E'
- Figure EV1A, B, D
- Figure EV3A, B, C
- Figure EV4C, D, F
- Figure EV5A, A', D
- Appendix Figure S3A, B
- Appendix Figure S4B
- Appendix Figure S7A, B
- Appendix Figure S8A, B, C, C'
- Appendix Figure S9A, B

For the reviewers' convenience, we added the new figures to the rebuttal letter below.

Comments from the editor

Ref #2 mentioned a very recent publication (PMID: 32808683) related to the current study. While the other publication would not affect our evaluation of the novelty of the present manuscript due to the EMBO press "scooping protection policy", attention should be given to placing the current study in the context of existing literature.

This recent study by Steel et al., (2020) was published when our paper was under review. It describes 1 *VPS41* patient, showing an overlapping phenotype with the patients described in our study. The paper does not provide insightful information on the cellular effects of *VPS41* deletion, as we extensively do in our paper. We discuss the paper on page 5, 21 and 22 and included it in the reference list.

Ref #2 pointed out that it would be more informative to generate knock-in *C. elegans* mutants followed by a more in-depth phenotypic analysis, which we would encourage you to address. However, generating knock-in mutants is not mandatory for publication.

We thank you for this considerate remark.

The referees' concerns with regards to the clinical data and mutational analyses need to be carefully addressed.

We have extended the description of exome sequencing analysis, added additional clinical data and wrote the clinical representation in a consistent manner to better compare the three presented patients, thereby addressing all concerns raised by the reviewers.

Of note: During the revision we came across an error in *VPS41* variant analysis of the third patient. The correct identified mutations are as follows:

Paternal variant in the TPR-like domain [NM_014396.3:c.1423-2A>G p.?] and a maternal nonsense variant in the CHCR domain [c.1984C>T,NP_055211.2:pArg662Stop(R662*)] The c.1423-2A>G variant is a splice site variant that destroys the canonical splice acceptor site in intron 17 and is predicted to cause abnormal gene splicing. The R662* nonsense variant is shared between all three patients and results in a truncated protein.

The third patient will be addressed as *VPS41*^{1432-2A>G/R662*}.

Reviewer's comments

Referee #1 (Comments on Novelty/Model System for Author):

This is a very interesting and exciting manuscript identifying and analyzing the first patients with hereditary mutations in the *VPS41* gene. This manuscript meets the aims of EMBO Mol Med and the high interest of the Cell Biology/Trafficking community in the endosome/autophagosome fusion with lysosomes as well the scientists community interested in neurodegenerative diseases will bring a strong attention to this manuscript. However, the manuscript needs some improvements which should be easily practicable for the authors. The applied imaging techniques and in particular the double immunogold EM images are brilliant.

We kindly thank the reviewer for the compliment and positive evaluation of our work.

Referee #1 (Remarks for Author):

In addition to its role as subunit of the HOPS tethering complex which mediates the fusion between lysosomes and late endosomes and autophagosomes, the *VPS41* protein has HOPS-independent functions in the transport of lysosomal membrane proteins, regulated secretion of neuropeptides, and protection of neurodegeneration in a *C. elegans* model of Parkinson disease, as previous studies from various laboratories have shown. In this manuscript van der Welle et al. report on the identification of the first compound heterozygous *VPS41* patients characterized among others by mental retardation, cerebellar atrophy, retinal dystrophy, ataxia, and kyphosis. To gain insight into the molecular pathomechanisms of this novel phenotype, the authors analyzed fibroblasts from the patients and their parents, as well as rescue experiments using wildtype and mutant *VPS41* in *VPS41*-ko HeLa, PC12 cells, and in the *C.elegans* model of PD. The data confirmed that in patient cells the HOPS complex is dysfunctional and prolongs the kinetics of cargo along the endocytic and autophagic pathways. This is associated with dissociation from lysosomes and inactivation of mTORC1 leading subsequently to translocation of the transcriptional activator of lysosomal/autophagy genes, TFE3, into the nucleus. This is an important and well performed study combining a wide range of cell biological and biochemical approaches including high quality immunofluorescence and immunogold electron microscopy.

We again thank the reviewer for the appreciation of our work

The manuscript lacks some information on the ko cells generated, careful proof-reading, and few additional experiments to confirm some of the statements.

We apologize for these flaws. We have tried to address all inconveniences of this kind to the best of our abilities.

Main points:

1. The VPS41 ko in PC12 cells should be documented by Western blotting, and information on the mutation generated by CRISPR/Cas in the used HeLa and PC12 clones have to be provided.

We have added the information on generation of CRISPR/Cas HeLa and PC12 knock-out lines to the Material and Methods section (p28). The western blots validating depletion of *VPS41* in PC12 KO cells and of *VPS11* or *VPS18* in HeLa KO cells is added to the Supplemental figures. Since there are no antibodies available to validate depletion of *VPS39* on western blot we used TIDE detection to confirm knock-out of *VPS39* in HeLa cells (**New Appendix Figure S7A, B and Appendix Figure S9A, B respectively; see also below**). In short, these figures show the predicted site for Cas9-mediated DNA cleavage, resulting in a full KO of *VPS41* in PC12 cells (S7A, B), and a confirmed depletion of *VPS11* and *VPS18* on western blot (S9A). The presented indel spectrum and TIDE detection confirms deletion in *VPS39* (S9B). The western blot confirming *VPS41* depletion in HeLa cells was already shown in the original paper (Fig. 2B).

2. Fig. 2 and text p8: what is the explanation for the higher number of small size lysosomes in *VPS41* defective fibroblasts?

Lysosomal size is controlled by a diversity of pathways (see recent review by Araujo et al., (2020), now added to the paper), the combination of which determines their actual size. Within this complexity, a plausible explanation for the small lysosomes in *VPS41*-deficient cells is an increased lysosomal biogenesis by constitutive activation of TFE3/TFEB (as described in Settembre et al., (2011) and Martina et al., (2014)), combined with the HOPS dependent defect in endo-lysosomal fusion (providing less membrane). We have extended our statements and conclusions on these findings in the discussion (p23).

3. p8, 3rd paragraph: the normal cathepsin B activity and presence of Lamp1 are not sufficient markers for 'small-sized, enzymatically active lysosomes'. Additional specific activities of lysosomal enzymes should be measured.

We did more assays than the reviewer indicates here, using a combination of microscopy and biochemistry methods. In addition to monitoring the presence of Cathepsin B and LAMP-1 we used Activity-based probe MagicRed-Cathepsin B, Cathepsin D fluorescence microscopy, Lyso tracker (to define acidic compartments), Endocytic tracers (Dextran, BSA gold), and (immuno)-electron microscopy. By immuno-EM, we defined the LAMP-1-positive lysosomes and show that they are in reach of endocytic markers, albeit with slowed-down kinetics due to the absence of VPS41. By EM, we show that cells lack a typical lysosomal storage phenotype indicative for enzyme dysfunction. Concomitantly, as mentioned at page 6 and 8, metabolic diagnostics of all patients did not reveal any aberrancies. In addition to these assays, but not added to the paper, we performed a western blot on GBA levels of patient 2. GBA is a lysosomal enzyme that is transported independent of mannose 6-phosphate receptor and a causative gene for Gaucher disease. This showed a slight increase in activity in fibroblasts, consistent with an upregulation of lysosomal activity. For the reviewer's convenience we show these data below (**See Figure below**). None of these approaches, nor previous studies in human or fly cells depleted for *VPS41*, point to a defect in lysosomal enzyme transport, processing or activity.

4. Fig. 3a and text p8 last paragraph: the text does not fit at all with the western blot: no independent control (healthy patient's fibroblasts) is shown (and should be included);

We sincerely apologize for the inconsistencies in this figure. We have included the independent control (*VPS41^{WT/WT}*) to the Western Blot (**revised Figure 3A, A'**; see also **below**).

The VPS41-S285P/R662* shows 70% of VPS41-WT/R662* signal (instead of complete loss). We do not expect a complete loss of VPS41^{S285P/R662*}, since the VPS41^{S285P} variant is expressed and the protein is not degraded. The complete loss we are referring to concerns the R662* variant, which is not detectable on western blot.

VPS41 H466R/R662* is shown in the quantification but not in the blot, etc. Actually, the c.1423-2A>G variant is visible on the blot, but admittedly very vaguely. We added an extra panel showing longer exposure times, to better visualize this band (**revised Figure 3A, A' and Appendix Figure S3A, see also below**).

5. p9: delete the sentence: The substitution of serine to proline in VPS41S285P is predicted to induce a defect in protein folding (<http://genetics.bwh.harvard.edu/pph2/>)42. This prediction program failed in many mutational analyses, and for your H466R mutation as well We deleted the sentence. We provide a statement on protein characteristics (e.g. differences and similarities between the substituted amino acids and possible consequences for protein functionality) in the discussion (p22).

6. Not the H466R mutation results in 10% of VPS41-WT/R662*, but the H466R/R662*. Indeed, patient 3 expresses only $\pm 10\%$ of the VPS41 levels seen in VPS41^{WT/WT}. Since the R662* mutation is not expressed at all, the only VPS41 variant present in these cells is VPS41^{1423-2A>G}. As far as we can see we indicated this rightfully in the paper. The western blot presented in Figure 3A illustrates that the R662* mutant is not expressed. Treatment with MG132 did not result in a visible band of the R662* mutant, indicating that the lack of expression is not caused by proteasomal degradation (**revised Figure 3A, A' and Appendix Figure S3A**).

Since the overexpressed R662* mutant appears stable (Fig. 3b), quantitative RT-PCR should be performed to analyze mRNA decay/stability in patient fibroblasts

We performed the requested quantitative RT-PCR experiment on fibroblasts of patient 2 and his parents. Indeed, this shows that both the maternal (WT/R662*) and the patient (S285P/R662*) have decreased mRNA levels of VPS41, which is indicative of Nonsense-mediated decay. These new data are included in the manuscript (**New Appendix FigureS3B, see also Figure below**) and described on page 11.

7. P10: does the re-expression of VPS41 mutants lead to increased number of small size lysosomes?

To better understand this phenotype, we performed additional experiments in HeLa cells knockout for HOPS components VPS11, VPS18 and VPS39. This showed that KO of either of these 3 HOPS subunits lead to a similar increase in lysosome number as KO of VPS41, thus establishing this as a HOPS dependent phenotype (**new Figure EV5A, A', see also**

below). Since HOPS function is maximally compromised in *VPS41*^{KO} cells, this will not further increase by re-expression of the *VPS41* mutants, which lack HOPS function. In agreement herewith, we showed that reintroducing *VPS41*^{WT} rescues the TFE3 phenotype, whereas the mutants do not (Figure 5D, D'). We edited the text on this on p18.

8. How much is the -fold increase in mRNA level of WT and mutant *VPS41* in comparison with the endogenous *VPS41* mRNA level in HeLa cells?

In this study we use a transient expression system to express the *VPS41* variants in HeLa KO cells, which on average leads to transfection of 50% of the cells. Consequently, we cannot use RT-PCR to quantify the expression levels of mutant and WT *VPS41* mRNA. It is likely that these rescue experiments result in expression levels higher than endogenous *VPS41*. To minimize the effects of expression levels, for our IF experiments we selected cells with equal signal levels of WT and mutant *VPS41*. Hence, these quantitative imaging studies were done in cells with comparable expression levels.

9. p15: the message of Fig. 6d is unclear for me

We have rewritten this text as follows: "As previously shown, depletion of *VPS41* from PC12 cells resulted in decreased cellular SgII protein levels, as well as in a reduction in the regulated secretion of the proteins that are present (Asensio et al, 2013). Reintroducing *VPS41*^{WT} in PC12 *VPS41*^{KO} cells rescued cellular SgII levels and recovered regulated secretion (Fig 6C, D). Interestingly, similar results were obtained by expression of the *VPS41*^{S285P} variant (Fig 6C, D), indicating that its role in regulated secretion has been preserved." (p19)

Minor points

p3, line 4: the *VPS41* S285 mutation is incomplete

Thank you for notifying us, this is corrected.

p4, 2nd paragraph- define the chromosomal localization of Chr7p14.1

As far as we know, this is the correct way of indicating the chromosomal location of *VPS41*. We have specified this in the text as: "(*VPS41* is found on Chromosomal location Chr7p14.1)" (p4).

p10: endocytosis defect of dextran 568 in SirLyso-loaded patient fibroblasts: did you use during the complete study the same passages of different patient cells which might be responsible for the unexpected defect in maternal *VPS41*-WT/R662* cells?

This is an important point. Indeed, we took care to avoid differences between cells due to culture conditions. All cell cultures were started at the same time and throughout our experiments comparable passages were used. Moreover, experimental replicates were performed on different passage numbers. We never used cells beyond passage 20.

Suppl. Fig.S6a: I wonder about the low colocalization of endocytosed Dextran568 after 2h with SirLyso-labeled lysosomes even in VPS41^{WT}-APEX2-V5 rescued HeLaVPS41 ko cells? After 2 hours, 60% of SirLyso-labeled lysosomes are reached by Dextran-568 in HeLa VPS41^{WT} rescued cells. These levels are comparable to colocalization levels in WT/WT fibroblasts (visualized in Fig. 3E and F). Extended uptake (up to 24 hours, see Fig. 3H) increases localization levels to 80%. Similar percentages are described in other papers (Bright et al., (2016) and Pols et al., (2013)). These data suggest that cells contain a population of lysosomes that is not reachable by endocytic markers.

p13, last line: Fig. S9a,b should be Fig. 8a,b-please provide a representative blot consequently p14: (Fig. S10a and S10b, quantified in S10b') should be Fig. S9a, b, b' and in the 2nd paragraph S10c should be Fig. S9c

We apologize for this mistake. We have renumbered the figures and added the requested western blots showing increased LAMP-1 and Cathepsin B protein levels in patient fibroblasts, as well as additional western blots for LAMP-1 and Cathepsin D in VPS41^{KO} cells (**New Appendix Figure S7, see also below**).

Referee #2 (Remarks for Author):

In this manuscript the authors report the identification of mutations of the VPS41 gene, a component of the HOPS complex, in three patients with a neurodegenerative phenotype characterized by intellectual disability, ataxia and dystonia, thus linking HOPs dysfunction to neurodegeneration. Next, the authors analyzed the cellular phenotype of patient-derived cell lines and performed biochemical and molecular analyses, with the aim of identifying the

mechanism by which loss-of-function of VPS41 results in the patients' phenotype. The link between HOPS loss-of-function and neurodegeneration is potentially novel and interesting. However, a very recent publication (see below) reporting the identification of mutations in several components of the HOPS complex, including VPS41, in patients with dystonia and lysosomal abnormalities diminishes, at least partially, the novelty of the present manuscript. For this reason the mechanistic aspects become very important as these would significantly increase the interest and novelty of the present paper. Unfortunately, most of the mechanistic studies performed in patients' cell lines and/or in transfected and gene-edited cell lines are superficial and inconclusive, thus making it difficult to determine the pathological relevance of the authors' observations. Much more in depth cell biology and biochemical studies are needed to determine the disease-relevant aspects of VPS41 dysfunction.

We agree with the reviewer that finding a direct link between a lysosomal lesion and a disease phenotype would have been highly interesting. However, given the complexity of lysosomal functioning in both metabolic and signaling pathways, which is also strongly dependent on cell type, a direct link between genotype and phenotype is unknown for many lysosomal disorders, even after decades of research. What we do show here is that *VPS41* mutations result in an entirely different cellular phenotype as observed in classical lysosomal disorders, with a profound and unexpected effect on mTORC1 signaling, which is a potential target for therapy.

Finally, the part on *C. Elegans* does not add much to the manuscript because it was performed by overexpression of the VPS41 mutants. It would have been more informative to generate knock-in mutants followed by in depth phenotype analysis. In conclusion, in my opinion this paper reports many potentially interesting observations but none of them was analyzed in depth. One possibility to improve the paper would be to focus on fewer aspects and generate more conclusive and disease-relevant data.

Though we acknowledge the added value of generating knock-in mutants in *C. elegans*, we believe this is not feasible for two reasons:

1) There is a worm nonsense mutation in *vps-41(ep402)* that truncates the clathrin heavy chain and RING domain (Lackner et al., (2005)). These animals are quite unhealthy; they have maternal-effect embryonic lethality and the animals that do live have a partial larval arrest phenotype; we cannot work with these animals because very few survive to adulthood. Since the R662* mutant also results in a truncation of the clathrin heavy chain and RING domain, it is expected that a R662* knock-in would result in a similar phenotype to this *vps-41(ep402)* mutant.

2) creation of knock-ins, outcrossing, genetic crossing to strains for analysis (alpha-syn, etc.) and then the subsequent phenotypic analysis would take at least 6 months. This is outside of the timeline provided by the journal for resubmission (3 months). Given that it is quite risky to think a R662* knock-in will actually survive and thrive, we are inclined to accept the offer from the editor for not performing this reviewer experimental suggestion.

CLINICAL DATA AND MUTATION ANALYSES

In general, I found this part incomplete. Specific points below:

1. Given the recent publication of several cases with pathogenic variants in the HOPS complex (PMID: 32808683), the authors need to compare the clinical findings and the degree of clinical overlap of their cases with those from this very recent publication. For instance the facial phenotype of patient 3 of the present manuscript is very similar to the one of patients with specific types of lysosomal storage diseases (e.g. mucopolysaccharidoses). This is potentially very interesting but the authors did not even comment on this aspect.

The manuscript PMID: 32808683 of Steel et al., reporting the first patient with early onset dystonia due to a homozygous *VPS41* splice site variant (NM_014396.3c.450+1G>T), was published after the submission of our paper. We now discuss our findings in relation to this paper (p21, 22). In short, the patient described by Steel et al. presented with global developmental delay, severe infant-onset dystonia and pale optic disks. He could speak a few words, but never sat unsupported. The disorder progressed from the age of 6 years

onwards, and cerebellar atrophy and slimming of the corpus callosum were noted on brain MRI. No further clinical details are given. The 3 *VPS41* mutated patients described in our study share features very similar to this reported patient, namely the autosomal recessive inheritance, the early onset of generalized dystonia with developmental delay, and the brain MRI characterized by thinning of the corpus callosum and atrophy of the cerebellar vermis. In addition, one of our patients (Patient 1) presents a slightly different phenotype, characterized by a less severe intellectual disability and ataxia and absence of overt dystonia, clearly indicating that the phenotypic expression of *VPS41* related disorders is variable. Concluding, our data confirm the findings in this recent publication and extend these by providing cellular insight in patient derived fibroblasts as well as *VPS41* depleted cells. Collectively the data indicate *VPS41* patients may represent a novel class of lysosomal disorders.

2. It would be helpful to show the pedigrees of the two families. Are there unaffected siblings in both families who can be evaluated for the presence of the *VPS41* variants?

We have added the pedigrees of the two families to the manuscript (**New Figure EV1A, D, see also Figure below**). Indeed, there is an unaffected sibling in the second family (sibling of patient 3), who is heterozygous for *VPS41*^{WT/R662*} (IX:1).

3. For each *VPS41* variant, gnomAD frequency and prediction of pathogenicity by multiple available software should be presented, possibly in a table format.

We have added the requested table (**New Table 1, see also Table below**)

VPS41 (NM_014396.3)	gnomAD global	SIFT	CADD	PolyPhen2 HDIV	PolyPhen2 HVAR
c.853 T>C p.S285P	0.0000071; 2/282128	0.048 damaging	23.5	0.956 possibly damaging	0.361 benign
c.1984 C>T p.R662*	0.00035; 98/282180	(b)	40	(b)	(b)
c.1423-2A>G p.? (a)	0.000032; 1/31398	(b)	34	(b)	(b)

Frequency of the specific *VPS41* variants in the general population based on the [gnomad](http://gnomad.broadinstitute.org/) database ([variants observed]/[total of individuals studied]) (<http://gnomad.broadinstitute.org/>) and prediction of pathogenicity based on the programs SIFT, (<https://sift.bii.a-star.edu.sg/>), CADD (<https://cadd.gs.washington.edu/>) and PolyPhen (<http://genetics.bwh.harvard.edu/pph2/>)

(a): Three different splice site analysis programs (MaxEnt, NNSPLICE and SSF) predict a total loss of wild type acceptor splice site (Interactive Biosoftware - Created by Alamut Visual v.2.15.0)

(b): not determined because SIFT and PolyPhen2 programs can only be used for analysis of missense mutations

4. A list of rare variants found by exome sequencing would be important instead of the supplementary table 1 on the targeted gene analysis that is not informative.

We have added the requested lists of identified rare variants for all three patients as supplementary files; **New Dataset EV1 and Dataset EV2**.

5. The reason for discarding fibroblasts of patient 2 (*this should be patient 1, RvdW and JK*) and 3 from the analyses is confusing. Are the 1q21.1 dup and the *UPF3A* variants considered pathogenic? If they are not, why not including these cells in the analyses to confirm the findings observed in cells from patient 1?

We added the requested information on the 1q21.1 duplication and the *UPF3A* variants to the manuscript (p7, 10). In summary, several studies have shown that a 1q21.1 duplication can cause developmental delay and intellectual disabilities, indicated as 1q21.1 syndrome (e.g. Brunetti-Pierri et al., 2008 & Mefford et al., 2008). The duplication found in patient 1 partially overlaps with a previously reported, syndrome-causing duplication. The phenotype of patient 1 is much more severe than the 1q21.1 duplication syndrome, apparently due to the *VPS41* mutation. However, we cannot rule out that fibroblasts are also affected by the 1q21.1 duplication. The homozygous *UPF3A* mutation identified in patient 3 is of unknown pathogenicity. Patient 3 has an unaffected sibling with a heterozygous mutation in *UPF3A*. Therefore, we cannot conclude if the disease phenotype of patient 3 could in part be caused by the homozygous *UPF3A* mutation. Based on these considerations we decided to focus our studies in fibroblasts on those derived from patient 2, which gave us the opportunity to specifically study the cellular consequences of mutations in *VPS41*.

Are GAG levels or activities of lysosomal enzymes affected in fibroblasts?

The following information on the GAG levels and activities of lysosomal enzymes were added to the manuscript:

Patient 1&2: Extensive investigation for metabolic diseases, including lysosomal and mitochondrial disorders, showed no detectable abnormalities. No urinary mucopolysaccharides and oligosaccharides were detected. Cerebrospinal fluid analysis for neurotransmitter levels showed no abnormalities (p6).

Patient 3: Extensive metabolic investigation including analysis for peroxisomal and lysosomal disorders, including urinary mucopolysaccharides and oligosaccharides as well as CDG (sialotransferrines), showed no abnormalities (p8).

6. The clinical description of all three cases should be more consistent and should include in all three cases: growth parameters (especially head circumference over time), developmental milestones, and cognitive development. Do patients 1 and 2 have liver and spleen enlargement, retinal dystrophy, or skeletal abnormalities like case 3? Was an echocardiogram performed?

We rephrased the description of the patients to make this more uniform and to the best of our abilities address the additional requested information (p6-9). In summary, patients 1 and 2 did not have hepatosplenomegaly and their echocardiographs were normal. We added the head circumference growth charts over time of all three patients (**new Figure EV1B, see also below**). Of note, the patients were monitored and treated in different countries: patient 1 and 2 are based in Toronto, Canada, and patient 3 in Groningen, The Netherlands.

7. The genomic coordinate of the 1q21.1 duplication should be included. Does this region overlap with the 1q21.1 duplication syndrome?

Microarray analysis revealed a de novo duplication of 2.568Mb at 1q21.1 [GRCh37/hg19 chr1:145,804,790-148,817,029]. This region indeed partially overlaps with the 1q21.1 duplication syndrome, but, as also indicated above, the clinical features of patient 1 are much more severe than in individuals suffering from 1q21.1 duplication syndrome. The information regarding identification and localization of the 1q21.1 duplication is now added to the case report in the manuscript (p7, 10).

8. The hand X-ray of case 3 is interesting because it shows metacarpal pointing, a skeletal finding often seen in lysosomal storage disorders.

A phenotypical similarity between patients suffering from a classical lysosomal storage disorder and *VPS41* patients would have indeed been interesting. However, our data (EM images, immunofluorescence data and metabolic studies performed on patient fibroblasts, leucocytes and urine), show that *VPS41* patients do not suffer from a lysosomal storage disorder. There are no enlarged lysosomes with undegraded material visible, and lysosomal functionality in terms of acidification and enzyme activity is normal. We have extended the comparison between *VPS41* disease and lysosomal storage disorders in the manuscript (p10).

9. The metabolic studies performed need to be specified. Were GAG levels measured in the patients? Were lysosomal enzyme activities measured?

The metabolic studies are now specified in the manuscript. Metabolic diseases including lysosomal and mitochondrial disorders showed no detectable abnormalities and no urinary mucopolysaccharides and oligosaccharides were detected (p6 and p8).

LYSOSOMAL/AUTOPHAGIC PHENOTYPE

It is unclear how the lysosomal-autophagic compartments are affected by VPS41 dysfunction. It is equally unclear how the cellular phenotype is an indication of a disease-relevant dysfunction. Specific points below:

We show that the molecular basis of *VPS41* disease is a deregulation of the HOPS complex. Without HOPS the fusion of lysosomes with late endosomes and autophagic membranes is inhibited and enzymatically active lysosomes are poorly reached by endocytic/autophagy material. Novel fundamental information coming from our studies is that HOPS dysfunction leads to partial inhibition of the mTORC1 complex and specific activation of the TFE3/TFEB signaling pathway. Hence, *VPS41* patient cells have higher basal autophagy levels, and react less well to autophagic stimuli. Since the autophagy pathway is important for clearance of cytoplasmic aggregates, defects in autophagy are known to cause neurodegeneration. Defects in autophagy are also possible explanations for the importance of *VPS41* in Alzheimers and Parkinsons disease (Griffin et al., (2018) and Harrington et al., (2012)). mTORC1 is a druggable pathway that is investigated in the context of other neurological disorders (Walters and Cox, (2018), Thellung et al., (2019) and Walter et al., (2020)).

10. Immuno-EM analyses in Figure 1 show that cells carrying *VPS41* mutants display smaller lysosomes. This analysis was performed on lysosomes labeled with BSA, which, however, the authors then show to only label a subpopulation of lysosomes, as endocytosis is defective in *VPS41*-deficient cells (Figure 3).

For the quantifications shown in Fig. 2E-F, lysosomes were defined by immuno-EM as LAMP-1-positive organelles with morphological characteristics specific for lysosomes (i.e. the presence of dense, amorphous, degraded material). BSA-Au⁵ was added to the cells to visualize the endocytic flow to lysosomes, but not to define the lysosomes. This approach showed that in patient fibroblasts lysosomes are smaller and less well reached by BSA-Au⁵, but contain normal levels of LAMP-1. We have specified this more clearly in the text (“morphologically identified, LAMP-1 positive lysosomes in patient derived cells were significantly smaller than in *VPS41*^{WT/WT}, *VPS41*^{WT/S285P} or *VPS41*^{WT/R662*} fibroblasts (Fig. 2E)” (p11).

What is the size of BSA-negative lysosomes in *VPS41*-mutated cells?

BSA-negative lysosomes in patient fibroblasts are significantly smaller than BSA-negative lysosomes in parental and control cells. For the reviewer's convenience, we added the quantifications of these measurements below (**See Figure below**).

Most importantly, why are lysosome smaller and is this decrease in size relevant to the disease phenotype (i.e. are lysosomes dysfunctional?). LAMP staining?

Lysosomal size is controlled by a diversity of pathways (see recent review by Araujo et al., (2020), now added to the paper), the combination of which determines their actual size. Within this complexity, a plausible explanation for the small lysosomes in *VPS41*-deficient cells is the combination of increased lysosomal biogenesis by constitutive activation of TFE3/TFEB (as described in Settembre et al., (2011) and Martina et al., (2014)), with the HOPS dependent defect in endo-lysosomal fusion (providing less membrane). We have extended our statements and conclusions on these findings in the discussion (p23). We can only speculate on how a decrease in lysosomal size contributes to the observed disease phenotype. The small lysosomes are functional, they contain active lysosomal enzymes (as shown by our live cell assays), but the kinetics by which they are reached by endocytic or autophagic cargo are slowed down. It may be that small sized lysosomes simply reflect a lack of cargo supply. Alternatively, the small size may impact the function of lysosomes as residence site for signaling platforms, such as the mTORC1 complex (p23, 24).

11. Analysis of the autophagic flux requires further investigation. The defective increase of LC3-II in patient fibroblasts indicates lack of autophagy induction, rather than lack of fusion. To address this question, we re-invested the LC3 expression in fibroblasts by calculating the LC3II-LC3I ratio on western blots and by performing immuno-fluorescence using the Cosmo Bio LTD, MAP1LC3A/LC3 (CAC-CTB-LC3-1-50) antibody. In a parallel study in our lab (De Maziere et al., paper in preparation), we recently found that this antibody greatly exceeds other available anti-LC3 antibodies in performance in fluorescence microscopy, resulting in an optimal signal to background ratio. We first determined if the basal autophagy levels in patient cells are increased by establishing the LC3II:LC3I ratio. On western blot we indeed found a higher LC3II-LC3I ratios in patient cells, indicating more autophagosomes (**new Fig 4A, A', see also below**). Concomitantly, immunofluorescence labeling of LC3 in patient cells showed in steady state more LC3-positive compartments (**new Fig 4B, B', see also below**). After starvation, both control and patient fibroblasts showed an increase in LC3-positive compartments. However, whereas in control cells this amounted up to a 13.4-fold increase, in patient cells only a 3.8-fold increase in LC3 spots/autophagosomes was seen. These findings indicate that, like *VPS41*^{KO} cells, patient fibroblasts have a higher basal level of autophagy and are impaired in their ability to react on nutrient stress. We think this is a crucial addition to the paper and thank the reviewer for challenging us to do these experiments!

Accordingly, if newly synthesized autophagosomes are not able to fuse with lysosomes, there should be a higher increase of LC3-II in patient cells compared to controls, whereas the data indicate the opposite.

As shown above, with our new measurements, we show that the LC3II-LC3I ratio in steady state conditions is more than doubled in patient cells and with the new LC3 antibody we find by immunofluorescence more LC3 spots in patient fibroblasts than control cells. Moreover, whereas starvation does result in an increase of LC3-positive autophagosomes in both control and patient cells, in patient cells this increase is 3.5 times lower than in control fibroblasts, showing a clear defect in the starvation response of patient fibroblasts.

In order to properly evaluate this point and discriminate induction from fusion, the experiments performed in Figure 4d using GFP-RFP-LC3 should be performed not only in VPS41-deficient cells but also in cells reconstituted with VPS41-mutants.

We performed the requested rescue experiment, but unfortunately double transfection (LC3-GFP.RFP + VPS41) was not tolerated by the cells; we could not find cells expressing both constructs at experimental levels.

In addition, the authors should add treatment with bafilomycin to better follow the autophagy defect and also IF of LC3 on fibroblasts.

To address this question, we set out to quantitate western blots of LC3II-LC3I ratios in non-treated versus Bafilomycin (Baf1A) treated fibroblasts. We used both control and patient fibroblasts, in steady state and after starvation conditions. Unfortunately, Baf1A treated fibroblasts showed a very high standard deviation between experiments, which prevented us to draw statistically relevant conclusions. We then decided to perform the same Baf1A treatment in HeLa^{WT} and HeLa^{VPS41^{KO}} cells, which offers a more uniform population of cells (**New Figure EV3A, see also below**). In all conditions, BafA1 treatment resulted in increased LC3-II levels in WT cells. By contrast, VPS41^{KO} cells were relatively insensitive to BafA1 (**New figure EV3B, see also below**), indicating that lack of VPS41 inhibits lysosomal

degradation of LC3. Since LC3 is a CLEAR target, starvation will result in increased expression due to nuclear translocation of TFE3/TFEB. Interestingly, starved WT cells (autophagy induction) with BafA1 (impaired lysosomal degradation) showed LC3 levels similar to starved *VPS41*^{KO} cells without BafA1 (**New Figure EV3C, see also below**). These data show that the increased LC3-levels in *VPS41*^{KO} cells is a combined effect of increased induction and impaired autophagosome – lysosome fusion.

Of note, the autophagy phenotype in addition to increased LC3II levels also comprises the low level of lysosome-associated mTORC1 and nuclear localization of TFE3, both independent of nutrient state, and found in both patient and HeLa^{*VPS41*KO} cells (Figure 5A – C and Figure EV4B).

Why are LC3 levels significantly higher in the *VPS41* KO HeLa compared to patients' fibroblasts?

These are different cell types and therefore difficult to compare; some cells may have higher basal levels of autophagy than others. Besides this, the patient fibroblasts might have developed epigenetic compensatory mechanisms to avoid high autophagy levels under basal conditions. Also, patient fibroblasts express a *VPS41* variant, whereas the HeLa cells are completely depleted of *VPS41*.

12. Co-ip experiments in Figure 3b show a weak interaction of *VPS41* only with one overexpressed member of the HOPS complex (*VPS11*). To assess whether *VPS41* mutants correctly assemble into the HOPS complex, endogenous components of the complex (*VPS18*, *VPS11*, *VPS33A*) should be assessed for their interaction with WT and mutant *VPS41* (as previously described in manuscript PMID: 28931724) transfected in *VPS41*-KO cells.

To address this comment, we performed additional Co-IPs with endogenous *VPS18*, *VPS11* and *VPS39* in HeLa^{*VPS41*KO} cells. This showed that *VPS41*^{WT} as well as *VPS41*^{S285P} bind endogenous *VPS18*, whereas no interaction between *VPS41*^{R662*} and endogenous *VPS18* was observed. The co-IPs with endogenous *VPS11* and *VPS39* were inconclusive, since the

available antibodies fail to detect endogenous protein levels. We added the new data to the Supplementary figures (**New Appendix Figure S4B, see also below**).

mTORC1 SIGNALING AND TFE3/TFEB

The role of VPS41 in the modulation of mTORC1 signaling and regulation of TFEB/TFE3 function is unclear. Based on the data presented, the disease-relevance of the observed TFE3 nuclear translocation remains questionable. What is the consequence of TFE3 activation?

Constitutive nuclear translocation, and therefore activation, of TFE3 results in the transcription of genes involved in lysosomal biogenesis and autophagy (Sardiello et al., (2009) and Palmieri et al., (2011)). The precise consequences of constitutive activation of TFE3 for disease-development in *VPS41* patients remains unknown, but our data clearly show that the regulation of this important signaling pathway is comprised.

Also, what was the rationale of analyzing TFE3 and not TFEB?

TFE3 and TFEB are both members of the MITF gene family, transcription factors that bind to CLEAR elements of genes involved in lysosomal biogenesis and autophagy (Settembre et al., (2011) and Martina et al., (2014)). We choose for TFE3 since in contrast to TFEB, endogenous TFE3 can be easily labeled for fluorescence microscopy obviating the need for transfection with TFEB constructs. To show that the TFE3 data also hold for TFEB we have now added a western blot to our revised manuscript, visualizing endogenous TFEB levels. This shows in HeLa^{WT} cells a molecular weight shift upon (de)phosphorylation of TFEB. This shift is not observed in *VPS41*^{KO} cells, indicating that TFEB, like TFE3, is not phosphorylated in *VPS41*^{KO} cells (**new Figure EV4F, see also below**).

Some specific points:

13. What is the mechanism by which VPS41 loss-of-function impairs mTORC1 lysosomal localization?

This is a very interesting question on which we at this moment can only speculate. A recent publication by Napolitano *et al.*, (2020) shows that the phosphorylation of distinct mTORC1 substrates is differentially regulated, and that regulation of TFE3/TFEB is dependent on folliculin. If VPS41, as part of the HOPS complex, is indeed involved in this folliculin-dependent regulatory step remains to be elucidated, and is currently investigated in our lab.

Why would this effect on mTORC1 affect only TFE3 phosphorylation (and likely TFEB) and does not affect S6K and 4E-BP1 phosphorylation?

Like the reviewer we were surprised and initially even skeptical by our finding that *VPS41* depletion causes a defect in TFE3 but not S6K or 4E-BP1 phosphorylation. We repeated these experiments many times, but this was the consistent outcome. At the time of submitting the original paper there were some indications in literature that mTORC1 could differentially phosphorylate distinct substrates, which we mentioned in our discussion (Lawrence *et al.*, (2019)). As also referred to in the previous response, very recently these findings were backed up by the publication of Napolitano *et al.*, (2020) showing that mTORC1 activity towards different substrates is indeed differentially regulated in which phosphorylation of TFE3/TFEB is dependent on Folliculin. Why *VPS41*/HOPS only affects the TFE3/TFEB arm of the mTORC1 signaling pathway and if this is in collaboration with Folliculin we can only speculate, but it is interesting to note that *VPS41* deficiency, which induces defects in lysosome biogenesis, specifically affects the lysosome pathway (TFE3/TFEB) and not cell growth (S6K, 4E-BP1). How *VPS41*, as part of the HOPS complex, is involved in TFE3/TFEB regulation is an ongoing project in our lab.

The data on mTORC1 activity (only Fig. 5f!!!) are too limited and not convincing.

To our knowledge, S6K1 and 4EBP1 are the main substrates of mTORC1 and often used in publications to measure mTORC1 (in)activity (e.g., Hesketh *et al.*, 2020). To further strengthen our data and add an additional substrate, we performed a western blot of phospho-ULK1, a substrate of mTORC1 involved in autophagy initiation (**new Figure EV5D, see also below**). This shows no difference in phospho-ULK1 levels between HeLa^{WT} and HeLa^{VPS18KO}, HeLa^{VPS39KO} or HeLa^{VPS41KO} cells, further indicating a specific role for HOPS in regulation of the TFE3/TFEB axis.

14. Fig 5A: Analysis of mTORC1 lysosomal localization is puzzling. How was quantification of mTOR localization at the lysosome performed? How many cells?

We added the requested information to the figure legends and methods section (p27, 44).

The authors claim that mTOR lysosomal localization is impaired in patient fibroblasts compared to control cells. However, Figure 5a' shows a strong impairment in mTOR lysosomal localization in WT/R662* and WT/S285P fibroblasts in both basal and re-feeding conditions, in which the levels of mTOR-LAMP-1 colocalization appear even lower than in

starved control cells. Thus, no conclusions can be made on mTORC1 lysosomal localization based on fibroblast.

We realized that the presentation of the data in this figure was unclear and we changed the manner of data visualization. The data clearly show that the parental cell lines respond to nutrient starvation and restimulation, whereas patient fibroblasts show no response. However, it should be noted that all data are presented relative to the steady state colocalization levels *per cell line*. Therefore, colocalization between cell lines cannot be compared. Based on these re-quantifications we added a new figure to the manuscript (**new Figure 5A', see also below**) and removed the old figure.

To this end, the same analysis should be done on VPS41-KO HeLa cells reconstituted with WT and mutant VPS41.

We performed the requested analysis on VPS41^{KO} HeLa cells reconstituted with WT and mutant VPS41 (**new Figure EV4B, C, see also below**). This showed that reintroducing VPS41^{WT} increased mTORC1/Cathepsin D colocalization significantly, whereas expression of either mutant (S285P or R662*) had little effect on the lysosomal recruitment of mTORC1.

15. How was the "% nuclei positive for TFE3" calculated?

Cells with a defined nuclear outline due to the nuclear signal of TFE3 were scored as "positive". We added this clarification to the materials and methods section (p27).

In the images in Figure 5d, VPS41-KO cells reconstituted with WT-VPS41 show reduced nuclear TFE3 with no increase in cytosolic signal, which does not seem a "rescue" to WT cells. An increase in cytosolic signal is difficult to visualize since the signal is much more dispersed than in the contained volume of the nucleus. Absence or presence of TFE3 in the nucleus is widely used as a read-out for TFE3 localization (and thus activity), rather than increased or decreased cytosolic fluorescent intensity. For instance, see papers by Martina et al., (2014) and Roczniak-Ferguson et al., (2012).

Is it possible that the APEX-V5 tag on VPS41 affects its function?

We are confident that the APEX2-V5-tag does not affect VPS41 function. We have shown that this construct is able to interact with known VPS41 interactors (HOPS subunit VPS33A, Rab7 and Arl8b) and rescues the endo-lysosomal fusion defect in *VPS41^{KO}* cells. Moreover, we show that the tagged *VPS41* constructs normally localize to endo-lysosomal compartments. We need the presence of a tag to distinguish between transfected and non-transfected cells, since there is no antibody against VPS41 that works convincingly in immunofluorescent experiments.

These and other key experiments of the paper should be reproduced with untagged versions of WT and mutant VPS41.

We addressed this question above.

Finally, a more extensive and unbiased analysis of TFE3/TFEB targets should be analyzed by RNA-seq analysis.

The TFE3/TFEB targets were identified by Sardiello et al., (2009), and since then the CLEAR network has been topic of various extensive and focused studies (Palmieri et al., (2011), Roczniak-Ferguson et al., (2012)), reviewed by Napolitano and Ballabio (2016). We do not see the added value of another extensive analysis in the context of this manuscript. We do

show that the protein levels of two TFE3/TFEB targets (LAMP-1 and Cathepsin B, Appendix Figure S7) are increased in *VPS41* patient fibroblasts, and added western blot analysis on LAMP-1 and Cathepsin D in *VPS41*^{KO} cells (**new Figure Appendix S7C, C'**, see also below), which is in agreement with increased expression of the CLEAR network and consistent with the observed nuclear TFE3 localization.

16. Fig 5B: The authors should check the phosphorylation status of both TFE3 and TFEB either by using specific phospho-antibodies or by following molecular weight shift in presence and absence of mTOR inhibitors.

To address this question, we checked the phosphorylation status of TFEB in the absence or presence of mTOR inhibitor Torin-1 by Western Blotting. HeLa^{WT} cells clearly responded to starvation and Torin1 treatment by a shift in molecular weight of TFEB, indicating (de)phosphorylation. By contrast, HeLa^{VPS41KO} cells did not show a molecular shift, indicating that TFEB is not phosphorylated. The blot is added to Supplementary figures (**New Figure EV4F, see also below**).

17. Fig S9 in the text actually corresponds to S8. S10 in the text is actually S9.
We apologize for this mistake and corrected the numbering.

Referee #3 (Comments on Novelty/Model System for Author):

Although it would have been preferable to examine more mammalian neuronal models, the rise of cell lines to address basic questions of mutant VPS41 function was adequate.

Referee #3 (Remarks for Author):

This manuscript presents the identification of rare human VPS41 mutations as the basis for a complex neurodevelopmental/neurodegenerative disorder. The functionality of the patient mutations is further investigated in human cell lines and a *C. elegans* model of dopaminergic neuron degeneration. These efforts generally support a loss of function model wherein the mutant VPS41 can no longer fully support function of the HOPS complex in which it resides. This has consequences on a broad range of lysosome properties. Such changes are consistent with a well-known role for the HOPS complex in supporting fusion between endosomes and lysosomes. Collectively this manuscript identifies new human mutations in VPS41 as a cause of a neurological disorder and establishes that this is due to a loss-of-function mechanism via a variety of cell biological assays. The experiments are performed to a high standard of quality and the conclusions drawn are generally reasonable. The concerns that I outline below are relatively modest and should be readily addressable.

1. The title claims defects in mTORC1 signaling that are not supported by the data that is presented in Figure 5E which shows normal activation of mTORC1 signaling by amino acids in Figure 5F. This is the only piece of data in the manuscript that examines mTORC1 signaling and it contradicts the title. My suggestion is that the title should better represent the data.

We accept this suggestion and changed the title to "Neurodegenerative VPS41 variants inhibit HOPS function and mTORC1-dependent TFEB/TFE3 regulation".

2. Can the authors better explain the basis for defining the human condition arising from VPS41 mutations as neurodegenerative rather than neurodevelopmental?

The MRI findings for patient 1 and 2 are progressive and thus more consistent with a neurodegenerative rather than neurodevelopmental process. Patient 3 shows progressive hypertonia (the patient becoming more and more rigid during the years) and contractures. Unfortunately, we cannot document neurodegeneration based on MRI data for patient 3. The extremely severe and progressive scoliosis of the patient does not allow sedation or anesthesia for a follow-up MRI scan.

Although the *C. elegans* model is interesting, it does not establish that neurodegeneration is happening in humans with these mutations.

Previously we have published in *C. elegans* that knockdown of VPS41 in dopamine neurons only, as well as pan-neuronally, leads to enhanced neurodegeneration; this effect worsens with age (Harrington et al., (2012), Fig. 3B, C). Although we are aware that one should be cautious when translating *C. elegans* phenotypes to humans, the *C. elegans* data, together with the progressiveness observed in the MRIs of patient 1 and 2, are strong indicators that these VPS41 specific mutations cause neurodegeneration.

3. Speaking of LC-3 lipidation (page 12), the authors state that: "Notably, the autophagy defect is specific for the patient derived fibroblasts and not observed in any of the parental cell lines, indicative that this phenomenon is important for disease development." However, no evidence was presented to indicate a cause and effect relationship between defective autophagy and disease development.

The sentence is rewritten to: Notably, the autophagy defect is specific for the patient derived fibroblasts and not observed in any of the parental cell lines (p18).

4. Key mTORC1 signaling data in Figure 5F should be quantified.

We have performed the quantification and added this information to the manuscript (**new Figure 5E', see also below**).

Ideally, the mTORC1 phosphorylation site on ULK1 should also be examined as it is a major link between mTORC1 and autophagy.

We performed a western blot of phospho-ULK1 (**new Figure EV5D, see also below**). There is no difference in phospho-ULK1 between HeLa^{WT} and HeLa^{VPS18KO}, HeLa^{VPS39KO} and HeLa^{VPS41KO} cells, further indicating a specific role of HOPS in the regulation of TFE3/TFEB. We describe these data in the manuscript at p18.

5. The basis for the following concluding statement from page 20 is not clear: "In summary, our data suggest that the VPS41 patients may represent a novel class of lysosomal disorders in which lysosomes are enzymatically active but due to delayed trafficking kinetics are insufficiently reached by nutrients." It is not clear how the authors arrived at this conclusion or which nutrients this refers to. Although defects were noted in the subcellular localization of TFE3 and its regulation by changes in amino acid availability, mTOR signaling as presented in Figure 5E responded normal to the same stimuli. This suggests that the defect may be more narrowly related to TFE3 regulation rather than nutrient sensing mechanisms that are shared by TFE3 and mTORC1. It may not be necessary to perform further experiments to address this matter if the discussion can be modified to address this concern.

Since the regulation of phosphorylation of S6K and 4EBP1 is not altered in VPS41^{KO} cells compared to WT cells, we can conclude that nutrient sensing in VPS41^{KO} cells is not affected. Indeed, as the reviewer points out, VPS41 is more likely involved in the specific regulation of TFEB/TFE3 phosphorylation, rather than affecting nutrient sensing in general. A possible hypothesis could be the necessity of membrane association for TFEB/TFE3 for its

phosphorylation by mTORC1. In the *VPS41*^{KO} cells we observed reduced mTORC1 association on the lysosomal membrane, but phosphorylation of S6K and 4EBP1, substrates that do not associate to the lysosomal membrane, was not affected. If mTORC1 is not sufficiently retained on the lysosomal membrane, no (efficient) phosphorylation of TFEB/TFE3 will occur, resulting in constitutive activity of the transcription factors. If and how HOPS is involved in mTORC1 retention or TFEB/TFE3 recruitment remains to be elucidated, and is currently an ongoing project in our lab. We rewrote the discussion to address this comment of the reviewer (p23, 24).

6. In addition to reference 60 (page 19), the first demonstration of TFE3 translocation to the nucleus in response to chloroquine was in the following study: PMID: 22692423
We added this study to the manuscript and reference list.

7. Author names are missing from some references such as #21 and #30.
To the best of our knowledge, we corrected all errors in the reference list.

1st Feb 2021

Dear Dr. Klumperman,

Thank you for the submission of your revised manuscript to EMBO Molecular Medicine. We have now received the enclosed report from the three referees who were asked to re-assess it. As you will see, the referees are now overall supportive. I am pleased to inform you that we will be able to accept your manuscript pending the following amendments:

1. We would ask you to experimentally address Referee #2's concerns about the autophagy experiments in the patients' cell lines instead of eliminating these data from the paper.
2. Please address the rest minor concerns raised by Referees #2 and #3 in writing.

On a more editorial level:

1. In the main manuscript file, please do the following:

- remove the yellow color font
- remove "data not shown " (p.41). As per our guidelines, on "Unpublished Data" the journal does not permit citation of "Data not shown". All data referred to in the paper should be displayed in the main or Expanded View figures. "Unpublished observations" may be referred to in exceptional cases, where these are data peripheral to the major message of the paper and are intended to form part of a future or separate study, the names of the persons that reported the observation should be listed in brackets. Personal communications (Author name(s), personal communications) must be authorised in writing by those involved, and the authorisation sent to the editorial office at time of submission.
- Fig 6B is not called out, please fix.
- provide a section title to the Conflict of Interest statement.
- in Materials and Methods, include a statement that informed consent was obtained from all subjects and that the experiments conformed to the principles set out in the WMA Declaration of Helsinki and the Department of Health and Human Services Belmont Report.

2. Checklist:

- please enter both co-corresponding authors' names.

3. Please add a formal "Data Availability" section (placed after Materials & Method). Since this study does not generate large-scale datasets, please include the following sentence in this section- "This study includes no data deposited in external repositories".

4. For more information: There is space at the end of each article to list relevant web links for further consultation by our readers. Could you identify some relevant ones and provide such information as well? Some examples are patient associations, relevant databases, OMIM/proteins/genes links, author's websites, etc...

5. We now encourage the publication of source data, particularly for electrophoretic gels, blots, but also microscopy images with the aim of making primary data more accessible and transparent to the reader. Would you be willing to provide a PDF file per figure that contains the original, uncropped and unprocessed scans of all or key gels used in the figure (including molecular weight markers)?

The PDF files should be labeled with the appropriate figure/panel number (1 file/figure), and should have molecular weight markers; further annotation may be useful but is not essential. The PDF files will be published online with the article as supplementary "Source Data" files. If you have any questions regarding this just contact me.

6. I have slightly modified and shortened the synopsis text. Please let me know if it is fine like this or if you would like to introduce further modifications.

Mutations in VPS41 were identified in patients with a neurodegenerative phenotype. VPS41 variants cause decreased cargo transfer to lysosomes by inhibiting the HOPS complex and lead to an inhibition of mTORC1-mediated TFE3 regulation.

- Patients bearing compound heterozygous mutations in VPS41 suffer from a neurodegenerative disorder.
- Mutations or depletion of the VPS41 gene results in inefficient cargo delivery to lysosomes due to impaired HOPS complex functionality.
- HOPS complex impairment causes constitutive activation of TFE3 and TFEB, resulting in increased lysosomal biogenesis and autophagosome formation.
- Phosphorylation of other mTORC1 substrates is not affected, suggesting a role for the HOPS complex in regulating the differential phosphorylation of mTORC1 substrates.

7. Please move "the Paper Explained" to the main manuscript file. Also, would you mind expanding it a bit more? Please refer to any of our published articles for an example.

8. As part of the EMBO Publications transparent editorial process initiative (see our Editorial at <http://embomolmed.embopress.org/content/2/9/329>), EMBO Molecular Medicine will publish online a Review Process File (RPF) to accompany accepted manuscripts.

-In the event of acceptance, this file will be published in conjunction with your paper and will include the anonymous referee reports, your point-by-point response and all pertinent correspondence relating to the manuscript. Let us know if you do NOT agree with this.

I look forward to seeing a revised version of your manuscript as soon as possible.

Sincerely,
Jingyi

Jingyi Hou
Editor
EMBO Molecular Medicine

*** Instructions to submit your revised manuscript ***

To submit your manuscript, please follow this link:

<https://embomolmed.msubmit.net/cgi-bin/main.plex>

- 1) a .docx formatted version of the manuscript text (including Figure legends and tables)
- 2) Separate figure files*
- 3) supplemental information as Expanded View and/or Appendix. Please carefully check the authors guidelines for formatting Expanded view and Appendix figures and tables at <https://www.embopress.org/page/journal/17574684/authorguide#expandedview>
- 4) a letter INCLUDING the reviewer's reports and your detailed responses to their comments (as Word file).
- 5) The paper explained: EMBO Molecular Medicine articles are accompanied by a summary of the articles to emphasize the major findings in the paper and their medical implications for the non-specialist reader. Please provide a draft summary of your article highlighting
 - the medical issue you are addressing,
 - the results obtained and
 - their clinical impact.This may be edited to ensure that readers understand the significance and context of the research. Please refer to any of our published articles for an example.
- 6) For more information: There is space at the end of each article to list relevant web links for further consultation by our readers. Could you identify some relevant ones and provide such information as well? Some examples are patient associations, relevant databases, OMIM/proteins/genes links, author's websites, etc...
- 7) Author contributions: the contribution of every author must be detailed in a separate section.
- 8) EMBO Molecular Medicine now requires a complete author checklist (<https://www.embopress.org/page/journal/17574684/authorguide>) to be submitted with all revised

manuscripts. Please use the checklist as guideline for the sort of information we need WITHIN the manuscript. The checklist should only be filled with page numbers where the information can be found. This is particularly important for animal reporting, antibody dilutions (missing) and exact values and n that should be indicated instead of a range.

9) Every published paper now includes a 'Synopsis' to further enhance discoverability. Synopses are displayed on the journal webpage and are freely accessible to all readers. They include a short stand first (maximum of 300 characters, including space) as well as 2-5 one sentence bullet points that summarise the paper. Please write the bullet points to summarise the key NEW findings. They should be designed to be complementary to the abstract - i.e. not repeat the same text. We encourage inclusion of key acronyms and quantitative information (maximum of 30 words / bullet point). Please use the passive voice. Please attach these in a separate file or send them by email, we will incorporate them accordingly.

You are also welcome to suggest a striking image or visual abstract to illustrate your article. If you do please provide a jpeg file 550 px-wide x 400-px high.

10) A Conflict of Interest statement should be provided in the main text

11) Please note that we now mandate that all corresponding authors list an ORCID digital identifier. This takes <90 seconds to complete. We encourage all authors to supply an ORCID identifier, which will be linked to their name for unambiguous name identification.

Currently, our records indicate that the ORCID for your account is 0000-0003-4835-6228.

Link Not Available

12) The system will prompt you to fill in your funding and payment information. This will allow Wiley to send you a quote for the article processing charge (APC) in case of acceptance. This quote takes into account any reduction or fee waivers that you may be eligible for. Authors do not need to pay any fees before their manuscript is accepted and transferred to our publisher.

Photos 400-800 DPI

*Additional important information regarding figures and illustrations can be found at <https://bit.ly/EMBOPressFigurePreparationGuideline>

The system will prompt you to fill in your funding and payment information. This will allow Wiley to send you a quote for the article processing charge (APC) in case of acceptance. This quote takes into account any reduction or fee waivers that you may be eligible for. Authors do not need to pay any fees before their manuscript is accepted and transferred to our publisher.

***** Reviewer's comments *****

Referee #1 (Comments on Novelty/Model System for Author):

Van der Welle et al. report on the clinical description and cell biological changes in fibroblasts of patients with a novel disease caused by mutations in the VPS41 gene. Using various high quality techniques and state-of-the-art methods they provide new insight into the molecular pathomechanisms of this novel phenotype.

Referee #1 (Remarks for Author):

I am pleased with the requested changes and additional experiments which improved significantly the manuscript and will be of interest for a broad readership.

Referee #2 (Remarks for Author):

I would like to congratulate the authors for the excellent job they did in revising the manuscript and addressing all of my concerns. As a result of this, the manuscript has greatly improved.

Below are just a few minor points that the authors should consider:

In Table 1 please include the number of homozygotes for each variant.

In point 10 of my previous review I asked the authors to determine why lysosomes are smaller in VPS41 KO cell line and whether the smaller size contributed to the disease phenotype. Retrospectively, I must admit that it would not be so easy to answer these questions. As the authors suggest, the presence of small lysosomes may reflect induction of lysosomal biogenesis by TFEB/TFE3 as an attempt to compensate for a lysosomal defect. However, the authors also seem to propose that the presence of "small" lysosomes in VPS41-KO cells is a cellular phenotype that may be linked to disease pathogenesis. The authors should clarify that these two interpretations are in contrast. In my opinion, the first hypothesis (i.e. TFEB-induced biogenesis of new lysosomes as an attempt to compensate) is more likely to be the answer. In principle, this hypothesis could be tested by depleting TFEB and TFE3 in VPS41-KO cells to assess whether small lysosomes disappear and whether this approach results in an amelioration or worsening of the phenotype. However, at this point this may be out of the scope of this manuscript.

Point 11 of my previous review relates to the analysis of the autophagic pathway. I find that whereas the data obtained using the VPS41 KO cell line are solid, the data obtained in the patients' cell lines are weak and too variable. Contrary to the data obtained in the VPS41 KO cell line, the autophagy data on the patients' cell line do not show a clear impairment of the starvation response. Such variability is not unexpected, as it is well known that working with patients' cell lines may lead to variable results. In addition, there are some technical problems in the figures. In Fig 4A to compare autophagy levels the immunoblot should show both control and patient samples on the same gel. Otherwise one can only determine how autophagy activation or shutdown works for each

cell line, not being able to compare them. Accordingly, quantification in A' should be corrected and two different plots must be shown, one for WT and a different one for patient samples. Due to these difficulties, I suggest one of the following options: 1) repeat the autophagy experiments in the patients' cell lines to obtain clean and robust data or 2) eliminate these data from the paper and, based on the data obtained in the VPS41 KO cell line, speculate what the situation may be in the patients.

Referee #3 (Remarks for Author):

The authors have addressed my concerns. However, the revised manuscript contains 2 pieces of data that are seemingly at odds with the conclusions. It would be helpful if the authors could provide an explanation.

1, Figure 5A' reports that VPS41 S285P/R662* mutants results in constitutive localization of mTOR to lysosomes regardless of nutrient status. This contrasts with conclusions elsewhere that VPS41 is required for the localization of mTOR to lysosomes.

2. Figure 5D and D' report that TFE3 is localized to the nucleus prior to starving the cells expressing WT VPS41. This does not fit with conclusions elsewhere that TFE3 is largely excluded from the nucleus of WT cells under fed conditions.

Dear Jingyi Hou,

We hereby submit the re-revised version of our manuscript in which we included a new figure 4A and made textual changes to address the final questions raised by the reviewers. We thank you for the careful handling of this manuscript and hope you now find our work acceptable for publication.

Sincerely,
Judith Klumperman

Comments from the editor

1. We would ask you to experimentally address Referee #2's concerns about the autophagy experiments in the patients' cell lines instead of eliminating these data from the paper.

We have addressed the comments from Referee #2 by adding a new, representative blot to the manuscript (Fig 4A), and elaborated on the autophagy data obtained in patient fibroblasts in the manuscript.

2. Please address the rest of the minor concerns raised by Referees #2 and #3 in writing.

The minor comments from Referees #2 and #3 are addressed below and, where appropriate, we made changes in the manuscript.

Editorial concerns

1. In the main manuscript file, please do the following:

- remove the yellow color font

Yellow text is removed.

- remove "data not shown " (p.41). As per our guidelines, on "Unpublished Data" the journal does not permit citation of "Data not shown". All data referred to in the paper should be displayed in the main or Expanded View figures. "Unpublished observations" may be referred to in exceptional cases, where these are data peripheral to the major message of the paper and are intended to form part of a future or separate study, the names of the persons that reported the observation should be listed in brackets. Personal communications (Author name(s), personal communications) must be authorised in writing by those involved, and the authorisation sent to the editorial office at time of submission.

"Data not shown" was removed upon submission. We also removed the personal communications mentioned in the discussion.

- Fig 6B is not called out, please fix.

Reference to Fig 6B is added to the manuscript (p19). Our apologies for the mistake.

- provide a section title to the Conflict of Interest statement.

Title "Conflict of Interest" is added (p38).

- in Materials and Methods, include a statement that informed consent was obtained from all subjects and that the experiments conformed to the principles set out in the WMA Declaration of Helsinki and the Department of Health and Human Services Belmont Report.

The requested statement is added to the Materials and Methods section (p26).

2. Checklist:

- please enter both co-corresponding authors' names.

The names of both corresponding authors (Judith Klumperman and David Chitayat) are now mentioned in the Checklist.

3. Please add a formal "Data Availability" section (placed after Materials & Method). Since this study does not generate large-scale datasets, please include the following sentence in this section- "This study includes no data deposited in external repositories".

The "Data Availability" section is added to the manuscript stating that our study does not include data deposited in external repositories (p32).

4. For more information: There is space at the end of each article to list relevant web links for further consultation by our readers. Could you identify some relevant ones and provide such information as well? Some examples are patient associations, relevant databases, OMIM/proteins/genes links, author's websites, etc...

We have added a "For more information" section at the end of the manuscript with the following links (p38):

- <https://omim.org/entry/605485>
- <https://www.movementdisorders.org/>
- <https://www.cellbiology-utrecht.nl/research/klumperman.html>
- <https://www.lmp.utoronto.ca/faculty/david-chitayat>
- <https://www.michaeljfox.org/>

5. We now encourage the publication of source data, particularly for electrophoretic gels, blots, but also microscopy images with the aim of making primary data more accessible and transparent to the reader. Would you be willing to provide a PDF file per figure that contains the original, uncropped and unprocessed scans of all or key gels used in the figure (including molecular weight markers)? The PDF files should be labeled with the appropriate figure/panel number (1 file/figure), and should have molecular weight markers; further annotation may be useful but is not essential. The PDF files will be published online with the article as supplementary "Source Data" files. If you have any questions regarding this just contact me.

We have added Source data of the key blots shown in the manuscript and uploaded these separately. We have added Source data for the following Figures:

- Figure 2
- Figure 4
- Figure 5
- Figure EV3
- Figure EV4
- Figure EV5
- Appendix Figure S3
- Appendix Figure S4
- Appendix Figure S8
- Appendix Figure S9

6. I have slightly modified and shortened the synopsis text. Please let me know if it is fine like this or if you would like to introduce further modifications.

We made some small additional changes in the synopsis text and uploaded the new version in our re-submission.

7. Please move "the Paper Explained" to the main manuscript file. Also, would you mind expanding it a bit more? Please refer to any of our published articles for an example.

We have expanded "The Paper Explained" and added it at the end of the manuscript (p55).

Reviewer comments

Referee #1 (Remarks for Author):

I am pleased with the requested changes and additional experiments which improved significantly the manuscript and will be of interest for a broad readership.

We kindly thank the reviewer for the positive evaluation of our work.

Referee #2 (Remarks for Author):

I would like to congratulate the authors for the excellent job they did in revising the manuscript and addressing all of my concerns. As a result of this, the manuscript has greatly improved.

We are happy to see this comment.

Below are just a few minor points that the authors should consider:

In Table 1 please include the number of homozygotes for each variant.

The three patients described in our manuscript all suffer from compound heterozygote mutations in VPS41. Therefore, there are no homozygotes for any of the variants. To clarify this in table 1, we have added an extra column and added information about the patients (see also below).

VPS41 (NM_014396.3)	gnomAD global	SIFT	CADD	PolyPhen2 HDIV	PolyPhen2 HVAR	Patient
c.853 T>C p.S285P	0.0000071; 2/282128	0.048 damaging	23.5	0.956 possibly damaging	0.361 benign	1 and 2
c.1984 C>T p.R662*	0.00035; 98/282180	(b)	40	(b)	(b)	1, 2 and 3
c.1423-2A>G p.? (a)	0.000032; 1/31398	(b)	34	(b)	(b)	3

Frequency of the specific VPS41 variants in the general population based on the gnomad database ([variants observed]/[total of individuals studied]) (<http://gnomad.broadinstitute.org/>) and prediction of pathogenicity based on the programs SIFT, (<https://sift.bii.a-star.edu.sg/>), CADD (<https://cadd.gs.washington.edu/>) and PolyPhen (<http://genetics.bwh.harvard.edu/pph2/>).

(a): Three different splice site analysis programs (MaxEnt, NNSPLICE and SSF) predict a total loss of wild type acceptor splice site (Interactive Biosoftware - Created by Alamut Visual v.2.15.0).

(b): Not determined; SIFT and PolyPhen2 programs can only be used for analysis of missense mutations.

In point 10 of my previous review I asked the authors to determine why lysosomes are smaller in VPS41 KO cell line and whether the smaller size contributed to the disease phenotype. Retrospectively, I must admit that it would not be so easy to answer these questions. As the authors suggest, the presence of small lysosomes may reflect induction of lysosomal biogenesis by TFEB/TFE3 as an attempt to compensate for a lysosomal defect. However, the authors also seem to propose that the presence of "small" lysosomes in VPS41-KO cells is a cellular phenotype that may be linked to disease pathogenesis. The authors should clarify that these two interpretations are in contrast. In my opinion, the first hypothesis (i.e. TFEB-induced biogenesis of new lysosomes as an attempt to compensate) is more likely to be the answer. In principle, this hypothesis could be tested by depleting TFEB and TFE3 in VPS41-KO cells to assess whether small lysosomes disappear and whether this approach results in an amelioration or worsening of the phenotype. However, at this point this may be out of the scope of this manuscript.

Indeed, we propose that the small lysosomes may be the result of increased lysosomal biogenesis caused by constitutive activation of TFEB/TFE3 as well as of the reduced supply of membrane due to impaired HOPS mediated fusion. We do not regard this as contrasting statements, both mechanisms can occur. We do consider the small lysosomes as a cellular

phenotype linked to disease, by which we mean that we observe this phenotype in the patient fibroblasts and after VPS41 depletion. But we do not imply that small lysosomes cause the patients phenotypes. As mentioned before, a direct link between genotype and phenotype is unknown for many lysosomal disorders, and at this point we do not know if the small lysosomes actually contribute to the clinical symptoms or represent a secondary, perhaps compensatory, defect. We carefully checked our paper and confirmed that we refrain from making such statements. We acknowledge that the proposed experiment would be interesting. However, since depletion of TFEB and TFE3 will not overcome the possible fusion defect due to the impaired HOPS complex in *VPS41^{KO}* cells, rescue of the smaller lysosomes might not be trivial to achieve and we believe this experiment is indeed out of scope for this manuscript.

Point 11 of my previous review relates to the analysis of the autophagic pathway. I find that whereas the data obtained using the VPS41 KO cell line are solid, the data obtained in the patients' cell lines are weak and too variable. Contrary to the data obtained in the VPS41 KO cell line, the autophagy data on the patients' cell line do not show a clear impairment of the starvation response. Such variability is not unexpected, as it is well known that working with patients' cell lines may lead to variable results. In addition, there are some technical problems in the figures. In Fig 4A to compare autophagy levels the immunoblot should show both control and patient samples on the same gel. Otherwise one can only determine how autophagy activation or shutdown works for each cell line, not being able to compare them. Accordingly, quantification in A' should be corrected and two different plots must be shown, one for WT and a different one for patient samples. Due to these difficulties, I suggest one of the following options: 1) repeat the autophagy experiments in the patients' cell lines to obtain clean and robust data or 2) eliminate these data from the paper and, based on the data obtained in the VPS41 KO cell line, speculate what the situation may be in the patients.

We want to stress that the quantifications shown in Figure 4A' are based on the ratio LC3II:LC3I and not absolute LC3 levels normalized to the actin loading control. This means that each cell line is its own internal control and for our quantitation's we do not need to compare between different cell types or blots. We have clarified this better in the re-revised manuscript to avoid misinterpretation (p15). That said, we do acknowledge the request to analyze all cell types on the same blot, just to avoid any variations that could be caused by loading conditions. We therefore repeated the experiment and added a new blot to the manuscript, replacing the old Figure 4A (**see also below**). The data confirm our original experiments that patient cells overall have a higher LC3:LC3I ratio than control cells, in all nutrient conditions, indicating a defect in the autophagy response.

Indeed, the magnitude of the autophagy phenotype in patient cells is less conspicuous than in *VPS41^{KO}* cells; patient cells show some response to starvation (increased LC3II:LC3I ratios and more LC3-positive compartments after starvation), whereas *VPS41^{KO}* show complete irresponsiveness to all autophagic clues. This discrepancy in data may be, as the reviewer points out, due to the high variation in the fibroblast cultures, but also to compensatory mechanisms developed in the patient cells or cell type dependent differences. Most

importantly, we notice in different cell types and using various assays the same trend towards a defect in autophagy in the absence of functional VPS41.

Referee #3 (Remarks for Author):

The authors have addressed my concerns. However, the revised manuscript contains 2 pieces of data that are seemingly at odds with the conclusions. It would be helpful if the authors could provide an explanation.

1. Figure 5A' reports that VPS41 S285P/R662* mutants results in constitutive localization of mTOR to lysosomes regardless of nutrient status. This contrasts with conclusions elsewhere that VPS41 is required for the localization of mTOR to lysosomes.

In the first round of revisions, reviewer 2 requested a re-quantification of these data (reviewer comment #14) which resulted in current Figure 5A'. The quantifications show that the patient fibroblasts do not respond to nutrient starvation in terms of lysosomal mTORC1 association. It is important to note that Figure 5A' shows the quantifications of mTORC1/LAMP-1 colocalization per cell line, relative to steady state conditions. The numbers do not represent absolute colocalization values and consequently the different cell lines cannot be compared to each other, only internally as response to starvation. To clarify this, we added additional information about the quantification in the manuscript (p17), and made small alterations to the figures (see also below).

2. Figure 5D and D' report that TFE3 is localized to the nucleus prior to starving the cells expressing WT VPS41. This does not fit with conclusions elsewhere that TFE3 is largely excluded from the nucleus of WT cells under fed conditions.

In this rescue experiment we show that the TFE3 phenotype is rescued when overexpressing VPS41^{WT}, but that neither of the mutants is able to do so. The data the reviewer is referring to, is the nuclear localization of TFE3 in VPS41^{KO} cells re-transfected with VPS41^{WT} in steady state conditions. This is a good point. Indeed, under normal conditions, TFE3 should be

cytosolically retained and not be present in the nucleus. However, transfection of constructs, as is performed in this experiment, often induces re-localization of TFE3/TFEB to the nucleus, probably caused by damaged endolysosomal membranes and resulting lysosomal stress (Sancak *et al*, 2010 & Pan *et al*, 2019). The TFE3 localization after starvation and restimulation are the most informative panels of this experiment, representing the most favorable conditions for cytosolic retention of TFE3. This shows that in VPS41^{WT} expressing cells, TFE3 is no longer localized to the nucleus after restimulation, hence the phenotype is rescued. To clarify the interpretation of this experiment we added a more elaborate text on these results and their interpretation in the manuscript (p17).

18th Feb 2021

Dear Dr. Klumperman,

Thank you for the submission of your revised manuscript to EMBO Molecular Medicine. We have now received the enclosed reports from the referees that were asked to re-assess it. As you will see the referees are now globally supportive and I am pleased to inform you that we will be able to accept your manuscript pending the following final amendments:

1. Please address the minor concern raised by Referee #3.

On a more editorial level:

1. Figure EV1E: We understand that the parents have given written consent for the publication of clinical pictures. However, to minimize any potential risk of being exploitative, we would recommend adding a narrow black box to cover the eyes on the three panels. The sentence "Parents provided written informed consent for publication of clinical pictures" should be clearly stated in the paper.

2. Our data editor has made a couple of comments on your manuscript (see attached). Please address these issues in a .docx formatted version of the manuscript and keep the track mode on.

I look forward to reading a new revised version of your manuscript as soon as possible.

Sincerely,
Jingyi

Jingyi Hou
Editor
EMBO Molecular Medicine

*** Instructions to submit your revised manuscript ***

To submit your manuscript, please follow this link:

<https://embomolmed.msubmit.net/cgi-bin/main.plex>

- 1) a .docx formatted version of the manuscript text (including Figure legends and tables)
- 2) Separate figure files*
- 3) supplemental information as Expanded View and/or Appendix. Please carefully check the authors guidelines for formatting Expanded view and Appendix figures and tables at <https://www.embopress.org/page/journal/17574684/authorguide#expandedview>
- 4) a letter INCLUDING the reviewer's reports and your detailed responses to their comments (as Word file).
- 5) The paper explained: EMBO Molecular Medicine articles are accompanied by a summary of the articles to emphasize the major findings in the paper and their medical implications for the non-specialist reader. Please provide a draft summary of your article highlighting
 - the medical issue you are addressing,
 - the results obtained and
 - their clinical impact.This may be edited to ensure that readers understand the significance and context of the research. Please refer to any of our published articles for an example.
- 6) For more information: There is space at the end of each article to list relevant web links for further consultation by our readers. Could you identify some relevant ones and provide such information as well? Some examples are patient associations, relevant databases, OMIM/proteins/genes links, author's websites, etc...
- 7) Author contributions: the contribution of every author must be detailed in a separate section.
- 8) EMBO Molecular Medicine now requires a complete author checklist (<https://www.embopress.org/page/journal/17574684/authorguide>) to be submitted with all revised manuscripts. Please use the checklist as guideline for the sort of information we need WITHIN the manuscript. The checklist should only be filled with page numbers where the information can be found. This is particularly important for animal reporting, antibody dilutions (missing) and exact values and n that should be indicated instead of a range.
- 9) Every published paper now includes a 'Synopsis' to further enhance discoverability. Synopses are displayed on the journal webpage and are freely accessible to all readers. They include a short stand first (maximum of 300 characters, including space) as well as 2-5 one sentence bullet points that summarise the paper. Please write the bullet points to summarise the key NEW findings. They should be designed to be complementary to the abstract - i.e. not repeat the same text. We encourage inclusion of key acronyms and quantitative information (maximum of 30 words / bullet point). Please use the passive voice. Please attach these in a separate file or send them by email, we will incorporate them accordingly.

You are also welcome to suggest a striking image or visual abstract to illustrate your article. If you do please provide a jpeg file 550 px-wide x 400-px high.

10) A Conflict of Interest statement should be provided in the main text

11) Please note that we now mandate that all corresponding authors list an ORCID digital identifier. This takes <90 seconds to complete. We encourage all authors to supply an ORCID identifier, which will be linked to their name for unambiguous name identification.

Currently, our records indicate that the ORCID for your account is 0000-0003-4835-6228.

Link Not Available

12) The system will prompt you to fill in your funding and payment information. This will allow Wiley to send you a quote for the article processing charge (APC) in case of acceptance. This quote takes into account any reduction or fee waivers that you may be eligible for. Authors do not need to pay any fees before their manuscript is accepted and transferred to our publisher.

Photos 400-800 DPI

*Additional important information regarding figures and illustrations can be found at <https://bit.ly/EMBOPressFigurePreparationGuideline>

The system will prompt you to fill in your funding and payment information. This will allow Wiley to send you a quote for the article processing charge (APC) in case of acceptance. This quote takes into account any reduction or fee waivers that you may be eligible for. Authors do not need to pay any fees before their manuscript is accepted and transferred to our publisher.

***** Reviewer's comments *****

Referee #2 (Remarks for Author):

The authors have carefully and satisfactorily addressed all of my criticisms.

Referee #3 (Remarks for Author):

I am satisfied that the authors have made reasonable efforts to address the concerns that I raised.

However, I still have a minor correction to request. On page 17, they state "Transfection of cells causes lysosomal stress, which is known to activate TFE3/TFEB resulting in their nuclear localization (Sancak et al, 2010)". I don't dispute the idea that transfection causes lysosomal stress and that this would likely activate TFE3/TFEB but the reference that was provided did not show this. I therefore suggest that the authors state this idea in a more speculative manner rather than give the impression that the matter was definitively solved by Sancak et al, 2010.

Dear Jingyi Hou,

We hereby submit the re-revised version of our manuscript in which we made some small textual changes. As requested, we have kept the track mode on, and did not alter the changes already made by the data editor. We thank you for the careful handling of this manuscript and hope you now find our work acceptable for publication.

Sincerely,
Judith Klumperman

Comments from the editor

1. Figure EV1E: We understand that the parents have given written consent for the publication of clinical pictures. However, to minimize any potential risk of being exploitative, we would recommend adding a narrow black box to cover the eyes on the three panels. The sentence "Parents provided written informed consent for publication of clinical pictures" should be clearly stated in the paper.

We have added the requested black box to the patient pictures presented in Figure EV1. The informed consent from the parents is stated in the manuscript (p27).

2. Our data editor has made a couple of comments on your manuscript (see attached). Please address these issues in a .docx formatted version of the manuscript and keep the track mode on.

We have addressed the comments in the manuscript using track changes.

Reviewer comments

Referee #2 (Remarks for Author):

The authors have carefully and satisfactorily addressed all of my criticisms.

We kindly thank the reviewer for the positive evaluation of our work.

Referee #3 (Remarks for Author):

I am satisfied that the authors have made reasonable efforts to address the concerns that I raised.

We thank the reviewer for this comment.

However, I still have a minor correction to request. On page 17, they state "Transfection of cells causes lysosomal stress, which is known to activate TFE3/TFEB resulting in their nuclear localization (Sancak et al, 2010)". I don't dispute the idea that transfection causes lysosomal stress and that this would likely activate TFE3/TFEB but the reference that was provided did not show this. I therefore suggest that the authors state this idea in a more speculative manner rather than give the impression that the matter was definitively solved by Sancak et al, 2010.

We agree with the reviewer that indeed Sancak *et al*, 2010 do not show lysosomal stress caused by transfection with nuclear localization of TFEB/TFE3 as a result. We have slightly altered our statement and rephrased it to the following:

"Re-expression of VPS41^{WT} in HeLa^{VPS41KO} cells rescued the TFE3 phenotype after starvation and restimulation, whereas expression of VPS41^{S285P} or VPS41^{R662} had no effect, TFE3 was present in the nucleus in all conditions (Fig 5D, quantified in 5D'). Notably, transfection of cells affects membrane integrity causing lysosomal stress. This likely explains the nuclear localization of TFE3 in HeLa^{VPS41KO} cells transfected with VPS41^{WT} when cultured under steady state conditions (Fig 5D)." (p17).

Please note that we made these textual changes with the "track changes" mode on.

22nd Feb 2021

We are pleased to inform you that your manuscript is accepted for publication and is now being sent to our publisher to be included in the next available issue of EMBO Molecular Medicine.

We would like to remind you that as part of the EMBO Publications transparent editorial process initiative, EMBO Molecular Medicine will publish a Review Process File online to accompany accepted manuscripts. If you do NOT want the file to be published or would like to exclude figures, please immediately inform the editorial office via e-mail.

Please read below for additional IMPORTANT information regarding your article, its publication and the production process.

Congratulations on your interesting work,
Jingyi

Jingyi Hou
Editor
EMBO Molecular Medicine

Follow us on Twitter @EmboMolMed
Sign up for eTOCs at embopress.org/alertsfeeds

***** Reviewer's comments *****

*** ** IMPORTANT INFORMATION *** **

SPEED OF PUBLICATION

The journal aims for rapid publication of papers, using the advance online publication "Early View" to expedite the process: A properly copy-edited and formatted version will be published as "Early View" after the proofs have been corrected. Please help the Editors and publisher avoid delays by providing e-mail address(es), telephone and fax numbers at which author(s) can be contacted.

Should you be planning a Press Release on your article, please get in contact with embomolmed@wiley.com as early as possible, in order to coordinate publication and release dates.

LICENSE AND PAYMENT:

All articles published in EMBO Molecular Medicine are fully open access: immediately and freely available to read, download and share.

EMBO Molecular Medicine charges an article processing charge (APC) to cover the publication costs. You, as the corresponding author for this manuscript, should have already received a quote with the article processing fee separately. Please let us know in case this quote has not been received.

Once your article is at Wiley for editorial production you will receive an email from Wiley's Author Services system, which will ask you to log in and will present you with the publication license form for completion. Within the same system the publication fee can be paid by credit card, an invoice, pro forma invoice or purchase order can be requested.

Payment of the publication charge and the signed Open Access Agreement form must be received before the article can be published online.

PROOFS

You will receive the proofs by e-mail approximately 2 weeks after all relevant files have been sent to our Production Office. Please return them within 48 hours and if there should be any problems, please contact the production office at embopressproduction@wiley.com.

Please inform us if there is likely to be any difficulty in reaching you at the above address at that time. Failure to meet our deadlines may result in a delay of publication.

All further communications concerning your paper proofs should quote reference number EMM-2020-13258-V4 and be directed to the production office at embopressproduction@wiley.com.

Thank you,

Jingyi Hou
Editor
EMBO Molecular Medicine

Corresponding Author Name(s): Judith Klumperman, David Chitayat

Manuscript Number: #EMM-2020-13258